# Pluripotency reprogramming by competent and incompetent POU factors uncovers temporal dependency for Oct4 and Sox2

Vikas Malik [1,2,3], Laura V. Glaser [4], Dennis Zimmer[1,2,3], Sergiy Velychko[5], Mingxi Weng[6], Markus Holzner [6], Marius Arend[6], Yanpu Chen[1,3], Yogesh Srivastava [1,2,3], Veeramohan Veerapandian[1,2,3,7], Zahir Shah[2,3], Miguel A. Esteban[1,3,8,9], Huating Wang[10], Jiekai Chen[1,3,9], Hans R. Schöler[5,11], Andrew P. Hutchins[12], Sebastiaan H. Meijsing[4], Sebastian Pott[13] & Ralf Jauch[1,3,6]

Oct4, along with Sox2 and Klf4 (SK), can induce pluripotency but structurally similar factors like Oct6 cannot. To decode why Oct4 has this unique ability, we compare Oct4-binding, accessibility patterns and transcriptional waves with Oct6 and an Oct4 mutant defective in the dimerization with Sox2 (Oct4$^{defSox2}$). We find that initial silencing of the somatic program proceeds indistinguishably with or without Oct4. Oct6 mitigates the mesenchymal-to-epithelial transition and derails reprogramming. These effects are a consequence of differences in genome-wide binding, as the early binding profile of Oct4$^{defSox2}$ resembles Oct4, whilst Oct6 does not bind pluripotency enhancers. Nevertheless, in the Oct6-SK condition many otherwise Oct4-bound locations become accessible but chromatin opening is compromised when Oct4$^{defSox2}$ occupies these sites. We find that Sox2 predominantly facilitates chromatin opening, whilst Oct4 serves an accessory role. Formation of Oct4/Sox2 heterodimers is essential for pluripotency establishment; however, reliance on Oct4/Sox2 heterodimers declines during pluripotency maintenance.

[1] CAS Key Laboratory of Regenerative Biology, Joint School of Life Sciences, Guangzhou Institutes of Biomedicine and Health, Chinese Academy of Sciences, Guangzhou Medical University, 510530 Guangzhou, China. [2] University of Chinese Academy of Sciences, 100049 Beijing, China. [3] Guangdong Provincial Key Laboratory of Stem Cell and Regenerative Medicine; Guangzhou Institutes of Biomedicine and Health, Chinese Academy of Sciences, 510530 Guangzhou, China. [4] Max Planck Institute for Molecular Genetics, Ihnestraße 63-73, 14195 Berlin, Germany. [5] Department of Cell and Developmental Biology, Max Planck Institute for Molecular Biomedicine, 48149 Münster, Germany. [6] School of Biomedical Sciences, Li Ka Shing Faculty of Medicine, The University of Hong Kong, Hong Kong SAR, China. [7] Department of Developmental Biology, School of Basic Medical Sciences, Southern Medical University, 510515 Guangzhou, Guangdong, China. [8] Laboratory of RNA, Chromatin, and Human Disease, Key Laboratory of Regenerative Biology and Guangdong Provincial Key Laboratory of Stem Cells and Regenerative Medicine, Guangzhou Institutes of Biomedicine and Health, Chinese Academy of Sciences, 510530 Guangzhou, China. [9] Guangzhou Regenerative Medicine and Health Guangdong Laboratory (GRMH-GDL), 510005 Guangzhou, China. [10] Department of Orthopaedics and Traumatology, Prince of Wales Hospital, Li Ka Shing Institute of Health Sciences, The Chinese University of Hong Kong, Hong Kong, China. [11] Medical Faculty, University of Münster, 48149 Münster, Germany. [12] Department of Biology, Southern University of Science and Technology, 518055 Shenzhen, Guangdong, China. [13] Department of Human Genetics, The University of Chicago, Chicago, IL 60637, USA. Correspondence and requests for materials should be addressed to R.J. (email: ralf@hku.hk)

Defined cocktails of transcription factors (TFs) are capable of interconverting cell types from different lineages and to reprogram somatic donor cells to self-renewing pluripotent cells[1–3]. The induction of pluripotent stem cells (iPSCs) from fibroblasts by Oct4, Sox2, Klf4, and c-Myc (OSKM) is a widely used model system to study the underlying molecular mechanisms of cell fate conversion[4–8]. Reprogramming-capable TFs originate from diverse structural families without any obvious evolutionary relationship, and even closely related factors can vary widely in their reprogramming competency. Reprogramming capability is a function of special molecular features on TFs, but what these features are remains poorly understood (reviewed in ref. [9] and ref. [10]). For example, the reprogramming function of Sox2 and Oct4 appears to rely on subtle molecular underpinnings. Replacing Sox2 with other Sox factors such as Sox4, Sox7, or Sox17 obliterates the activity of reprogramming cocktails[11,12]. Similarly, Oct4 cannot be replaced by other paralogous POU (Pit-Oct-Unc) family members such as Oct6, Oct1, or Brn4 in standard retroviral reprogramming systems[12–15]. Further, Oct6 cannot substitute for Oct4 in a pluripotency maintenance assay[14,16]. POU III family members (Oct6, Brn2, Brn4) were used to transdifferentiate fibroblasts to post-mitotic neurons or neural progenitor cells[3,17–19]. Brn2 was reported to be unable to bind closed-chromatin in neural reprogramming[20] whilst Oct4 was regarded as a pioneer factor that can bind and possibly open up closed chromatin in human pluripotency reprogramming[21]. This suggests that at least one of the unique molecular features of reprogramming-competent versus incompetent factors is their ability to reconfigure chromatin.

We and others have begun to unravel the molecular basis for the disparate functions of Oct4 and Oct6[13,14,16,22,23]. As monomers, POU factors target the octameric ATGCAAAT consensus binding sites. Longer and more specific sequences can be targeted through DNA-dependent dimer formation with partner factors. For example, Oct4 pairs with Sox2 to bind the canonical SoxOct motif[24] or with Sox17 to bind the compressed SoxOct motif[25–27]. The interference with Oct4-Sox2 association by mutations that perturb the heterodimer interface abolishes reprogramming (Oct4$^{I21Y/D29R}$, henceforth termed Oct4$^{defSox2}$)[14,24,28] (Fig. 1a). Conversely, mutations in Sox7 and Sox17, that facilitate the interaction with Oct4 on the canonical SoxOct DNA element and reduce binding to the compressed SoxOct element, enable highly efficient reprogramming[11,25,26,29]. Homodimeric DNA recognition by POU factors is facilitated by the Palindromic Octamer Recognition Element (PORE, ATTTGAAATGCAAAT)[30] and the More-palindromic Octamer Recognition Element (MORE, ATGCATATGCAT)[31,32]. Oct4 has a preference to form heterodimers with Sox2 on the canonical SoxOct element whilst Oct6 preferentially homodimerises on the MORE element[14] (Fig. 1a). The MORE element has been associated with gene regulation in neural lineages by Oct6 and Brn2[22]. Modifying the interface that mediates binding of Oct6 to MORE contributes to the conversion of Oct6 into a pluripotency inducing factor but only if additional modification to the Sox2 interaction interface are introduced as well[14]. Together, molecular differences that set Oct6 and Oct4 apart have been characterized biochemically and functionally but how these differences affect gene regulation, chromatin binding and chromatin dynamics has remained unclear. To address this, we performed genome-wide analyses of Oct4, Oct6, and Oct4$^{defSox2}$ during somatic cell reprogramming to pluripotency.

We find that reprogramming cocktails with POU (Oct4-SK, Oct6-SK, and Oct4$^{defSox2}$-SK) and without POU factor (GFP-SK) can silence somatic genes but only the Oct4-SK cocktail can activate the pluripotency network. Although Oct6 binds to different sites than Oct4 during reprogramming, the Oct6-SK cocktail induces widespread chromatin opening including at Oct4 bound sites. Conversely, Oct4$^{defSox2}$ mostly targets a similar set of sites as Oct4 but interferes with the chromatin opening activity of Sox2 at Oct4-Sox2 co-bound sites containing SoxOct motifs. The presence of Sox2 in the reprogramming cocktail appears to be sufficient for bringing about most chromatin changes at the initial stages of reprogramming; whereas at later stage, the lineage safeguarding role of cooperative Oct4/Sox2 heterodimers is crucial to complete reprogramming.

## Results

**Oct4 is dispensable for somatic silencing.** To understand the unique role of Oct4 at different stages of somatic cell reprogramming, we first studied reprogramming outcomes and the gene expression dynamics. To minimize the heterogeneity of the analyzed cell populations, we used chemically defined iCD1 medium, three factor OSK (Oct4-Sox2-Klf4) cocktail[33] and mouse embryonic fibroblasts harboring an Oct4-GFP reporter (OG2-MEFs; referred to as MEFs hereafter)[34] (Fig. 1b). Under these conditions, about 8–10% of the plated MEFs reprogram in a highly synchronous manner giving rise to Oct4-GFP-positive and Nanog-positive colonies within 7 days (Fig. 1c, Supplementary Fig. 1A-B). iPSC lines derived with this system show a high rate of contribution to chimeric mice and germline transmission[33,35]. The cells pass through the MET (mesenchymal to epithelial transition)[36,37] at days 1–2 and the transgenic OG2 reporter is activated at day 3–4 (Supplementary Fig. 1A).

We compared cocktails expressing Oct4 with that of Oct6 and an Oct4$^{defSox2}$ mutant that is incapable to co-bind Sox2 on composite SoxOct DNA elements[14,24]. Both Oct6-SK and Oct4$^{defSox2}$-SK cocktails cannot generate GFP positive (GFP+) colonies in chemically defined medium (Fig. 1d) similar to serum/LIF conditions[14]. We next measured gene expression using bulk RNA-sequencing from MEFs (prior to reprogramming) and reprogramming cell intermediates at days 0, 1, 3, 5, and 8. We also included a POU-free control where Oct4 was replaced by retroviruses expressing GFP in a GFP-SK cocktail. At day 0, each cocktail showed nearly identical expression profiles that still clusters with the starting MEFs (MEF-like, Fig. 1e, Supplementary Fig. 1C). At day 1, the expression profiles remain similar for the four conditions but are distinct from MEFs.

At days 3 and 5 bulk expression profiles remain rather similar, however, by day 8, the expression profile of Oct4-SK is distinct from the other TF cocktails, and is closest to pluripotent cells (Fig. 1e, Supplementary Fig. 1C-D, Supplementary Table 1). RNA-seq was performed using unsorted cells explaining why day 8 Oct4-SK samples still show some differences to more homogeneous iPSC/ESC samples. Expression profiles for cells transfected with reprogramming incompetent POUs are similar at days 5 and 8 (Fig. 1e, Supplementary Fig. 1C). In all conditions, we observed a gradual loss of MEF identity (Fig. 1f, Supplementary Fig. 1E). However, Oct4$^{defSox2}$-SK, Oct6-SK, and GFP-SK conditions fail to activate early and late pluripotency genes. In the Oct6-SK condition, the activation of epithelial genes such as Cdh1 (encoding E-cadherin) is impeded. FACS analysis confirmed that from days 3–8 less than 20% of cells were E-cadherin positive in the Oct6-SK condition compared to 89% in Oct4-SK condition (Fig. 1g, Supplementary Fig. 2A–C). At days 3 and 5, E-cadherin levels in the Oct6-SK condition were even lower than in the conditions lacking POU factors (GFP-SK and SK conditions). Differential gene expression analysis using the GFP-SK condition as a reference showed a larger number of differentially expressed genes in the Oct4-SK condition by day 8 than for Oct6-SK and Oct4$^{defSox2}$-SK conditions (Supplementary Fig. 2D-E,

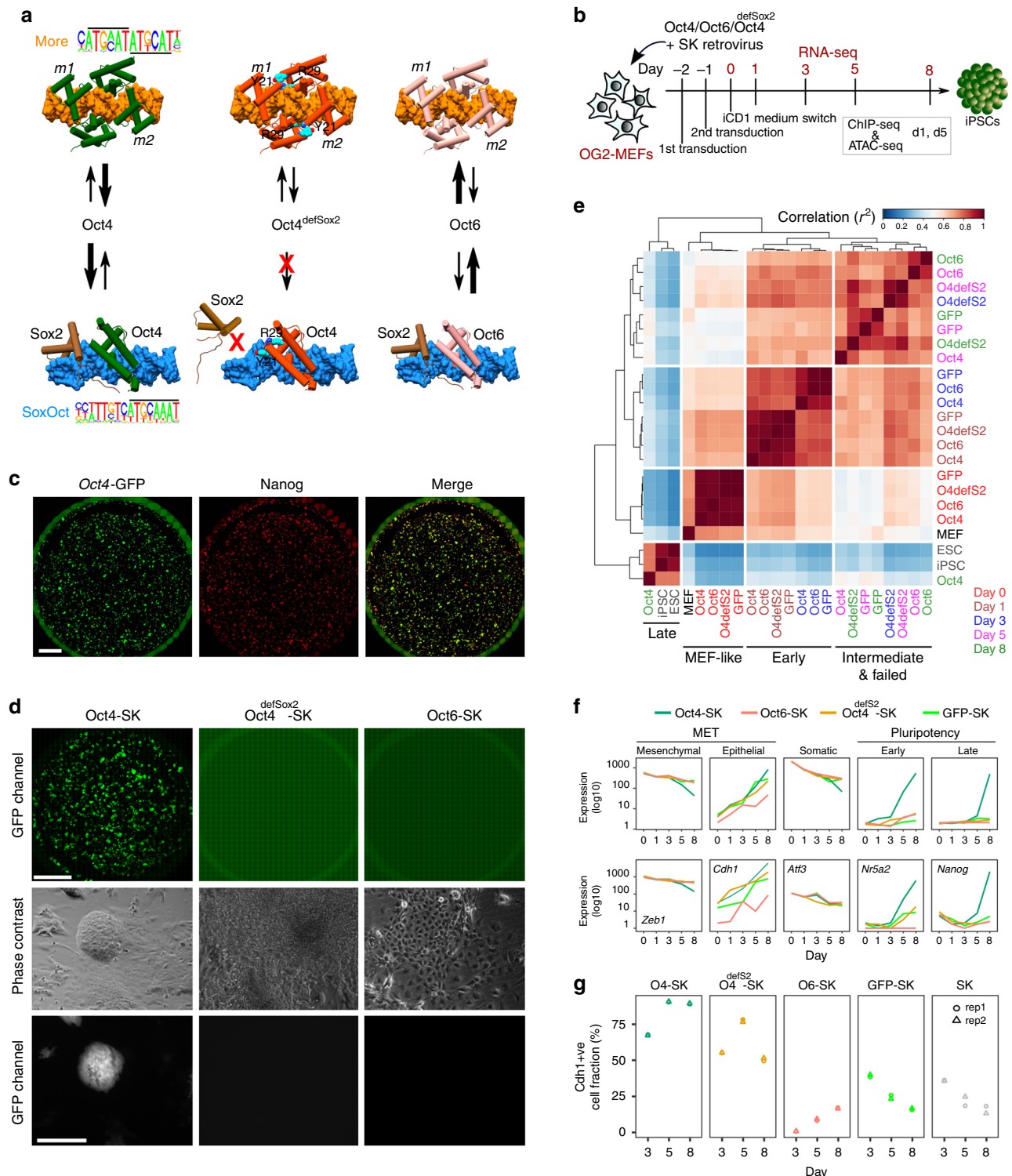

Supplementary Data 1). The upregulated genes in Oct4-SK at day 8 showed gene ontology terms enriched for embryo development, meiosis, blastocyst formation and DNA (de)methylation (Supplementary Fig. 2F). The set of genes upregulated by Oct6-SK showed enrichment of terms associated with somatic system development (e.g., circulatory and neuronal systems). Collectively, our data suggest that transcriptional responses at early stages of reprogramming do not require Oct4 but the induction of pluripotency genes are crucially dependent on Oct4, and related POU factors cannot substitute for this function.

**Cis-regulatory dynamics of Oct4 during reprogramming**. To link the expression profiles to genome occupancy, we performed ChIP-seq (Supplementary Fig. 3A-B, Supplementary Tables 2–3) and profiled the binding pattern of Oct4 at days 1, 5, and 7. We defined 7 trajectory groups (abbreviated with digits marking presence (1) or absence (0) of binding at days 1, 5, and 7) (Fig. 2a, Supplementary Data 2). De novo motif discovery on each trajectory group detected four types of POU motifs: the octamer; the composite SoxOct element; the palindromic MORE and a MORE + 1 variant (Fig. 2b–c). Around 50% of transiently bound sites

**Fig. 1** Reprogramming trajectories of cocktails containing Oct4, Oct6, Oct4$^{defSox2}$, or GFP in chemically defined medium. **a** The cartoons represent the preferences of the POU proteins for the DNA-dependent homodimerization on MORE versus heterodimerization with Sox2 on SoxOct elements determined using quantitative biochemical assays[14]. MORE DNA is shown in orange and SoxOct DNA in blue; m1 is monomer 1 and m2 is monomer2. The thickness of the arrows illustrates the DNA binding preferences. **b** Experimental design of reprogramming and RNA-seq, ATAC-seq, and ChIP-seq experiments. **c** Whole-well scans from a 6-well plate using GFP channel (*Oct4*-GFP), Nanog immunofluorescence and merged panels; for OG2-MEF cells reprogrammed with Oct4-SK at day 8; scale 5 mm. **d** Whole-well scans (upper panel) of wells from 12-well plate using GFP channel for three POU factors (Oct4, Oct4$^{defSox2}$, and Oct6); scale 5 mm. Representative phase contrast (middle panel) and corresponding *Oct4*-GFP fluorescence (lower panel) images of reprogramming experiments; time-point: day 8 post transduction; scale 200 μm. **e** Hierarchically clustered heatmap based on r2 correlation coefficients using RNA-seq reads as input. iPSCs and ESCs expression data are from GSE93029[42]. **f** Mean gene expression trajectories for indicated categories (upper panel) or a representative gene from each category (lower panel) in four reprogramming conditions. See Supplementary Fig. 1E for a larger panel of genes for each category. **g** Fraction of E-cadherin positive (Cdh1 + ve) cells at different stages of reprogramming in indicated cocktails. FACS was performed in technical duplicates ($n = 2$). Source data for FACS experiment are provided as a Source Data file

(100, 010) are devoid of any detectable composite POU motif and showed a stronger enrichment for somatic motifs potentially reflecting binding within accessible somatic enhancers (Fig. 2b–d). MORE motifs are most abundant at locations that are targeted early and remain occupied at subsequent time points (110, 111), whereas they are depleted in sites that are exclusively bound in the late stage ('001'). SoxOct motifs are enriched in locations occupied at late stages regardless of their binding pattern at earlier time points (Fig. 2b–e). Collectively, these analyses suggest that genome-wide Oct4 binding is dynamic (Fig. 2f, Supplementary Table 4), however the presence of MORE and SoxOct elements is associated with more persistent binding. To test whether MORE sites also possess enhancer activity, we selected two genomic sequences near the *Sox21* and *Spata13* genes that are constitutively bound by Oct4 and showed progressively increasing expression during reprogramming (Fig. 2g–h). First, we performed an EMSA (electrophoretic mobility shift assay) using probes with composite MORE (near *Spata13*) and SoxOct (near *Sox21*) elements (Fig. 2i, Supplementary Table 5). Oct4 binds the MORE$^{Spata13}$ but homodimer bands are only observed at high protein concentration suggestive of low homodimer cooperativity as reported previously for other MORE-like elements[14]. In contrast, on SoxOct$^{Sox21}$ DNA Oct4/Sox2 heterodimer bands predominate over monomeric bands indicating positive cooperativity (Fig. 2i). Both SoxOct$^{Sox21}$ and MORE$^{Spata13}$ sequences activated the expression of a GFP-reporter in mouse ESCs in a STARR-reporter assay[38] (Fig. 2j, Supplementary Tables 6–8). This activation was significantly reduced upon mutation of these DNA motifs indicating that both motif sequences are capable of mediating enhancer activity in ESCs. In summary, Oct4 binds chromatin dynamically using different motif signatures at different stages and predominantly acts as transcriptional activator.

**Switching MORE binding with a point mutation.** We had previously identified an amino acid at position 151 of the Oct4-POU which governs that Oct4 homodimerises less cooperatively than Oct6 on MORE DNA[14]. Exchanging this residue by generating mutant Oct4$^{S151M}$ and Oct6$^{M151S}$ proteins strongly increases homodimerisation for Oct4 and decreases homodimerisation for Oct6[14]. In four-factor (4F, OSKM) serum/LIF conditions the Oct6$^{M151S}$ mutation combined with mutations at the Sox2 interaction surface and the linker converts Oct6 into a pluripotency inducer[14]. To test whether the switch in motif preferences also occurs in a chromatin context, we performed ChIP-seq for Oct4$^{S151M}$ (Oct4SM) and Oct6$^{M151S}$ (Oct6MS) at day 1. Indeed, Oct4$^{S151M}$ shows an increased preference for MORE and MORE + 1 motifs (Supplementary Fig. 3C-F). Conversely, Oct6$^{M151S}$ loses its preference for the MORE element compared to Oct6. Yet, Oct6$^{M151S}$ does not show a marked gain in binding at SoxOct elements, in line with its inability to induce

pluripotency (Supplementary Fig. 3G). In the 4 F serum/LIF condition, we previously found that Oct4$^{S151M}$ reduces the iPSC colony yield to ~70% of that of Oct4[14]. In the 3 F iCD1 system used here, the iPSCs colony yield for Oct4$^{S151M}$ is reduced compared to Oct4 at day 4 but is similar at day 7 (Supplementary Fig. 3H). Accordingly, despite the binding to many MORE sites not targeted by Oct4, Oct4$^{S151M}$ remains bound to the critical set of Oct4 sites containing SoxOct motifs (Supplementary Fig. 3I).

**Oct4, Oct6, and Oct4$^{defSox2}$ show different motif preferences.** To explore the properties that endow Oct4 with the capacity to reprogram, we contrasted its binding profile with that of Oct6 and Oct4$^{defSox2}$ by ChIP-seq at days 1 and 5. All three exogenous POU constructs were expressed at comparable mRNA levels (Supplementary Fig. 4A, Supplementary Table 9). However expression of endogenous Oct4 was only activated by the Oct4-SK cocktail. Oct4, Oct6, and Oct4$^{defSox2}$ proteins were mainly present in the nuclear fraction with highest levels at day 1 followed by a moderate decrease over the subsequent days especially for Oct6 and Oct4$^{defSox2}$ (Supplementary Fig. 4B). The binding pattern of Oct4 at day 1 observed in this study correlates with previously reported early stages for secondary-mouse/Serum/LIF conditions, where cells reprogram slowly[4–6] (Fig. 3a). Oct4 binding at days 5 and 7 correlates with the maturation and pre-iPSCs stages in these systems[4,5]. Oct6 targets a unique set of genomic locations from the onset of its ectopic expression with only weak signals at sites occupied by Oct4 throughout the pluripotency reprogramming. Oct6 binding also does not correlate with POU III factors binding during neural reprogramming[18,20,22,39] (Fig. 3b–c, Supplementary Fig. 4C-D). The occupancy of Oct4$^{defSox2}$ correlates with Oct4 at day 1, but not at day 5 (Fig. 3a–c). Overlap of Sox2 binding in the respective conditions was only moderate with more uniquely bound than shared sites (Supplementary Fig. 4E-F). Motif analysis revealed that the canonical SoxOct element predominates in locations occupied by Oct4 and its fraction increases at the late stage (Fig. 3d–f). Surprisingly, Oct4$^{defSox2}$ also shows enrichment for the SoxOct motif at day 1, but at day 5 the MORE motif predominates, suggesting that Oct4$^{defSox2}$ can bind to composite SoxOct DNA. Oct6 preferentially targets MORE locations at both time points. Taken together, we found that the initial binding profile of Oct4$^{defSox2}$ resembles Oct4 whilst Oct6 binds a different set of locations, raising the question as to how these different binding landscapes correlate with chromatin state transitions.

**The Oct6-SK cocktail retains the capacity to open chromatin.** Three studies reported that Oct4 preferentially targets closed chromatin when reprogramming is initiated supporting the view that it acts as pioneer factor[6–8]. Work by two other laboratories concluded that Oct4 is directed to pre-opened and active

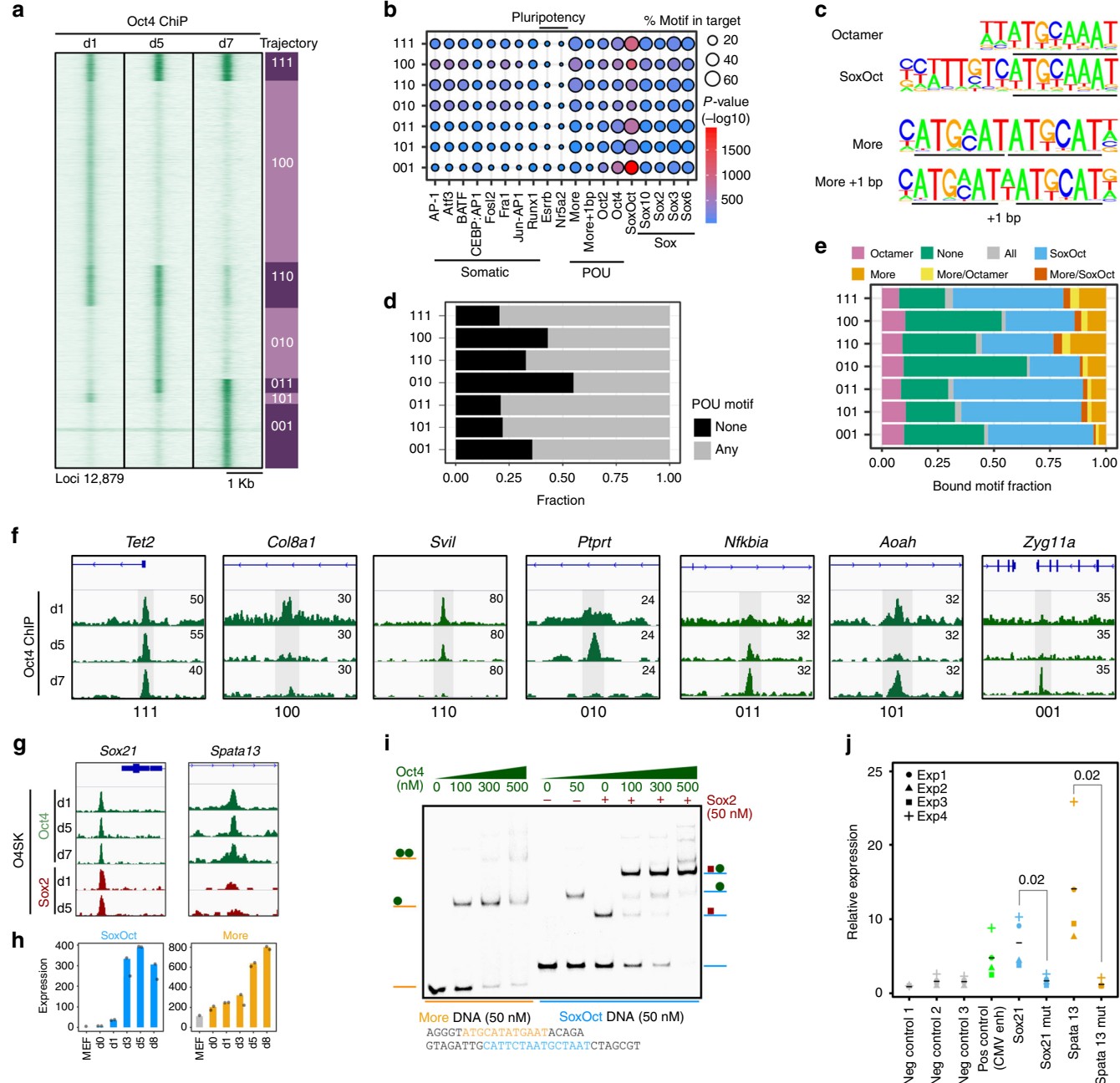

**Fig. 2** Determinants for the binding trajectories of Oct4. **a** Heatmap of Oct4 ChIP-seq signal within 2 kb window centered at Oct4 summits. Rows are Oct4 peaks arranged in 7 binary trajectories. 1 indicates presence and 0 absence of binding; left to right digits are days 1, 5 and 7. Summits were merged if within 100 bp. Genomic locations are provided in Supplementary Data 2. **b** Enrichment of selected TF motifs in Oct4 occupancy trajectories. Point size represents the proportion of sequences with the motif and color gradient (blue to red) the p-value score. **c** Consensus binding motifs of Oct4 represented as PWMs (position weight matrices). POU binding sites are underlined. **d** Proportion of ChIP-seq peaks in each occupancy trajectory featuring 'any' (consensus motifs shown in **c**) or 'none' of these POU motifs. **e** Fraction of binding locations for each occupancy trajectory containing the indicated POU motifs. 'All' refers to peaks where motif scanning concurrently detected all the POU motifs listed in (**c**) and 'none' the absence of any of these motifs. **f** Genome browser track of Oct4 ChIP-seq peaks (shaded gray) for selected trajectories. Genomic coordinates for the summits are listed in Supplementary Table 4. **g** Genome browser tracks of constitutively bound Oct4 and Sox2 ChIP-seq peaks containing either SoxOct (near *Sox21* gene) or MORE (near *Spata13* gene) motifs. **h** Gene expression (mean tag counts as bar and individual technical replicate as dots) of *Sox21* and *Spata13* in the Oct4 condition. **i** EMSAs using Oct4-POU and Sox2-HMG protein constructs and DNA probes containing SoxOct elements (near the *Sox21* gene) or MORE elements (near the *Spata13* gene). EMSA probes are provided in Supplementary Table 5. **j** STARR reporter assay[38] in ESCs with Oct4 bound regions from (**g**) near *Sox21* or *Spata13*. The positive and negative control sequences are shown in Supplementary Table 8. Each data point (n = 4, biological replicates) is shown with mean as black bar; indicated p-values were calculated by Student's t-test. Source data for **i** and **j** are provided as a Source Data file

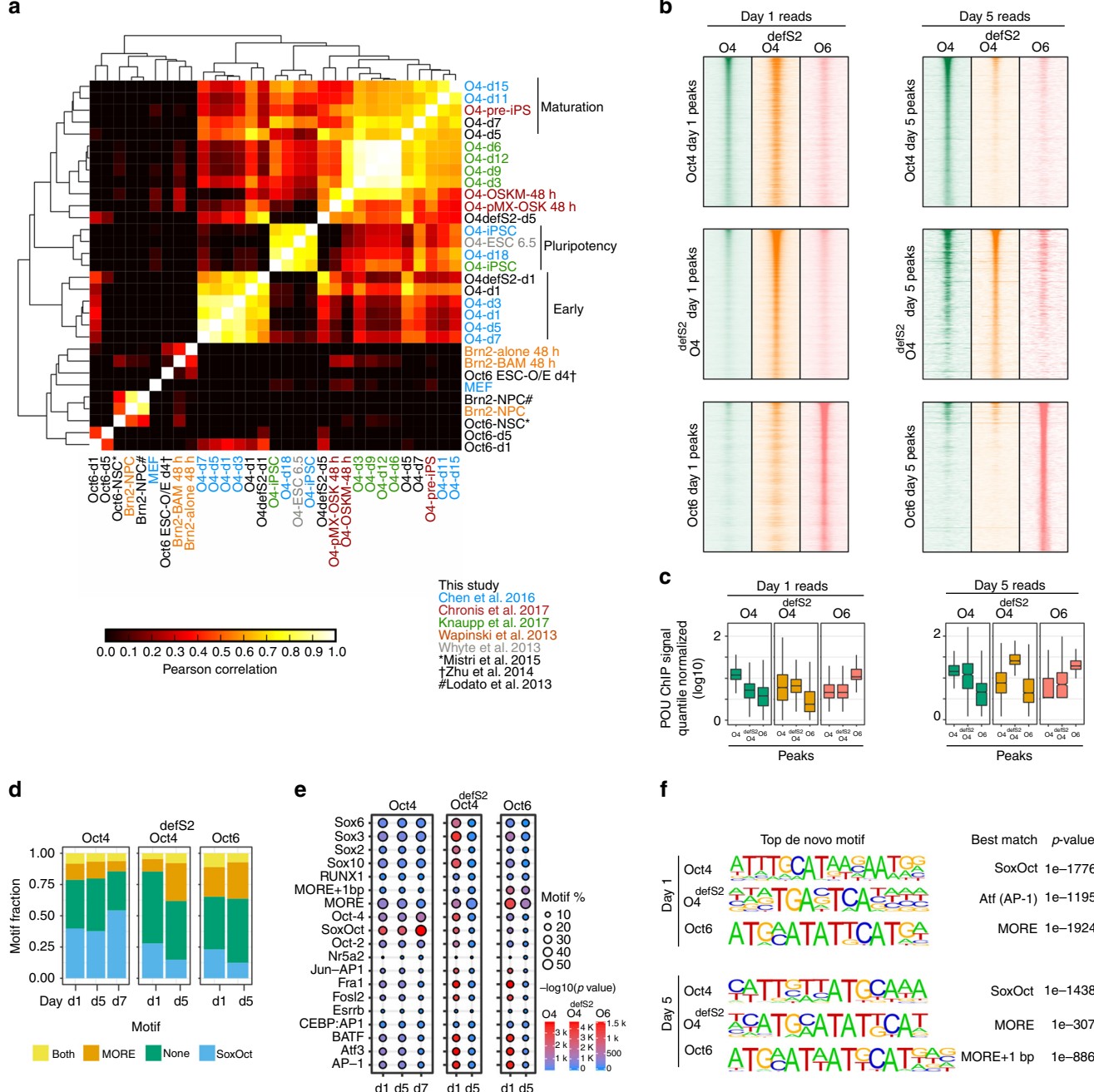

**Fig. 3** POU TFs exhibit distinctive binding profiles and motif preferences. **a** Hierarchical clustering of the pairwise correlation coefficients ($R^2$) of ChIP-seq signals from Oct4, Oct4$^{defSox2}$ and Oct6 ChIP-seq peaks with publically available ChIP-seq datasets for Oct4, Oct6 and Brn2[4–6,18,20,22,39,69]. **b** ChIP-seq signal heatmaps for Oct4 (green), Oct4$^{defSox2}$ (orange), and Oct6 (salmon) at days 1 (left panels) and 5 (right panels) centered on ChIP-seq peaks for Oct4 (top panels), Oct4$^{defSox2}$ (middle panels) and Oct6 (bottom panels). **c** Boxplots of quantile normalized ChIP-seq signals at Oct4, Oct4$^{defSox2}$ and Oct6 peaks at days 1 and 5 from heatmaps in (**b**). For the boxplots, the midline indicates the median, boxes indicate the upper and lower quartiles and the whiskers indicate 1.5 times interquartile range. **d** Fraction of binding locations containing MORE (including MORE variant with 1 bp spacer) or SoxOct elements at different reprogramming stages. 'Both' refers to peaks where motif scanning detected MORE and SoxOct motifs concurrently and 'none' the absence of either of the two motifs. **e** Enrichment of selected TF motifs in ChIP-seq peaks at days 1 and 5 for Oct4, Oct4$^{defSox2}$, and Oct6. Size represents fractional occurrences and color gradient the p-value scores. **f** Top de novo motifs for Oct4, Oct4$^{defSox2}$ and Oct6 at days 1 and 5. In the HOMER database the MORE motif is designated as Pit1 and MORE + 1 bp as Pit1 + 1bp[70]

enhancer controlling somatic genes[4,5]. To resolve these alternative views, we probed chromatin accessibility by ATAC-seq (assay for transposase-accessible chromatin using sequencing)[40] with Oct4-SK, Oct6-SK, and Oct4$^{defS2}$-SK cocktails at days 1 and 5 of reprogramming (Supplementary Fig. 5A). We used chromVAR[41] to test whether particular TF motifs are associated with

variations in chromatin accessibility across Oct4-SK, Oct6-SK, and Oct4$^{defS2}$-SK samples (Supplementary Fig. 5B). We found that the sites with canonical SoxOct motifs show an increased accessibility in Oct4-SK and Oct6-SK conditions at days 1 and 5 compared to MEFs, but not in the Oct4$^{defSox2}$-SK condition (Fig. 4a). However, loci with matches to single Sox motifs showed

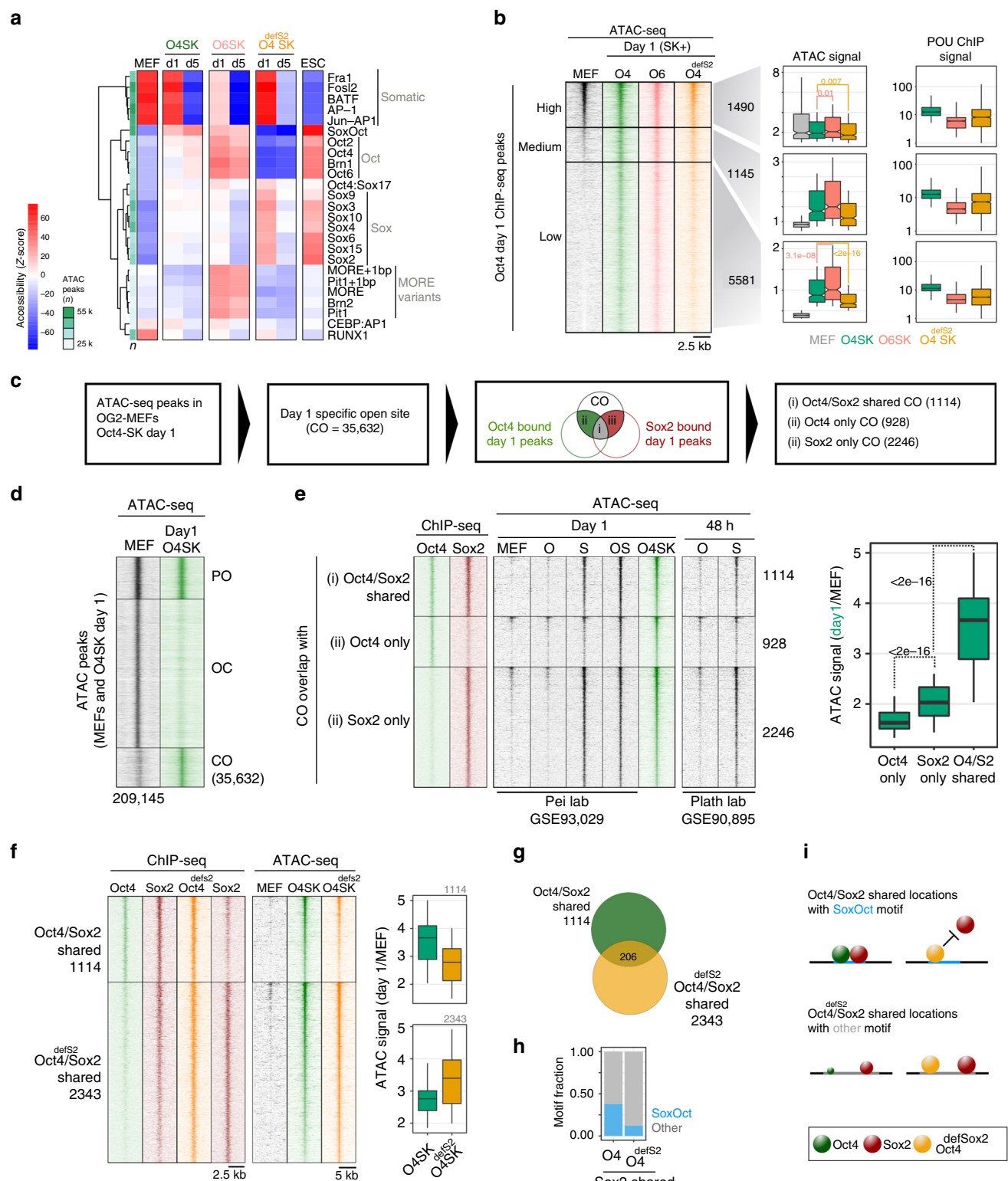

increased accessibility specifically in the Oct4$^{defSox2}$-SK sample. This suggests exogenous Sox2 or endogenously expressed Sox factors might bind and open alternative genomic sites in the Oct4$^{defSox2}$-SK condition. Oct6-SK expressing cells show a specific increase in accessibility at sites marked by MORE and octamer motifs (Fig. 4a). This suggests that the Oct6-SK cocktail is not compromised in its ability to open chromatin.

In order to link chromatin accessibility changes to Oct4 binding, we divided Oct4 bound sites based on their chromatin

accessibility levels in MEFs (high, medium, and low, Fig. 4b). The majority of Oct4 ChIP-seq peaks belong to the low accessibility group ($n = 5581$) with good ChIP-seq signals suggesting that Oct4 can effectively bind closed chromatin. The sites with low and medium accessibility in MEFs became more accessible in Oct4-SK and Oct6-SK conditions, but the increase in accessibility is significantly lower for the Oct4$^{defSox2}$-SK condition. This is inversely correlated with binding as the occupancy of Oct4$^{defSox2}$ is higher than the occupancy of Oct6 at these loci (Fig. 4b). Sox2

**Fig. 4** Sox2 facilitates chromatin opening and Oct4 augments it. **a** Accessibility variation of ATAC-seq peaks categorized by the presence of selected motifs was determined by chromVAR. Data are scaled row-wise. The color gradient denotes high (red) and low (blue) accessibility. $n$ = number of ATAC-seq peaks containing indicated motifs. **b** Oct4 ChIP-seq peaks at day 1 were ranked by ATAC-seq read coverage in MEFs and grouped into sites with high (read counts > 30, $n$ = 1490), medium (read counts = 15 to 30, $n$ = 1145) and low-accessibility in MEFs (read counts < 15, $n$ = 5581). Box plots show ATAC-seq and POU ChIP-seq signals at peaks from three corresponding groups. **c** Analysis flow chart to interrogate the role of Oct4 and Sox2 in facilitating chromatin opening. **d** ATAC-seq signal heatmaps grouped by three chromatin trajectories: PO (permanently open), OC (open to close), CO (close to open) defined using ATAC-seq peaks from MEFs (GSE93029[42]) and the Oct4-SK day 1 condition. Numbers indicate peak numbers. **e** The CO category was further grouped by the presence of Sox2 and Oct4 ChIP-seq peaks from day 1Oct4-SK condition. ChIP-seq and ATAC-seq signal heatmaps were generated using data from this study and indicated sources. Boxplots represent the day1 by MEF ATAC-seq signal ratio. **f** Heatmaps showing the ChIP-seq (left) and ATAC-seq (right) signals at ChIP-seq peaks defined by co-binding of Oct4/Sox2 or Oct4$^{defSox2}$/Sox2. Boxplots are day 1/MEF ATAC-signal ratios over the two peak sets. **g** Overlap of Oct4/Sox2 and Oct4$^{defSox2}$/Sox2 ChIP-seq peaks. **h** Fraction of Oct4/Sox2 and Oct4$^{defSox2}$/Sox2 peaks with matches to the canonical SoxOct motif. **i** Oct4$^{defSox2}$ prevents Sox2 from targeting sites with canonical SoxOct elements whilst co-binding of Oct4$^{defSox2}$/Sox2 at alternative locations is permitted. The size of spheres schematically represent binding preferences. $p$-values in **b** and **e** were calculated using the unpaired Wilcoxon rank sum test (R function pairwise.wilcox.test) adjusted for multiple testing using the Holm method. ATAC-seq boxplots show signals normalized using EAseq (DNA fragments per kilobase pairs (kbp) per million (m) reads) and POU ChIP-seq read coverage boxplots were quantile normalized. Genomic locations for **b**, **d**, **e**, and **f** are provided in Supplementary Data 2. For the boxplots, the midline indicates the median, boxes indicate the upper and lower quartiles and the whiskers indicate 1.5 times interquartile range

---

occupancy levels are comparable at the sites with high accessibility in MEFs among the three POU conditions, but it is reduced at sites with low accessibility in MEFs in the Oct4$^{defSox2}$ condition (Supplementary Fig. 5C).

Collectively, our data support a view that Oct4 pre-dominantly binds to locations closed in MEFs rather than having a bias for pre-opened MEF enhancers. Oct6-SK cocktails retain the ability to bring about chromatin accessibility changes at Oct4 bound sites and loci marked with SoxOct motifs whilst this activity is mitigated in the Oct4$^{defSox2}$-SK condition.

**Sox2 directs chromatin opening augmented by Oct4**. We next asked whether Sox2 or Oct4 is more potent at facilitating chromatin opening. To address this, we used ATAC-seq data in MEFs and at day 1 in the Oct4-SK condition and defined sites that undergo closed-to-open (CO) transition (Fig. 4c–d). We next defined CO locations co-bound by Oct4 and Sox2 or only bound by one of the factors (Fig. 4c–e). We found a significantly stronger increase in accessibility at sites bound by Sox2-only compared to sites bound by Oct4-only (Fig. 4e). Yet, the increase in accessibility was most profound in Oct4/Sox2-shared sites. To verify this finding, we downloaded publicly available ATAC-seq data where Sox2 and Oct4 were overexpressed alone or in pairs by two different laboratories[5,42]. Data from both studies revealed a stronger increase in chromatin accessibility when Sox2 is expressed alone (S) as compared to Oct4 alone (O) expression. We performed an analogous analysis with data from the Oct4$^{defSox2}$-SK condition and observed a stronger ATAC-seq signal for Sox2 only over Oct4$^{defSox2}$-only bound sites and a further elevated accessibility at Oct4$^{defSox2}$/Sox2-shared sites (Supplementary Fig. 5D–F). Thus, we conclude that Sox2 appears to be a better facilitator of chromatin accessibility changes than Oct4. Nevertheless, Oct4 augments chromatin opening concurrently with Sox2.

**Oct4$^{defSox2}$ impedes chromatin opening at SoxOct motifs**. We next sought to uncover differences between Oct4/Sox2 and Oct4$^{defSox2}$/Sox2 co-bound locations. Accessibility changes at Oct4/Sox2-shared and Oct4$^{defSox2}$/Sox2-shared sites correlate with binding in the two samples indicative of a direct causal relationship between binding and an increase in accessibility (Fig. 4f). Oct4$^{defSox2}$ (but not co-expressed Sox2) remains bound at Oct4/Sox2-shared sites and Sox2 (but not co-expressed Oct4) remains bound at Oct4$^{defSox2}$/Sox2-shared sites (Fig. 4f). The overlap between co-bound Oct4$^{defSox2}$/Sox2 and Oct4/Sox2 locations is low (Fig. 4g). Oct4$^{defSox2}$ has been designed to inhibit

DNA-dependent heterodimerisation with Sox2 in the context of canonical SoxOct element, but not when the half-sites are arranged differently (i.e., if the spacing is increased[24]). Consistently, motif analysis revealed that only 10% of Oct4$^{defSox2}$/Sox2-shared locations have matches to the canonical SoxOct motif compared to 40% Oct4/Sox2 co-bound locations (Fig. 4h). This suggests that Oct4$^{defSox2}$ interferes with Sox2 mediated chromatin binding and opening in the context of the SoxOct motif (Fig. 4i).

To further test this hypothesis, we first categorized genomic locations according to occupancy patterns using ChIP-seq peaks for Oct4, Oct4$^{defSox2}$, and Sox2 at day 1 and then inspected ATAC signals at day 1 across these categories (Supplementary Fig. 6A–C). We found elevated chromatin accessibility at sites bound by Sox2 compared to sites where it is not bound (Supplementary Fig. 6B–D). At sites co-bound by Oct4 and Oct4$^{defSox2}$, the ATAC-seq signal is stronger in the Oct4 condition compared to the Oct4$^{defSox2}$ condition whether or not Sox2 is present. Second, we inspected sites with a putative role in pluripotent cells and defined Oct4 bound sites that are closed in MEFs but open in ESCs. We split this set into sites with matches to the canonical SoxOct motif and sites lacking SoxOct motif (Supplementary Fig. 7A). We found find that opening at these sites is compromised in the Oct4$^{defSox2}$ condition in the subset of sites with SoxOct motifs (Supplementary Fig. 7A–B). At sites lacking SoxOct motifs, the difference in opening between Oct4 and Oct4$^{defSox2}$ conditions is less profound. Singly expressed Sox2 (S) shows a stronger accessibility increase than singly expressed Oct4 (O) or Klf4 (K) at sites with and without SoxOct motifs. The combination of Sox2 and Oct4 (OS) markedly increased accessibility (Supplementary Fig. 7A–D).

Sites with SoxOct motifs are most likely critical for conducive reprogramming. Oct4$^{defSox2}$ targets SoxOct sites and mitigates accessibility changes facilitated by Sox2 at these locations, but not at alternative sites with a different *cis*-regulatory architecture (Fig. 4i). In contrast, Oct6-SK opens SoxOct locations reminiscent to Oct4-SK and differences only become apparent for a subset of pluripotency enhancers at later reprogramming stages (Supplementary Fig. 7E–G).

**Oct4/Sox2 heterodimers are critical for persistent binding**. We further asked if the DNA motifs affect the initial Oct4/Oct4$^{defSox2}$ recruitment and persistence of binding. Oct4$^{defSox2}$ binds to many SoxOct sites also bound by Oct4 at day 1 (Fig. 2a, Supplementary Fig. 8A). We focused on sites constitutively bound by Oct4 (110, 111) and observed that Oct4$^{defSox2}$ effectively targets

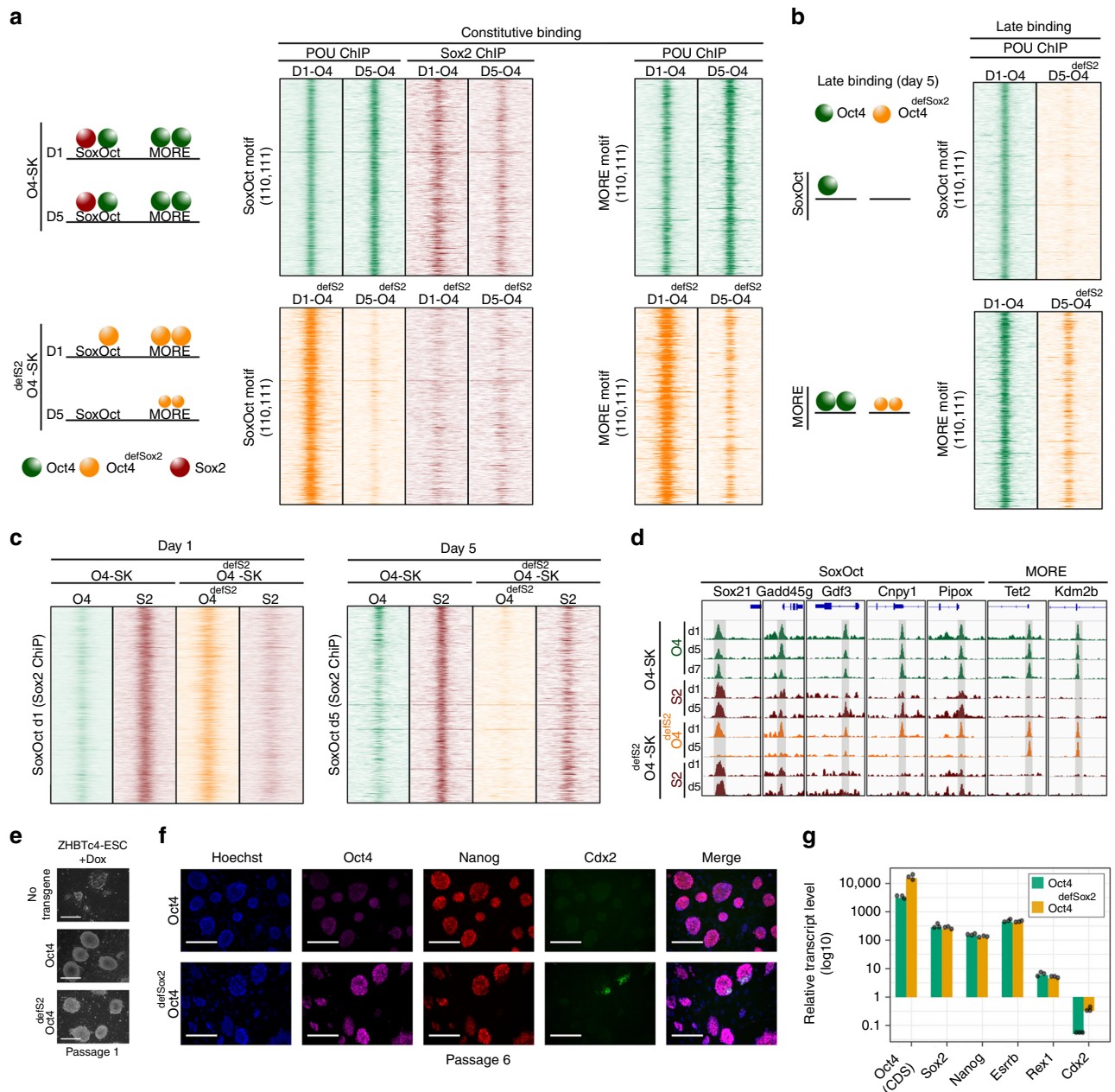

**Fig. 5** Oct4^defSox2 targets pluripotency enhancers but eventually derails. **a** Heat maps of ChIP-seq reads at days 1 and 5 for Oct4, Oct4^defSox2 and Sox2 in Oct4-SK (top panels) or Oct4^defSox2-SK (bottom panels) cocktails centered at Oct4 constitutively bound occupancy trajectories ('110' and '111') containing SoxOct motifs (left panels) or MORE motifs (right panel). A cartoon on the left summarizes the results. **b** ChIP-seq reads at day 5 for Oct4 and Oct4^defSox2 for Oct4 late bound occupancy trajectories '010' and '011' at SoxOct (top) or MORE (bottom) locations. **c** Heatmap of Oct4, Oct4^defSox2 and Sox2 ChIP-seq signals at days 1 and 5. Rows are Sox2 ChIP-seq peaks called in the Oct4-SK condition containing SoxOct motifs at day 1 (left panel) and day 5 (right panel). ChIP-seq signal heatmaps in **a**, **b** and **c** are normalized using EAseq (DNA fragments per kilobase pairs (kbp) per million (M) reads) and shown for 2 kb genomic intervals centered on midpoints or motifs. **d** Genome browser tracks of selected Oct4, Oct4^defSox2 and corresponding Sox2 ChIP-seq peaks are shown at different days of reprogramming. The nearest genes and motif under the peaks are mentioned on the top. Genomic coordinates for the summits are listed in Supplementary Table 4. **e** Phase-contrast images of conditional Oct4-knockout ZHBTc4[43] ESCs rescued with wild-type Oct4 or Oct4^defSox2 after culturing for 6 days in the presence of Dox (passage 1). Scale = 100 μm. **f** Fluorescence images of ZHBTc4 ESCs rescued with wild-type Oct4 or Oct4^defSox2 after 6 passages in the presence of Dox. Scale = 100 μm. The antibodies list is in Supplementary Table 11. **g** qRT-PCR analysis of selected gene expression of ZHBTc4 ESCs rescued with wild-type Oct4 or Oct4^defSox2 after 6 passages in the presence of Dox. The expression is relative to untransduced control induced with Dox for 7 days. Rpl37a was used as a house keeping gene. Individual data points are shown as black jitter plots (n = 3, technical replicates). The primers for Oct4 CDS (coding DNA sequence) and other markers are mentioned in Supplementary Table 10 and Source data are provided as a Source Data file

them at day 1 regardless of motif signatures (Fig. 5a, Supplementary Fig. 8B). However, by day 5 virtually all Oct4$^{defSox2}$ binding to SoxOct sites is lost, whilst residual binding to MORE sites persists (Fig. 5a). Sox2 effectively co-targets constitutively bound SoxOct sites in the presence of Oct4 but not in the presence of Oct4$^{defSox2}$ (Fig. 5a). Similarly, Oct4$^{defSox2}$ is unable to target SoxOct and sites devoid of POU motifs (none) in the Oct4 010 and 011 trajectories but retains some binding to MORE locations (Fig. 5b, Supplementary Fig. 8C). We next explored the occupancy of Oct4 and Oct4$^{defSox2}$ at sites containing SoxOct elements that were identified in Sox2 ChIP-seq data in Oct4-SK condition at days 1 and 5 (Fig. 5c). At day 1, Oct4$^{defSox2}$ effectively occupies these sites whilst the signal for co-expressed Sox2 was depleted. Yet, at day 5 only Oct4 showed co-occupancy with Sox2 in Oct4-SK condition but Oct4$^{defSox2}$ was not bound at these sites which now showed elevated signals for co-expressed Sox2 (Fig. 5c–d). We conclude that early Oct4 binding does not rely on heterodimer formation with Sox2, but it is critical for persistent binding and recruitment to sites at later stages of reprogramming.

**Oct4 does not require co-bound Sox2 to maintain pluripotency.** To elucidate the importance of Oct4/Sox2 heterodimers in the maintenance of pluripotency, we used ZHBTc4 ESC line carrying an *Oct4* transgene under the control of a tet-off promoter[43]. The addition of Dox leads to the depletion of the Oct4 protein after 24 h and trophectodermal differentiation (Fig. 5e). The exogenous introduction of Oct4 but not of Oct6 rescues pluripotency[14,16]. Surprisingly, Oct4$^{defSox2}$-expression could also rescue the maintenance of pluripotency (Fig. 5e, Supplementary Fig. 8D). ESCs expressing Oct4$^{defSox2}$ could maintain high expression levels of pluripotency markers, such as *Sox2, Nanog, Esrrb*, and *Rex1* even after 6 passages (Fig. 5f–g, Supplementary Table 10) indicating that the Oct4-Sox2 interaction might not be critical for pluripotency maintenance. However, in an analogous assay for Sox2, Sox2 mutants deficient in the DNA-dependent dimerization with Oct4 cannot rescue pluripotency[28]. This suggests that in the context of ESCs where Oct4 most likely binds already accessible targets, Oct4 alone is able to maintain an undifferentiated state. This is consistent with a report that Sox2 knockout ES cells could be rescued by the elevated expression of Oct4[44]. Oct4$^{defSox2}$ showed a higher transgene expression than cells expressing Oct4 (Fig. 5f–g). This indicates that in the absence of Oct4-Sox2 dimers, Oct4$^{defSox2}$ is required at a higher dosage than Oct4 for pluripotency maintenance. Yet, Oct4$^{defSox2}$ is a less potent suppressor of the trophectoderm lineage as indicated by elevated *Cdx2* expression and occasional Cdx2 + cells (Fig. 5f–g). We conclude that Oct4 is more critical than Sox2 in maintaining pluripotency and at elevated expression levels Oct4 alone can substitute for Oct4/Sox2 heterodimers.

**Oct6 binds loci without enhancer activity in ESCs.** To further delineate the reason for the non-redundant functions of Oct4 and Oct6, we defined fifteen occupancy groups for the binding patterns of Oct4 and Oct6 at reprogramming days 1 and 5 (Fig. 6a, Supplementary Data 2). Oct4 and Oct6 target a large set of genomic locations that are not shared (1000, 0010 where the first two digits indicate Oct4 binding (1) or absence (0) at days 1 or 5 and last two digits absence/presence of Oct6). The set of sites uniquely and transiently bound by Oct6 at individual days (0010, 0001) are depleted of known POU motifs but sites persistently bond by Oct6 are enriched for MORE motifs (Fig. 6b).

We next compared the co-binding of Oct6 or Oct4 with Sox2 in exclusive binding categories (1000, 0010, 1100, and 0011) and early shared binding (1010). Overall, Oct6 sites showed a weaker signal for Sox2 occupancy than Oct4 sites (Supplementary Fig. 9A) except for a subset of sites shared by both Oct4 and Oct6 at day 1 of binding (1010; Supplementary Fig. 9A). Sox2 peaks with matches to SoxOct motifs (when co-expressed with Oct4 or Oct6), had stronger Oct4 signals than Oct6 (Supplementary Fig. 9B–C).

In order to identify enhancers in ESCs, we performed STARR-seq (Self-transcribing active regulatory region sequencing)[38] assay containing regions of open chromatin isolated by formaldehyde-assisted isolation of regulatory elements (FAIRE)[45]. The STARR-input library showed a good correlation with DNase-seq (DNase I hypersensitive sites sequencing) performed in E14 ESCs[46] with 59% overlapping peaks (Supplementary Fig. 9D-E). The analysis for the overlap between STARR-seq regions occupied by Oct4 in ESC shows that around 50% (18,066) of Oct4 bound sites are covered by the STARR-input library and 1940 (11%) of these showed an overlap with active STARR-seq enhancers (Supplementary Fig. 9F).

We next analyzed the ESC enhancer activity of loci bound by Oct4 and Oct6 at days 1 and 5 containing either SoxOct or MORE motifs. We found that the representation in the STARR-seq library was higher for Oct4 bound regions (18% at day 1, 30% at day 5) than for Oct6 bound regions (9 and 14% for days 1 and 5, respectively; Supplementary Fig. 9G-H). Moreover, for Oct4 bound regions covered by the STARR-seq library, we found robust enhancer activity for SoxOct motifs and lower activity for MORE motifs (Fig. 6c, Supplementary Fig. 9H). The enhancer activity was highest for sites occupied at day 5, when cells gradually become more similar to ESCs. In contrast, Oct6 bound regions represented in the STARR-seq library are virtually devoid of enhancer activity, indicating that most of the Oct6 binding does not occur at ESCs enhancers. Despite the similar ability of Oct4-SK and Oct6-SK conditions to open chromatin, the genomic binding profiles of Oct4 and Oct6 are different. Oct6 binds regulatory regions that are not required for reprogramming and are without enhancer activity in ESCs.

## Discussion

In this study, we dissected the temporal dependencies of pluripotency reprogramming on chromatin binding, opening and gene regulation by Sox2 and Oct4. To gain detailed molecular insights into this process, we collected reference data by replacing Oct4 with two incapacitated POU factors, Oct6 and Oct4$^{defSox2}$. At day 1, the binding profile of Oct6 is distinct from Oct4 whilst accessibility changes particularly at Oct4-bound sites are similar (Fig. 6d). Yet the binding profile of the mutant Oct4$^{defSox2}$ resembles Oct4 but chromatin opening at SoxOct motifs is hampered. Sites with increased accessibility in the Oct4$^{defSox2}$-SK condition are enriched for single (non-composite) Sox motifs. This suggests that Oct4$^{defSox2}$ impedes accessibility changes in the context of the composite SoxOct element but it does not inhibit opening per se. Rather, Oct4$^{defSox2}$ perturbs the sequence of chromatin dynamics by interfering with Sox2 binding (or binding by endogenous Sox factors) on SoxOct elements. This might in turn cause Sox factors to bind ectopic sites. Sox2 alone can facilitate chromatin opening but Oct4 by itself has a limited potency to bring about accessibility changes (Fig. 4e). However, Oct4 can augment the chromatin opening by Sox2. Likewise, Oct4 binds to only a small fraction of active ESC enhancers at days 1 and 5 of reprogramming. This indicates that Oct4 does not engage pluripotency genes on-target and could explain why iPSC generation is rather slow and inefficient. Surprisingly, Oct4/Sox2 dimerization does not appear to be required for the initial target search of Oct4. Oct4$^{defSox2}$ can occupy many genomic locations similar to Oct4 at the onset of reprogramming (Fig. 3b).

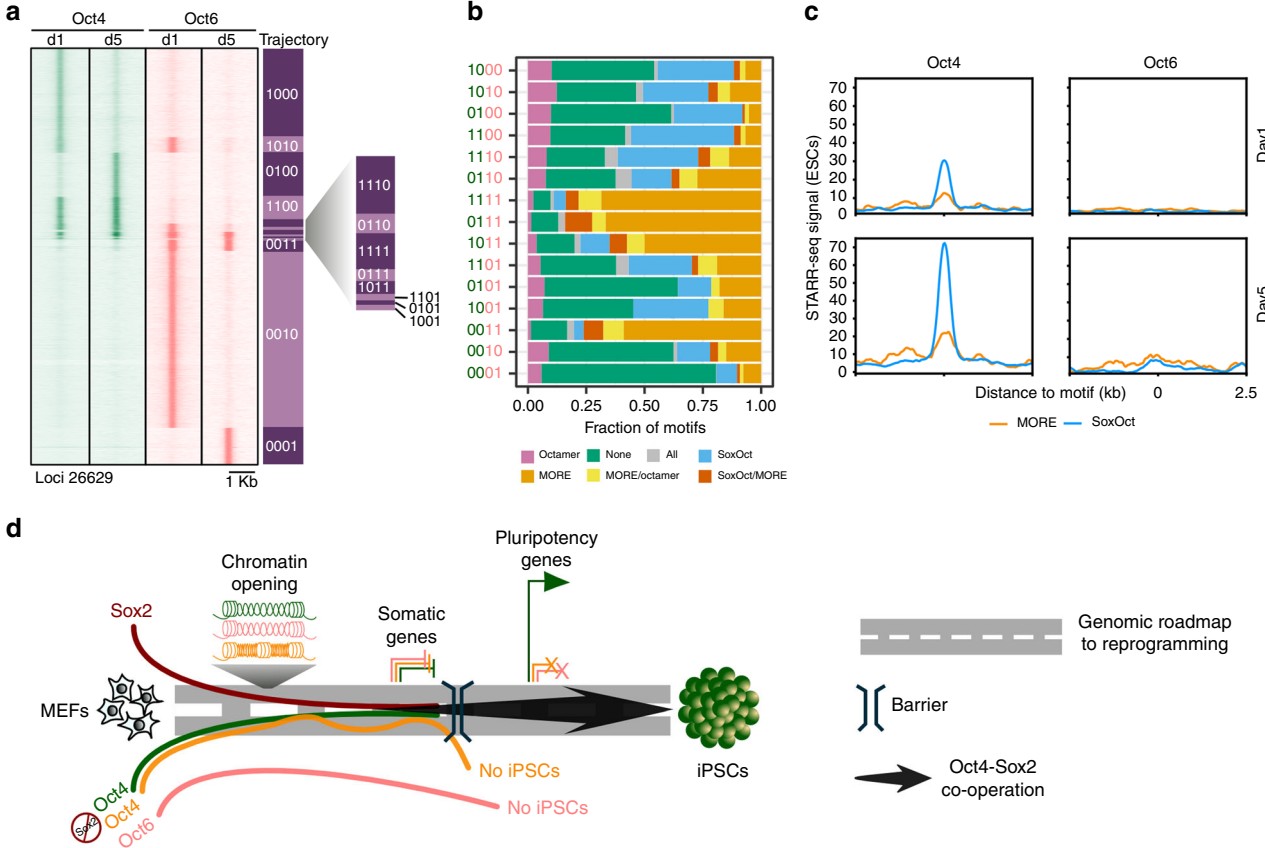

**Fig. 6** Oct6 binds to alternative targets and avoids pluripotency enhancers. **a** Comparison of Oct4 (green color) and Oct6 (salmon color) ChIP-seq signals at days 1 and 5 of reprogramming. Rows are 15 occupancy trajectories defined by the presence or absence of Oct4 or Oct6 peaks at day 1 or 5 (1 = binding, 0 = absence of binding). The summits were merged if they were within 100 bp. Genomic locations are provided in Supplementary Data 2. **b** Fraction of POU motifs mapping to binding sites in each of the occupancy trajectories. **c** Read pileup plots of STARR-seq signal (enhancer activity) in ESCs at Oct4 (left) and Oct6 (right) binding sites at days 1 (up) and 5 (bottom) of reprogramming containing either SoxOct or MORE motifs. **d** Model for unique and dispensable roles of Oct4 at different stages of somatic cell reprogramming

Evidently, Oct4$^{defSox2}$ is permitting transcriptional responses and exhibits binding patterns that correlate with aspects of Oct4-driven reprogramming. The uniqueness of Oct4 becomes evident at later stages of reprogramming. Due to its inability to partner with Sox2; Oct4$^{defSox2}$ will eventually fail to induce pluripotency (Fig. 6d). The dependence of the activation of the pluripotency network on Oct4/Sox2 heterodimers suggests that this heterodimer fulfills a role analogous to a lineage guardian without which faithful lineage specification from a plastic intermediary state cannot occur[47]. Whilst Oct4$^{defSox2}$ is incapable of inducing pluripotency, it was still found to retain the ability to maintain pluripotency. This indicates that the Oct4/Sox2 partnership is more important to induce the pluripotency than for its maintenance.

In the Oct6-SK condition, we observed an increased accessibility at ATAC-seq peaks marked by MORE or octamer motifs (Fig. 4a). Apparently, in addition to permitting opening at peaks containing SoxOct elements reminiscent to the Oct4-SK conditions, the Oct6-SK condition leads to additional opening presumably by the direct binding of Oct6. Therefore, our study does not support a model whereby Oct4 acquired unique features enabling it to act as pioneer factor in comparison to Oct6. Bulk gene expression profiles suggest that Oct4 is not required to silence somatic genes. The initial gene expression response appeared rather similar for all tested cocktails with one notable exception. Oct6-SK appears to counteract epithelialization demonstrated by a muted upregulation of E-cadherin compared

even to the two-factor SK conditions. As Oct6 mostly vacates Oct4 bound regions and prefers alternative binding elements (MORE) it is unlikely that it directly antagonizes Klf4, the key driver of the epithelial program[36,48]. We surmise that the restriction of the MET by Oct6 occurs indirectly through pathways that remain to be uncovered rather than by a direct competition with Klf4.

We show that Sox2 is critical for facilitating chromatin state changes, Oct4-Sox2 heterodimers are essential for the activation of pluripotency genes and Oct4 is the key factor for pluripotency maintenance (Fig. 6d). The initial cellular response to the overexpression of exogenous TFs appears to be dominated by binding that is not directly associated with reprogramming. Subtle modifications to the factor composition and specific mutations to reprogramming TFs can lead to drastically different outcomes of reprogramming experiments. We envisage this insight can be exploited to re-design TFs by protein engineering to improve reprogramming, transdifferentiation and forward programming for the fast and efficient generation of cells for regenerative medicine[49]. Understanding the sequence-structure-function relationships of reprogramming factors could enable to devise "one for all" factor libraries that would allow to screen and select a suitable mix of engineered TFs for any donor-target cell combination.

## Methods

**Cell lines and culture conditions.** Mouse embryonic fibroblasts (OG2-MEFs, Jackson laboratory, #004654) derived from E13.5 embryos carrying the *Oct4*

promoter driven *GFP* transgenic allele[34,50] were obtained from the core animal facility at GIBH. Animal care and experimental protocols were approved by the Guangzhou Institutes of Biomedicine and Health Ethical Committee. OG2-MEFs were maintained in high glucose Dulbecco's modified Eagle's medium (DMEM 4.5 g per L D-glucose, Thermo Fischer Scientific) supplemented with 10% fetal bovine serum (FBS, Natocor, #SFBE), 1× GlutaMax (Thermo Fischer Scientific #35050061), non-essential amino acids (1× NEAA, Thermo Fischer Scientific, #11140050). Retrovirus producing Plat-E cells[51] were maintained in DMEM containing 10% FBS. iPSCs were generated using the chemically defined iCD1 medium[33]. All the cells were cultured at 37 °C with 5% $CO_2$.

**Retrovirus infection and iPSCs induction.** Retroviral infection was performed as described[33,42] with some modifications. Briefly, Plat-E cells[51] were seeded at 6–7 × $10^6$ cells per 100 mm dish and cultured overnight followed by incubation with 10 μg of each of the three individual pMX based retroviral vectors along with 40 μg transfection reagent polyethyleneimine (PEI, Polysciences, #23966) dissolved in 1 mL Opti-MEM medium (Thermo Fischer Scientific, #31985070). Sox2 and Klf4, as a part of 3-F cocktail, were common in all samples while the 3rd factor was Oct4, Oct6, Oct4^defSox2 (Oct4^I21Y/D29R), Oct4^S151M, Oct6^M151S [14], or pMX-GFP. The volume was adjusted to 10 mL with DMEM + 10% FBS. The medium was changed within 10–14 h with fresh DMEM supplemented with 10% FBS, 1× GlutaMax and 1× NEAA. Virus containing supernatants were collected after 48 and 72 h, cleared with 0.45 μm filters (Millipore) and polybrene (Sigma, #40804ES76) was added to a final concentration of 8 μg per mL. OG2-MEF cells (3000–3500 cells per cm²) were cultured in 12 well dishes and transduced with 0.5 mL viral supernatant or using 100 mm dishes and 6 mL of each viral supernatant. After two infections at an interval of 24 h, the viral supernatants were replaced with iCD1 medium. The time point of change into iCD1 medium was defined as day 0 post-transduction (dpt). The cells were imaged on a regular basis with Axio Vert A1 microscope (Zeiss) and whole-well scans were taken at day 8 of reprogramming using an ImageXpress Micro XLS confocal High-Content Imaging System (Molecular Devices).

**Preparation of cytoplasmic and nuclear extracts.** MEF cells and reprogramming cells (3 × $10^6$ cells) were collected by trypsinisation at 0, 1, 3, and 5 days post transduction with retroviruses and washed twice with 1× PBS (Phosphate-buffered saline, Thermo Fischer Scientific, #14200–075). The outer cell membranes were lysed by incubation in 120 μL cytoplasmic extraction (CE) buffer (10 mM HEPES, 10 mM KCl, 0.1 mM EDTA, 1% IGEPAL CA-630, pH 7.9, freshly added protease inhibitor cocktail) for 15 min on ice and periodical vortexing. The lysate was centrifuged at 14,000 rpm for 5 min at 4 °C and the supernatant (cytoplasmic fraction) was carefully transferred into a new micro-centrifuge tube. The remaining pellet was washed once with CE wash buffer (10 mM HEPES, 10 mM KCl, 0.1 mM EDTA) and subsequently resuspended and incubated in 120 μL nuclear extraction (NE) buffer (20 mM HEPES, 0.4 M NaCl, 0.1 mM EDTA, 25% Glycerol, pH 7.9, 1% SDS) for 40 min on ice, with vortexing every 10 min for 10 s. The lysate was centrifuged at 14,000 rpm for 10 min at RT and the supernatant (nuclear fraction) was transferred into a new micro-centrifuge tube.

**Fluorescent western blot.** Cytoplasmic and nuclear fractions of OG2-MEF cells and reprogramming cells (Oct4, Oct6, and Oct4^defSox2; each on days 0, 1, 3, and 5) were used to perform western blots. SDS-PAGE sample loading buffer was added and samples were heated for 5–10 min at 90 °C and loaded onto SDS-polyacrylamide gel (stacking gel: 5% acrylamide, 125 mM Tris-HCl pH 6.8, 0.1% SDS, 0.1% APS, 0.15% TEMED; resolving gel: 12% acrylamide, 390 mM Tris-HCl pH 8.8, 0.1% SDS, 0.1% APS, 0.06% TEMED) and electrophoresed at 100 V for 120 min in SDS running buffer (25 mM Tris, 192 mM glycine, 0.1% SDS). Transfer was performed using Trans-Blot SD Semi-Dry Transfer Cell (Biorad) at 18 V for 60 min using PVDF membranes and transfer buffer (25 mM Tris, 192 mM glycine, 10% ethanol). The membrane was blocked in blocking buffer [PBS-T (Phosphate buffered Saline tween-20; 3.2 mM $Na_2HPO_4$, 0.5 mM $KH_2PO_4$, 1.3 mM KCl, 135 mM NaCl, 0.05% Tween-20, pH 7.4.) with 5% skinny fat milk] for 60 min and shaking at room temperature. Oct4, Oct6, and Actin antibodies were used at a concentration of 0.4 μg per mL in PBS-T with 5% BSA and the membrane was incubated overnight with rocking at 4 °C. On the next day, the membrane was washed three times for 10 min in 1× PBS-T. Multiplexed IRDye secondary antibodies (LI-COR) were used at a dilution 1:10,000 in 10 mL PBS-T with 5% BSA and the membrane was incubated for 60 min at room temperature. After washing with 1× PBST, the signals were detected using the 700 nm and 800 nm channels of the Odyssey CLx imaging system (LI-COR). List of antibodies used in this study are in Supplementary Table 11.

**Electrophoretic mobility shift assays (EMSAs).** EMSAs were performed using 5′ Cy5-labeled-dsDNA (Supplementary Table 5) as described previously[26]. Complimentary strands of DNA probes were annealed in equimolar concentration using 1× annealing buffer (20 mM Tris–HCl, pH 8.0; 50 mM $MgCl_2$; 50 mM KCl) by heating to 95 °C for 5 min followed by decreasing to 4 °C with a ramp-rate of 1 °C per min. Protein-DNA binding reactions were performed for 1 h in the dark at 4 °C in 1× EMSA buffer (20 mM Tris-HCl pH 8.0, 0.1 mg per mL bovine serum albumin, 50 mM $ZnCl_2$, 100 mM KCl, 10% (volume by volume) glycerol, 0.1%

(volume by volume) igepal CA630 and 2 mM β-mercaptoethanol). A 12% native 1× Tris-glycine (25 mM Tris pH 8.3; 192 mM glycine) polyacrylamide gel was pre-run at 100 V for 30 min at 4 °C. The protein-DNA complexes were separated at 4 °C for 30 min at 200 V. Cy5-labeled DNA bands were imaged using a Typhoon FLA7000 laser scanner.

**STARR reporter and STARR-seq assay.** Cloning: Reporter plasmids were generated by cloning either gBlocks (IDT) or fragments obtained by nested PCR from genomic ESC DNA into the STARR-seq screening vector (Addgene #71509[38]) using In-Fusion® HD Cloning Kit (Takara, Clontech). The DNA fragments used in reporter assays are listed in Supplementary Table 8. DNA fragments with described enhancer function in response to activation by the Glucocorticoid receptor, which is not expressed in stem cells, served as negative controls (nc1–3) for reporter activation and to assess basal promoter activity. The STARR-seq library containing DNA fragments isolated by formaldehyde assisted isolation of regulatory elements (FAIRE)[45] from E14 mouse ESC was obtained following the protocol described in ref. [38]. Reporter assays: To analyze individual reporter constructs, E14 mouse ESCs were cultured on gelatin-coated plates at a density of 2.5 × $10^4$ cells per well of a 24 well plate with ESC medium supplemented with 20% FBS and LIF (prepared in-house). The next day, cells were washed with 1 mL PBS and fresh 400 μL ESC medium was added. In addition, 100 μL transfection media containing 1 μg STARR (self-transcribing active regulatory region) plasmid constructs[38], 0.8 μL lipofectamine 2000 (Invitrogen) in opti-MEM medium was added to each well. Cells were collected after 24 h of transfection for FACS and qPCR analysis. For qPCR, RNA was isolated using the Qiagen RNeasy minikit. For cDNA synthesis, 500 ng RNA was used with gene-specific primers (GFP and Rpl19; Supplementary Table 6) using Takara PrimeScript RT reagent kit with following cycle conditions, 42 °C for 30 min, 85 °C for 5 s, 4 °C for storing. Reporter expression was quantified by qPCR specific for GFP and normalized to housekeeping gene (Rpl19) expression (primers in Supplementary Table 7). To test the DNA fragments obtained by FAIRE for their enhancer activity, 5 × $10^6$ E14 ES cells were transfected with 5 μg of the plasmid library using a Nucleofector™ 2b device (Lonza) with the Mouse ES Cell Nucleofector Kit (VAPH-1001, Lonza). For each biological replicate, 4 individual transfections were performed and subsequently combined for RNA isolation using the RNeasy Midi kit (Qiagen) 16 h after transfection. The polyA RNA fraction was enriched using Dynabeads Oligo(dT)$_{25}$ (Invitrogen), residual DNA was digested using Turbo DNase (Invitrogen) and finally RNA as cleaned-up with Agencourt RNAClean XP beads (Beckman Coulter).

**Real-time quantitative reverse transcription PCR.** MEF cells and reprogramming cells were washed once with PBS and lysed by the addition of TRIzol reagent with incubation of 5 min at room temperature. Chloroform (0.2 mL per 1 mL TRIzol) was added to samples and shaken vigorously for 15 s and incubated for 2–3 min at room temperature. The mixtures were centrifuged at 12,000×*g* for 15 min at 4 °C to separate them into three different phases. The upper aqueous phase was transferred into a new micro-centrifuge tube. An equal volume of 70% ethanol was added and vortexed. The samples were cleaned and eluted using PureLink RNA Mini Kit (*Ambion*) according to the manufacturer's instructions. cDNA synthesis was done using the ReverTra Ace qPCR RT Master Mix (*Toyobo*). qPCR reactions were conducted using the CFX96 system (Bio-Rad) and iTaq™ Universal SYBR Green Supermix (BioRad, #172–5121) in biological triplicates, each performed in technical triplicates, with a template amount of 100–150 ng cDNA for each reaction. The ΔΔ$C_T$ method[52] was used to analyze the cycle threshold ($C_T$) values. We used *Actin* (Supplementary Fig. 4A) or *Rpl37a* (Fig. 5g) as reference genes and, respectively, MEFs (passage 1) or ZHBTc4-ESCs cultured with Dox for 7 days as calibrator samples.

**ChIP-sequencing.** Reprogramming cells (7.5–10 × $10^6$ per sample) were collected by trypsinisation at 1, 5, and 7 dpt and washed twice with PBS (Thermo Fischer Scientific, #14200–075). Cells were resuspended in PBS at a density of 5 × $10^5$ per mL and then chemically crosslinked by the addition of 1% formaldehyde (Thermo Fischer Scientific, #28908) for 10 min at room temperature with gentle rotation. Crosslinking was stopped by the addition of glycine (Sigma, #G8790) to a final concentration of 125 mM for 5 min at room temperature. Cells were washed twice with 10 mL cold PBS and lysed in 10 mL Lysis Buffer 1 (150 mM NaCl, 0.5% NP40; 1 mM EDTA, 10% glycerol, 0.25% triton-X100, freshly added complete proteinase inhibitor cocktail) in 15 mL tubes at 4 °C for 10 min with gentle rocking. The lysate was centrifuged for 10 min at 1350 × *g* at 4 °C. The supernatant was discarded and the pellet was resuspended in 10 mL lysis buffer 2 (10 mM Tris-HCl, pH 8.0, 200 mM NaCl, 1 mM EDTA, 0.5 mM EGTA, protease inhibitor) and incubated at 4 °C for 10 min. The procedure was repeated using 10 mL lysis buffer 3 (1% SDS, 10 mM EDTA, 50 mM Tris-HCl, pH 8.0, protease inhibitor) and 10 min rocking at RT. Nuclei were collected by centrifugation and washed twice with sonication buffer (10 mM Tris-HCl, pH 8.0, 0.5 mM EDTA, protease inhibitor) and finally resuspended in sonication buffer. Samples were sonicated using a Covaris S220 device with default settings to obtain DNA fragments with a length from 150 to 500 bp using 130 μL microTUBEs (Covaris, #520045) for 5–8 cycles with 30 s ON and 60 s OFF. 10% Triton-X 100 (Sigma, #T8787) was added and the lysate was centrifuged at high speed (20,000 × *g* for 10 min at 4 °C). 1 by 10th of the

supernatant was stored as input control. ChIP was performed in an automated fashion using the IP-Star compacted automated system (Diagenode) using 10 μg ChIP-grade antibodies (Supplementary Table 11) for Oct4 (Santa Cruz Biotechnology, # Sc-8628X), Oct6 (Santa Cruz Biotechnology, # Sc-11661X) or 10 μg Sox2 (Santa Cruz Biotechnology, # sc-17320 X or Cell Signaling Technology, #2748 s) and 50–100 μL protein G Dynabeads (Thermo Fischer Scientific, #10004D). The coupling time for beads and antibody was 5 h followed by incubation with lysate for 15 h at 4 °C. Automated wash steps of 10 min at 4 °C were performed in the following order: low salt buffer (20 mM Tris-Hcl pH 8.0, 0.1% SDS 1% TritonX-100, 2 mM EDTA, 150 mM NaCl), high salt buffer (20 mM Tris-HCl pH 8.0, 0.1% SDS 1% TritonX-100, 2 mM EDTA, 500 mM NaCl), LiCl buffer (500 mM LiCl, 50 mM Tris-HCl pH 8.0, 2 mM EDTA, 1% NP-40, 0.5% Na-deoxycholate), TE buffer (10 mM Tris-HCl pH 8.0, 1 mM EDTA) with final elution in 10 mM Tris-HCl pH 8.0, 1 mM EDTA, 1% SDS buffer. The eluted chromatin and input samples were reverse-crosslinked by overnight incubation at 65 °C. Samples were treated with RNaseA (0.2 mg per mL final concentration, Sigma, #R6513) at 37 °C for 2 h followed by proteinase K (0.2 μg per mL final conc., Thermo Fischer Scientific, #25530–049) at 55 °C for 3 h. DNA was extracted by phenol:chloroform:isoamyl alcohol (25:24:1, (Fluka, #77617) and phase separated using heavy phase-lock tubes (TianGen, #WM5–2302831) by centrifugation at 10,000 rpm for 10 min at RT and precipitated with 100% ethanol and 20 μg per μL glycogen (Roche, #901393) as co-precipitant. The precipitate was washed with 80% ethanol and finally eluted in 20 μL water. DNA concentration was determined using Qubit dsDNA HS assay kit (Thermo Fischer Scientific, # Q32856).

**RNA-sequencing.** Reprogramming cells with cocktails containing SK along with either of the following factors Oct4, Oct6, Oct4[defSox2], and GFP were washed twice with 1× PBS and collected by adding 250 μL TRIzol (Thermo Fisher Scientific, #12183555) per well of a 12 well plate at days 0, 1, 3, 5, 8 in replicates. In addition, MEF cells were also collected. Total RNA was isolated and used for library constructions with TruSeq Nano RNA sample prep kit (Illumina). Sequencing was carried out on the Illumina HiSeq 2500 (Illumina). Each library was paired-end with a 150 bp read length. Both RNA and ChIP sequencing were carried out at Wuxi Next Code (https://www.wuxinextcode.cn/) and reads were provided in fastq format for data analysis.

**Assay for transposase-accessible chromatin using sequencing.** ATAC-seq was performed in replicate following this protocol[40,53]. Briefly, 50,000 cells (reprogrammed with Oct4-SK, Oct6-SK and Oct4[defS2]-SK retroviruses) were collected at days 1 and 5 of reprogramming and processed by Annoroad Gene Technology (http://en.annoroad.com/). Briefly, cells were washed once with 50 μL of cold PBS; centrifuged at 500 × g for 5 min at 4 °C and cell pellets were resuspended in 50 μL cold ATAC lysis buffer (10 mM Tris-HCl pH 7.4, 10 mM NaCl, 3 mM MgCl₂, 0.1% IGEPAL CA-630) by slowly pipetting up and down and immediately spun down at 500×g for 10 min at 4 °C to collect nuclei. Nuclei were washed in 1× PBS and subsequently re-suspended in 50 μL transposition reaction mix (25 μL 2× TD reaction buffer, 22.5 μL nuclease-free water, 2.5 μL Tn5 transposase) of Nextera DNA library preparation kit (Illumina, FC-121–1030). Samples were incubated at 37 °C for 30 min and DNA was isolated with minElute Kit (Qiagen). The transposed DNA was then amplified with custom primers as described for 1 cycle of 72 °C for 5 min, 98 °C for 30 s followed by 5 cycles of 98 °C for 10 s, 63 °C for 30 s, 72 °C for 1 min. To determine the suitable number of cycles required for the next round of PCR, the library was assessed by quantitative PCR as described in ref. [53]. Libraries were analyzed using the Bioanalyser high sensitivity DNA analysis kit (Agilent) followed by paired-end sequencing with the length of 150 nucleotides.

**Pluripotency maintenance assay.** ZHBTc4 ESCs are engineered tet-off cells with conditional repression of *Oct4* in the presence of doxycycline (Dox), allowing to test the *Oct4*-knockout rescuing ability by exogenously provided factors[43]. The addition of Dox leads to the removal of the Oct4 protein after 24 h and trophectodermal differentiation. The introduction of exogenous wild-type Oct4 with a constitutive EF1α promoter rescues pluripotency in the presence of Dox. ZHBTc4 ESCs were cultured in the absence of Dox in ESC medium (high-glucose DMEM medium (Life Technologies) supplemented with 15% KSR, 2% FCS, 1% Glutamax, 1% NEAA, 1% penicillin-streptomycin, 1% β-mercaptoethanol, and 20 ng per mL human recombinant LIF (purified in-house at the Max Planck Institute for Molecular Biomedicine) were split on gelatin-coated dish and simultaneously infected with non-concentrated lentiviral pLVTHM-Oct4 or pLVTHM-Oct4[defSox2] under constitutive EF1α promoter. After 48 h of infection, the cells were washed with PBS and split on C3H mitotically inactivated feeder cells in ESC medium in the presence of 2 μg per mL Dox (Sigma, #D9891). The cells were imaged after 6 days of culture (passage 1), and consequently passaged every 3–4 days in the presence of feeders, in 1 to 50 or 1 to 10 density for wild-type Oct4 or Oct4[defSox2], respectively. For RNA preparation for the qPCR experiment, the cells were passaged on gelatin-coated plates with no feeders in the presence in 2i medium [3 μM CHIR99021 (Selleck; #S2924), 1 μM PD0325901 (Selleck; #S1036)[54]]. For immunostaining, $1.6 \times 10^4$ cells per well were passaged in a 24 well plate with C3H feeders, cultured in ESC medium supplemented with 2 μg per mL Dox for 3 days. Briefly, cells were fixed with PFA, permeabilised with 0.1% TritonX100, blocked

with 5% BSA, incubated with primary antibodies overnight, washed with PBST, and incubated with donkey anti-mouse, anti-rat, and anti-goat Alexa-fluorophore-conjugated secondary antibodies for 1 h (Supplementary Table 11).

For alkaline phosphatase (AP) staining, the passage 1 rescued cells (after 6 days of Dox treatment), were fixed with paraformaldehyde (PFA) and stained with fast red pre-mixed with naphtol (Sigma; 855–20 mL).

**Flow cytometry.** MEFs and reprogramming intermediates at days 3, 5, and 8 were dissociated with trypsin. Cells were collected by centrifugation at $300 \times g$ for 5 min and kept on ice hereafter in polystyrene test tubes ($12 \times 75$ mm, non-sterile). Cells were washed twice with 1 mL FACS buffer (1× PBS, 5% fetal bovine serum, 20 mM HEPES pH 7.2–7.5). Cells were resuspended and blocked in blocking buffer (FACS buffer containing 5% normal mouse serum (Gemini bio-products, #100–113)) at a density of $1 \times 10^6$ cells per 100 μL and pre-incubated on ice for 10 min. Cdh1 antibody (Thermo Fisher Scientific, #50–3249−82; 0.5 μg per sample, Supplementary Table 11) or Mouse IgG1, κ Isotype Control (BD biosciences, #555746; 5 μg per sample) was added directly to cells in blocking buffer and incubated for 20–25 min in dark on ice. The unbound antibodies were washed twice with 1 mL of cold FACS buffer. DAPI (BD biosciences, #564907) was used to exclude dead cells. Cells were passed through a 40 μm cell strainer (Corning, # CLS431750–50EA) before analysis on LSRFortessa machine (BD biosciences). The gating strategy is shown in Supplementary Fig. 2B. Analysis was done with the FlowJo V10 software.

**Immunocytochemistry.** Cells reprogrammed with Oct4-SK were washed three times with PBS (1×) and fixed with 4% paraformaldehyde for 15 min at room temperature. Cells were permeabilised by incubation with 0.2% Triton X-100 for 15 min in 1× PBS at room temperature. The permeabilised cells were washed twice with 1× PBS and blocked with 5% BSA in 1× PBS for 1 h. Cells were incubated with primary Nanog antibody (Novus; #NB100–58842, 1:500, Supplementary Table 11) at 4 °C overnight. Cells were washed three times with 1× PBS and incubated with Alexa Fluor 594 Donkey anti-Rabbit IgG (H + L) secondary antibody (Invitrogen; #A21207, 1:1000) in the dark at room temperature for 1 h. Cells were washed three times with 1× PBS and further stained with 1× DAPI (Thermo Fisher Scientific; #R37606) and imaged with an Olympus CKX53 inverted fluorescence microscope. Whole-well scans were obtained by an ImageXpress Micro Confocal High-Content Imaging System (Molecular Devices).

**RNA-seq data processing.** The mouse transcriptome index was generated with RSEM[55] using the reference genome mm10 (ensemble gene annotation track v74). 150 bp paired-end raw reads were aligned to the mm10 transcriptome using RSEM and bowtie 2.0 (with options "rsem-calculate-expression -p 12–bowtie2–no-bam-out–paired-end")[56]. Per-gene read counts were calculated using RSEM and were GC normalized using EDAseq[57]. A threshold of GC-normalized read count of 20 in any two samples was applied to keep a gene, leading to a final set of 15,026 genes. This read count was used for differential gene expression analysis using DESeq2[58], to inspect the expression of individual genes and to perform global clustering with correlation heatmaps and principle component analysis in glbase[59]. PCA was generated using the get_pca function and correlation heatmaps were generated using correlation_heatmap function with option mode = r2 in glbase. The R packages data.table and ggplot2 were used for data analysis and visualization.

**STARR-seq data processing.** Sequencing libraries were generated using the Illumina Truseq dual index system with unique molecular identifier (UMI) containing primers for cDNA generation and barcoded PCR amplification. Libraries were sequenced with HiSeq 2500 (Illumina) to generate 50 bp paired-end reads. Sequencing reads were aligned to the mouse genome (mm10) using Bowtie2[56] (-X 800 –fr–very-sensitive). UMI-tools[60] was used for UMI-aware removal of PCR duplicates. Samtools[61] was used to filter reads for proper pairs, alignment and quality scores (-h -b -f 3 -F 780 -q 5), to select reads mapping only to regular chromosomes (chr1–19, chrX and chrY), and to remove reads mapping to blacklisted regions (ENCODE accession ENCFF547MET). Accessible regions covered by the input library were identified using MACS2[62] (-q 0.05–keep-dup all–call-summits–bw 200). Significantly active enhancers were called using MACS2 with the same settings but using the input library as control. The analysis was performed for each biological STARR-seq replicate individually, as well as for the merged reads from three replicates. Finally, peaks were only counted as true STARR-seq enhancers when they were called for the merged reads and also called for at least two of three replicates. Normalized STARR-seq signal for data visualization was generated using bamCompare of the deepTools package[63] (-of bigwig–operation subtract -bs 10 -e–normalizeUsing RPKM–effectiveGenomeSize 2652783500) and replicate-merged STARR-seq reads were normalized to Input library reads, as well as to sequencing depth.

Heatmaps which show STARR-seq signal distribution at selected regions were generated using computeMatrix (reference-point mode) and plotHeatmap tools of the deepTools package[63].

To assess Oct4 or Oct6 bound regions for their STARR-seq activity, at day1 and day 5 of reprogramming, ChIP-seq peaks were filtered for sites, which are represented in the STARR-seq library using bedtools intersect (Supplementary

Fig. 9G). Next, these overlapping regions were grouped by enriched sequence motifs (described in the motif analysis section) and for each group the input-normalized STARR-seq signal was plotted (Supplementary Fig. 9H).

To show STARR-seq enhancer activity distribution at Oct4 bound regions in mESCs, STARR-seq signal (not input normalized) was plotted at Oct4 bound sites (GSE90895[5]). Oct4 sites were ranked for mean STARR-seq signal and subsequently STARR-seq input signal was plotted for these ranked Oct4 sites (Supplementary Fig. 9F).

For the genome wide correlation analysis of read coverages, comparing FAIRE-STARR-seq with DNase-seq (GSM1014154[46]) coverage (Supplementary Fig. 9D), multiBamSummary of the deepTools package was used. To this end the genome was binned into 100 bp bins and reads mapping to each bin per sample were counted while reads mapping to the blacklisted regions were excluded (-bs 100 -e -bl ENCFF547MET). The resulting table was analyzed with R, the values (reads per bin) log10 transformed, a Pearson correlation coefficient was calculated and a correlation plot was printed with the smoothScatter function.

Intersection analyses were performed using bedtools[64] intersect (-wa -u); STARR-seq input library peaks were intersected with DNase-seq peaks (GSM1014154[46]) (Supplementary Fig. 9E), Oct4 bound sites (GSE90895[5]) with STARR-seq input peaks (Supplementary Fig. 9F top Venn diagram) and the resulting STARR-seq input covered Oct4 regions with active STARR-seq enhancers (Supplementary Fig. 9F bottom Venn diagram).

**ATAC-seq data processing.** ATAC sequencing data were mapped to mouse genome assembly (mm10) using bowtie2 (–very-sensitive–end-to-end–no-unal–no-mixed -X 2000). Low quality mapped reads were removed using samtools (view –q 30). PCR duplicates were removed using samtools (rmdup) to only keep uniquely mapped reads. The BAM files of time point replicates of each sample were merged with samtools (merge) prior to peak calling with MACS2 (-g mm -f BAMPE). BigWig files were generated using genomeCoverageBed from bedtools[64] and then bedGraphToBigWig. Library statistics are reported in (Supplementary Table 2).

**ChIP-seq data processing.** Sequencing was performed by Wuxi AppTec (Shanghai) Co., Ltd. The libraries were generated using the Truseq Nano DNA kit (Illumina) and sequenced with a Hi-seq 2500 or Hi-seq X (Illumina) with paired-end reads of 125 bp (day 7 Oct4-ChIP) or 150 bp for other samples. Library statistics are reported in (Supplementary Table 2). ChIP-seq reads were aligned to the mouse genome (mm10) using Bowtie2[56] with settings end-to-end and very-sensitive. Samtools[61] was used to discard reads with mapping quality < 10 (samtools view $q = 30$) and only unique reads were kept (samtools rmdup) to account for PCR bias. Output BAM files were sorted using BEDTools[64] sort function and converted into BigWig track files[65] using genomeCoverageBed followed by UCSC utility bedgraphToBigWig. Peaks were called using MACS2 with options–call-summits–to-large -p 0.0001 options[62].

**ChIP-seq correlation analysis.** For corrplot in Fig. 3a, the peak files were downloaded from the publically available GEO database [https://www.ncbi.nlm.nih.gov/geo/]. All the peaks were converted to mm10 using UCSC's LiftOver tool [http://genome.ucsc.edu/]. Hierarchical clustering of the pairwise (R[2]) correlation was performed using the glbase module compare with the options delta = 100, method = "collide", bracket = (0, 1), distance = "euclidean")[59].

**ChIP-seq binary group analysis.** We defined the binary groups for sites bound by Oct4 (Fig. 2a) or Oct4-and-Oct6 (Fig. 6a) at different stages of reprogramming based on binary division in glbase[59]. ChIP-seq reads files in bed format were subjected to glbase function "seqToTrk" with option format = format.bed to get trk files. These generated trk files and ChIP-seq peak summits were used for binary group definitions. The summits of the peaks from all time points (in Oct4 case, days 1, 5, and 7) or summits of peaks from both TFs (Oct4 and Oct6) from days 1 and 5 were merged if they were within 100 bp and a binary definition (1 = bound, 0 = not bound) for each locus was defined thus resulting in 7 (for Oct4 day 1, 5, and 7 ChIPseq) and 15 (for Oct4 and Oct6 ChIPs at days 1 and 5) groups, respectively.

**Motif analysis.** The findMotifsGenome.pl function of HOMER was used for de novo motif discovery in 200 bp regions (-size −100,100) of ChIP-seq summits with motif lengths set to 12, 14, and 16 bp (-len 12,14,16) with additional options -mm10 -p8[66]. For calculating motif fractions, motif scanning was performed using the annotatePeaks.pl function (mm10, -size −100,100 -mbed) with PWMs for SoxOct, MORE, MORE+1 and *Octamer* motifs. For palindromic motifs –norevopp option was used to avoid double counts.

**chromVAR analysis.** For ATAC-seq data, chromVAR[41] was used to compare the chromatin accessibility variation for peaks marked by different TF motifs. Peaks in narrow peak format called with macs2 from Oct4, Oct6, Oct4$^{\text{defSox2}}$ samples at days 1 and 5 (this study), as well as previously published MEFs and ESCs samples[42] were combined and resized to 500 bp yielding a count matrix where rows are peaks,

columns ATAC-seq conditions and fields read counts. For overlapping peaks, the peaks with higher peak scores or overall higher read counts were kept. Motif PWMs from the HOMER database were obtained with the chromVARmotif R package. Motif matches in peak regions were assigned using the motifmatchr package. GC bias corrected accessibility deviations were calculated based on merged reads from ATAC-seq replicates. For better comprehension, the two motifs from HOMER database are renamed as MORE (OCT:OCT(POU, Homeobox, IR1)) and MORE + 1 (OCT:OCT(POU, Homeobox)). Accessibility variability data are presented as a heatmap using the pheatmap R package.

**Gene ontology (GO).** Differentially expressed gene lists were generated from RNA-seq data from day 8 Oct4-SK and Oct6-SK conditions using GFP-SK as reference. Genes were filtered with log2 fold-change more than two as criteria and enriched GO terms were determined with the Metascape[67].

**ChIP-seq and ATAC-seq read intensity analysis.** EAseq v.1.04[68] was used to draw all the ChIP-seq and ATAC-seq read heat maps using EAseq normalization (DNA fragments per kilobase pairs (kbp) per million (M) reads) for each peak file from indicated libraries. For each location, the heatmap signal is displayed by centering at the peak midpoints or on TF motifs. In order to sort the heatmaps by ChIP or ATAC library signal, coverage from the corresponding library was calculated using bedtools multicov. For ATAC-seq and Sox2 ChIP-seq the EAseq normalized read intensities were used to plot boxplots. As the library size differences were bigger for POU ChIP-seq libraries quantile normalization was used for boxplots. POU ChIP-seq coverage was calculated using bedtools multicov, a pseudo-count of 1 was added to the raw coverage values and reads were subsequently quantile normalized using the normalize.quantiles.robust function in R (preprocessCore package). For all the ChIP-seq and ATAC-seq signal boxplots in paper, central rectangle represents the first to third quartiles with median as central line. The whiskers indicate 1.5 times interquartile range.

**Reporting summary.** Further information on research design is available in the Nature Research Reporting Summary linked to this article.

## Data availability
The datasets supporting the conclusions of this article are available in the Gene Expression Omnibus (GEO), under accession number GSE103980. The authors declare that the data supporting the findings of this study are available within the article and its Supplementary Information files, or from the corresponding author upon reasonable request. A Reporting Summary for this Article is available as a Supplementary Information file. The Source Data underlying Figs. 1g, 2i, 2j, 5g and Supplementary Figs. 2B-C, 3B, 3H, 4A, and 4B are provided as a Source Data file.

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

## Acknowledgements

R.J. is supported by the National Key Research and Development Program of China Stem Cell and Translational Research (2017YFA0105103, 2017YFA0105101), the National

Natural Science Foundation of China (Grant No. 31471238, 31771454, 31611130038), Ministry of Science and Technology (MOST) of China grant 2016YFA0100700, a 100 talent award of the Chinese Academy of Sciences, Science and Technology Planning Projects of Guangdong Province, China (2014B030301058, 2016A050503038, 2017B030314056), Research Grants Council of Hong Kong General Research Fund (RGC/GRF) project number 17128918, a Health and Medical Research Fund (06174006) and Germany/Hong Kong Joint Research Scheme sponsored by the Research Grants Council of Hong Kong and the German Academic Exchange Service (Reference No. G-HKU701/18). This work received support from the Max Planck-GIBH Joint Center for Regenerative Biomedicine. J.C. is supported by the Science and Technology Planning Project of Guangdong Province (2014B020225002) and The National Natural Science Foundation of China (31771424). Y.S. is supported by Chinese Government Scholarship (CGS) and University of the Chinese Academy of Sciences (UCAS). V.M. and V.V. thank the CAS-TWAS (Chinese Academy of Sciences-The World Academy of Sciences) President's Fellowship and UCAS (University of Chinese Academy of Science) for financial and infrastructure support. This work was supported by the Deutsche Forschungsgemeinschaft (ME4154/4–1 to L.G.). We thank Leina Lu and Alexander Strunnikov for help with the ChIP-seq experiment setup. We also thank Liu Jing for iCD1 medium supply.

## Author contributions

V.M. and R.J. conceptualized and designed the study and wrote the paper. V.M. performed somatic cell reprogramming, ChIP-seq, RNA-seq, and ATAC-seq experiments. L.V.G. and S.H.M. performed the STARR-seq assay and data analysis. D.Z. and V.M. performed western blots and qPCR. D.Z. and Y.C. performed EMSAs. S.V. performed the pluripotency maintenance assay supervised by H.R.S.; M.W. and M.H. performed ICC experiment and M.A. performed the chromVAR analysis. V.M. and Z.S. performed FACS experiments. Y.S. helped with structural modeling. V.V. helped with whole-well scanning and GEO submission. H.W. helped with initial setup of ChIP-seq assay; J.C. and M.A.E. helped with initial setup of somatic cell reprogramming in iCD1 medium. A.P.H. helped in setting up RNA-seq analysis pipeline. V.M. and R.J. performed bioinformatics analysis. This paper was reviewed and edited by S.P., S.M. and A.P.H. All authors approved the final version of the paper.

## Additional information

**Competing interests:** The authors declare no competing interests.

