## [Peer Review File · Nature Communications]

Reviewers' comments:

Reviewer #1 (Remarks to the Author):

While Oct4 induces pluripotency in somatic cells in combination with other reprogramming factors, highly paralogous factors such as Oct6 cannot induce pluripotency. In this manuscript, the authors tried to address why Oct4 has this unique ability by monitoring dynamic changes in occupancy landscape and accessibility patterns upon induction of Oct4, Oct6, and an Oct4 mutant that is defective in DNA-dependent dimerization with Sox2 during somatic cell reprogramming. Considering extensive previous studies and literature in the field, the message seems incremental in contributing to our understanding of the reprogramming process. In addition, the manuscript is very hard to read and it is not easy to follow the main claims of the manuscript due to the unclear manner in which they visualize their data analysis.

Major points:

1. The first claim of the manuscript, 'Oct6 and Oct4 mutant fail to conclude reprogramming', has been previously reported by the authors, at least partially.
2. It is unclear why Oct6 was used in this manuscript. Why not use Oct1, Oct2, or Oct7?
3. Page 4: 'Under these conditions about 10% of the plated MEF give rise to iPSC colonies within 7 days in a highly synchronous manner'. However, in Figure 2, the genes belonging to the classes 101 and 001 show strikingly different expression patterns (day 8) compared to those of ES cells. Does it mean that the reprogramming system used in this study has some issues? Please explain.
4. It is unclear how significant the motif analysis shown in Figure 2 is. For example, ~45-55% of peaks in classes 100 and 010 are not associated with any motifs. What is the meaning of this?
5. Page 7 bottom: 'The different binding profiles of Oct6 and Oct4defSox2 suggest that their deficiency to reprogram has different underlying causes.' Again, Oct6 is not a reprogramming factor and not highly expressed in ES/iPS cells. Therefore, the different binding patterns of Oct6 vs. Oct4 are not surprising at all. While Oct4 mutant (Oct4defSox2) data is meaningful, I don't see how the Oct6-related data significantly contributes to their claim or presents anything novel.
6. The heatmap data shown in Figures 2, 3, and 4 are very confusing. Especially to perform a combined analysis of TF occupancy data and ATAC-seq data, the authors should re-design their analysis for enhanced reader clarity. For example, there must be a panel showing corresponding ATAC-seq data for the Oct4 occupancy heatmap in Figure 2A. A similar analysis for Oct6 and Oct4 mutant should be performed for unbiased comparisons between factor occupancy and chromatin accessibility. In addition, ATAC-seq data should be obtained from the cells that the authors used for the current study.
7. Page 9: 'We found that sites bound by Oct4 at day 1 also exhibit an increased accessibility at day 1 in Oct6-SK conditions (Figure 4A). We reasoned that two possible scenarios could explain this finding. First, Oct6 could substitute for Oct4 and fulfil its pioneering role. Second, POU TFs lack pioneering function and the opening is achieved indirectly by a mechanism not requiring the presence of POU TFs at these opening sites.' The first scenario doesn't make sense at all. There is no binding of Oct6 but the chromatin is open. How can Oct6 be a pioneer factor in this case? In addition, what is the authors' conclusion about this issue? There is no clear answer in the manuscript.

Minor point: There are multiple typos in the text.

Reviewer #2 (Remarks to the Author):

Building on the known requirement for Oct-Sox DNA co-occupancy in reprogramming, Malik and colleagues aimed to reveal the Oct4-specific reprogramming capacity by comparing Oct4, Oct6 and an Oct4 mutant incapable of dimerizing with Sox2 (Oct4defSox2). Their results suggest that early patterns of chromatin opening are not predictive of reprogramming, while Sox2 and Oct4 co-binding is absolutely required for later pluripotency acquisition.

This work aims to extend concepts from their previously published research which showed that Sox17 can be converted into a reprogramming factor through mutations enabling Oct4-Sox17 heterodimerization (Jauch et al., Stem Cells 2011), and thematically from another study which characterized Oct6 preferential binding to MORE (Oct/Oct) motifs (Jerabek et al., EMBO 2017). This second study already showed that blocking Sox/Oct binding prevented reprogramming, and that mutations in Oct6 could convert it to a reprogramming factor by redistributing binding preference from MORE to Sox/Oct motifs. The impact of the current study could be significantly improved by distinguishing itself from these published works.

Major points

1. The functional differences between “palindromic MORE (more palindromic oct factor recognition element, ATGCATATGCAT-like) element” and “OctSox” need to be described more clearly in the introduction or Figure 1B where they are first presented, rather than waiting until Figure 2D. It is already known that MORE is related to somatic function while OctSox is related to pluripotency. This information makes it easy to interpret the following results. The preferential motif for Oct4 and Oct6 is also important, and needs to be described more clearly.
2. The current manuscript uses Oct6 and Oct4defSox2 to try to make a point that Oct4 occupancy at Sox/Oct motifs is key for reprogramming, although it is unclear if Oct6 and Oct4defSox2 fail for similar reasons. Rather than focus on reprogramming failure, it would be more appropriate to focus on reprogramming success. In their previously published work the author's Oct6 mutant, Oct6(151S), possesses a higher propensity to bind the Sox/Oct motif than MORE, resulting in high reprogramming capacity. In order to understand that profound observation, it is quite important for the authors to verify the Oct6(151S) binding profile with ChIP and compare it to the current Oct4 or native Oct6 data. This will help clarify the importance of interaction with Sox2 and potentially identify target motif/genes which are critical in determining the acquisition of reprogramming.
3. In Figure 1D and 1E, GFP-SK can induce the early phase of reprogramming based on epithelial gene activation, whereas mutant Oct proteins are unable to activate these epithelial genes. In fact, Oct6 appears to interfere with epithelialization. Is this related to the upregulation of neuronal and circulatory genes? Why does Oct4defSox2 interfere? The authors should discuss this.
4. Oct4 heterodimerization with Sox2 seems dispensable for maintenance of pluripotency. Oct6 is not capable of inducing pluripotency, but is it sufficient for maintenance?

Minor points

5. Please describe the relevance of Brn2 in Figure 3A as there seems to be no description in the main text.

6. In the discussion, "Further, the possibility of replacement of Oct4 by an alternative POU (here Oct6) to induce chromatin opening suggests that chromatin status is a poor indicator of successful reprogramming as Oct6-SK cocktails do not generate iPSC (Figure 6D)." This statement (conclusion) may be interpreted as being too dismissive, as we may have simply not identified the key (indicating) sites of open chromatin.

7. Abstract: the phrase "highly homologous paralogous factor" should be simplified to "similar" or "conserved".

Reviewer #3 (Remarks to the Author):

In this manuscript, Malik, et al, analysed where Oct6, an Oct4 mutant defective in the DNA dependent dimerization with Sox2 (Oct4defSox2) can binds when co-expressed with Sox2, Klf4. While neither of them can generate iPSCs and their early binding profile differs from Oct4, they found that initial cellular responses, i.e. MET, down-regulation of somatic gene expression occurred in a similar way to when Oct4, Sox2, Klf4 were overexpressed (Claim 1). They also observed that Oct4defSox2 fails to open pluripotency enhancers, and authors concluded that Oct4 has limited ability to open closed chromatin loci without help of Sox2 (Claim 2). On the other hand, +Oct6 reprogramming opens pluripotency enhancers in a similar way to +Oct4 reprogramming, even though Oct6 does not bind those loci, and it does not result in pluripotency gene up-regulation (Claim 3). Nevertheless, Oct4defSox2 could maintain ESC self-renewal in the absence of wild-type Oct4, highlighting difference between pluripotency induction and maintenance (Claim 4). Oct4 bound loci, but not Oct6 bound loci, in MEFs can act as enhancers in ESCs (Claim 5).

I found this manuscript was interesting, and some claims were more convincing than the others. Claim 1 – In a sense, it is obvious that the initial gene expression change that can be observed in a bulk population is not sufficient to generate iPSCs, since only since only ~1% of MEFs can become iPSCs. However, it is nice to see +Oct6 or +Oct4defSox2 reprogramming also shows similar MET gene, somatic TF expression changes. In the case of MET and early pluripotency genes, it seems true until day3 (Fig 1E). In the case of somatic TFs, it seems true until day 5 (Fig S1C). In my opinion, these data (as well as Fig S1B heatmap) should be highlighted more in the main figure and clarify how many days the 'initial' means. In addition, the authors should include data how many iPSCs colonies (not OG2 Oct4-GFP reporter+ colonies, but colonies with more reliable markers) are generated from how many MEFs in this reprogramming system. Fig 1C showed OG2 Oct4-GFP reporter+ colonies, but Fig S1B tells us that day8 is still far from ESCs (I guess the RNA-seq contains OG2 Oct4-GFP reporter-cells, so it is not very clear how close to ESCs the day 8 Oct4-GFP reporter+ cells are).

Claim 2 - In the abstract, the authors say 'The early binding profile of Oct4defSox2 resembles Oct4'. Seeing the Venn diagram, Fig S3C. Oct4 binding sites are 11,224, Oct4defSox2 binding sites are 43,654, of which only 6,600 (<60% of Oct4 targets) overlap on day1. They don't look very similar to me. The claim that Oct4defSox2 cannot open at CO sites (particularly with SoxOct motif) is based on the CO sites within Oct4 bound sites. Chromatin opening ability of Oct4defSox would need to be addressed by looking at sites bound by Oct4defSox. It would be ideal to analyse the Oct4 specific peaks (11,224 - 6,600 = 4,624), the common 6,600 peaks, Oct4defSox2 specific peaks (43,654 - 6,600 = 37,054) separately. In addition, I would like to know how many of them are co-occupied by Sox2, rather than classifying the peaks by the presence or absence of SoxOct motifs. The authors have the Sox2 ChIP-seq data. Then, how many of them are the closed in MEF, open in ESCs (in the published data), and of which how many of them become open in their day 1 data. This would clarify if Oct4, Oct4defSox2, Oct6 can open the target loci with or without Sox2.

Claim 3 - It is interesting, but I wonder why. If those loci are bound by Oct4defSox2, does it mean Oct4defSox2 inhibits opening? If Oct4defSox2 does not bind there, why they do not open in +Oct4defSox2 reprogramming? Is it possible that Sox2 is responsible for the opening those loci? Are Sox2 binding sites in +Oct4 similar to those in +Oct6 reprogramming, but different in +Oct4defSox2 reprogramming?

Claim 4 – The fact that Oct4defSox2 can replace Oct4 in ESCs is convincing, but the authors should show how many colonies Oct4defSox2 can generate compared to Oct4. Is the rescue frequency is the same as Oct4? If appearance of the rescued ESC colonies is very rare event, it is not very surprising that it does not work at all in iPSC generation.

Claim 5 – It is potentially very interesting, but the data, method have not been described sufficiently. What % of Oct4 bound regions are cloned into the STARR-seq vector? How many Oct4 bound regions showed the enhancer activity in ESCs? Plotting read counts from the library and the reporter (GFP?) +ve cells would be informative. Are the regions which has the enhancer activity not Oct4 targets in ESCs?

More specific points

Page 4, 'We found that both Oct6-SK and Oct4defSox2 SK reprogramming cocktails can correctly progress through the early stages of reprogramming' It is not well understood what is 'correct' progress. I assume the authors mean MET, loss of somatic TF expression. However, Cdh1 expression is lower in Oct6-SK reprogramming even on day 3, in Oct4defSox2 SK reprogramming on day 5 than OSK reprogramming. Flow cytometry analysis of Cdh1 would clarify how MET kinetics, efficiency is similar/dissimilar in those conditions.

Page 6 'Late binding events were significantly associated with elevated gene expression at day 8 compared to MEF (Figures 2J, 2K) – till end of the paragraph' Of course, you can find genes which fits with the binding, too (Fig 2K). But are they majority? I do not think violin plot is suitable to see the correlation between binding and expression over the time course. Perhaps the authors can do K-means clustering then show what % of genes which belong to the each group show similar gene expression pattern to the binding patterns (i.e. 010 -> transient up).

Page 8 'The majority of sites bound by Oct4 at day 1 were inaccessible in MEF (Figure 4A).' It is not clear what the authors consider as 'inaccessible'. A cut off line would be helpful.

Page 8 'These loci show an increased accessibility upon Oct4 binding at day 1 indicating that they undergo a closed-to-open transition. These sites are also accessible in Oct6-SK conditions but accessibility is muted in Oct4 defSox2-SK conditions at day 1.' I can somewhat see what the authors claim, but some quantification and statistics (what % became more open compare to MEFs?) would make it more convincing.

Page 8 'At sites bound by Oct4 at day 5, ChIP-seq signals for Oct6 and Oct4defSox2 are very weak. Nevertheless, chromatin accessibility in the Oct6 condition is comparable to Oct4 but weaker in the Oct4defSox2 condition (Figure 4A, lower panel).' Is it because of Sox2 present/absent at those loci?

Page 9 'The finding that Oct6-SK potently opens pluripotency enhancers establishes that these chromatin state stages by themselves are necessary but insufficient for successful reprogramming.' As far as I see Fig 4F's 3 pluripotency loci, opening by Oct6 is much more inefficient compared to Oct4, which correlates well with gene expression (Fig 4G). It is quite different from average of all O4 ChIP-seq peaks overlapped with CO (Fig 4E), which showed even more open by Oct6 than Oct4. A Fig like Fig 4E, but only with pluripotency gene loci would clarify if the authors' claim is true. In addition, even with Oct4, >90% of cells do not become iPSC. So, it is not very surprising that open chromatin at

early stages does not correlate with iPSC formation.

Minor points

Page 3, Introduction – ‘Replacing Sox2 with other Sox genes such as Sox1, Sox3, Sox4, Sox7 or Sox17 obliterates the activity of reprogramming cocktails’ It is not true. Sox1 could replace Sox2 (ref 21).

Page 4, ‘We found that both Oct6-SK and Oct4defSox2 SK reprogramming cocktails can correctly progress through the early stages of reprogramming’ It is not well understood what is ‘correct’ progress. What authors can say is MET, loss of somatic TF expression

Figure 2K - I see Nodal has binding at a more proximal site on d1. Is Nodal also in 100 group? I assume the numbers at the right top corners are scale of Y axis. Trp63, Zeb1 should have scale 40, 49, respectively.

Having read counts graphs (like Fig 4E) on the top (or bottom) of all heatmaps (like 4D) would be helpful.

Reviewers' comments:

Reviewer #1 (Remarks to the Author):

While Oct4 induces pluripotency in somatic cells in combination with other reprogramming factors, highly paralogous factors such as Oct6 cannot induce pluripotency. In this manuscript, the authors tried to address why Oct4 has this unique ability by monitoring dynamic changes in occupancy landscape and accessibility patterns upon induction of Oct4, Oct6, and an Oct4 mutant that is defective in DNA-dependent dimerization with Sox2 during somatic cell reprogramming. Considering extensive previous studies and literature in the field, the message seems incremental in contributing to our understanding of the reprogramming process. In addition, the manuscript is very hard to read and it is not easy to follow the main claims of the manuscript due to the unclear manner in which they visualize their data analysis.

Major points:

1. The first claim of the manuscript, 'Oct6 and Oct4 mutant fail to conclude reprogramming', has been previously reported by the authors, at least partially.
2. It is unclear why Oct6 was used in this manuscript. Why not use Oct1, Oct2, or Oct7?
3. Page 4: 'Under these conditions about 10% of the plated MEF give rise to iPSC colonies within 7 days in a highly synchronous manner'. However, in Figure 2, the genes belonging to the classes 101 and 001 show strikingly different expression patterns (day 8) compared to those of ES cells. Does it mean that the reprogramming system used in this study has some issues? Please explain.
4. It is unclear how significant the motif analysis shown in Figure 2 is. For example, ~45-55% of peaks in classes 100 and 010 are not associated with any motifs. What is the meaning of this?
5. Page 7 bottom: 'The different binding profiles of Oct6 and Oct4defSox2 suggest that their deficiency to reprogram has different underlying causes.' Again, Oct6 is not a reprogramming factor and not highly expressed in ES/iPS cells. Therefore, the different binding patterns of Oct6 vs. Oct4 are not surprising at all. While Oct4 mutant (Oct4defSox2) data is meaningful, I don't see how the Oct6-related data significantly contributes to their claim or presents anything novel.
6. The heatmap data shown in Figures 2, 3, and 4 are very confusing. Especially to perform a combined analysis of TF occupancy data and ATAC-seq data, the authors should re-design their analysis for enhanced reader clarity. For example, there must be a panel showing corresponding ATAC-seq data for the Oct4 occupancy heatmap in Figure 2A. A similar analysis for Oct6 and Oct4 mutant should be performed for unbiased comparisons between factor occupancy and chromatin accessibility. In addition, ATAC-seq data should be obtained from the cells that the authors used for the current study.

7. Page 9: 'We found that sites bound by Oct4 at day 1 also exhibit an increased accessibility at day 1 in Oct6-SK conditions (Figure 4A). We reasoned that two possible scenarios could explain this finding. First, Oct6 could substitute for Oct4 and fulfil its pioneering role. Second, POU TFs lack pioneering function and the opening is achieved indirectly by a mechanism not requiring the presence of POU TFs at these opening sites.' The first scenario doesn't make sense at all. There is no binding of Oct6 but the chromatin is open. How can Oct6 be a pioneer factor in this case? In addition, what is the authors' conclusion about this issue? There is no clear answer in the manuscript.

Minor point: There are multiple typos in the text.

Reviewer #2 (Remarks to the Author):

Building on the known requirement for Oct-Sox DNA co-occupancy in reprogramming, Malik and colleagues aimed to reveal the Oct4-specific reprogramming capacity by comparing Oct4, Oct6 and an Oct4 mutant incapable of dimerizing with Sox2 (Oct4defSox2). Their results suggest that early patterns of chromatin opening are not predictive of reprogramming, while Sox2 and Oct4 co-binding is absolutely required for later pluripotency acquisition.

This work aims to extend concepts from their previously published research which showed that Sox17 can be converted into a reprogramming factor through mutations enabling Oct4-Sox17 heterodimerization (Jauch et al., Stem Cells 2011), and thematically from another study which characterized Oct6 preferential binding to MORE (Oct/Oct) motifs (Jerabek et al., EMBO 2017). This second study already showed that blocking Sox/Oct binding prevented reprogramming, and that mutations in Oct6 could convert it to a reprogramming factor by redistributing binding preference from MORE to Sox/Oct motifs. The impact of the current study could be significantly improved by distinguishing itself from these published works.

Major points

1. The functional differences between "palindromic MORE (more palindromic oct factor recognition element, ATGCATATGCAT-like) element" and "OctSox" need to be described more clearly in the introduction or Figure 1B where they are first presented, rather than waiting until Figure 2D. It is already known that MORE is related to somatic function while OctSox is related to pluripotency. This information makes it easy to interpret the following results. The preferential motif for Oct4 and Oct6 is also important, and needs to be described more clearly.
2. The current manuscript uses Oct6 and Oct4defSox2 to try to make a point that Oct4 occupancy at Sox/Oct motifs is key for reprogramming, although is unclear if Oct6 and Oct4defSox2 fail for similar reasons. Rather than focus on reprogramming failure, it would be more appropriate to focus on reprogramming success. In their previously published work the author's Oct6 mutant,

Oct6(151S), possesses a higher propensity to bind the Sox/Oct motif than MORE, resulting in high reprogramming capacity. In order to understand that profound observation, it is quite important for the authors to verify the Oct6(151S) binding profile with ChIP and compare it to the current Oct4 or native Oct6 data. This will help clarify the importance of interaction with Sox2 and potentially identify target motif/genes which are critical in determining the acquisition of reprogramming.

3. In Figure 1D and 1E, GFP-SK can induce the early phase of reprogramming based on epithelial gene activation, whereas mutant Oct proteins are unable to activate these epithelial genes. In fact, Oct6 appears to interfere with epithelialization. Is this related to the upregulation of neuronal and circulatory genes? Why does Oct4defSox2 interfere? The authors should discuss this.

4. Oct4 heterodimerization with Sox2 seems dispensable for maintenance of pluripotency. Oct6 is not capable of inducing pluripotency, but is it sufficient for maintenance?

Minor points

5. Please describe the relevance of Brn2 in Figure 3A as there seems to be no description in the main text.

6. In the discussion, "Further, the possibility of replacement of Oct4 by an alternative POU (here Oct6) to induce chromatin opening suggests that chromatin status is a poor indicator of successful reprogramming as Oct6-SK cocktails do not generate iPSC (Figure 6D)." This statement (conclusion) may be interpreted as being too dismissive, as we may have simply not identified the key (indicating) sites of open chromatin.

7. Abstract: the phrase "highly homologous paralogous factor" should be simplified to "similar" or "conserved".

Reviewer #3 (Remarks to the Author):

In this manuscript, Malik, et al, analysed where Oct6, an Oct4 mutant defective in the DNA dependent dimerization with Sox2 (Oct4defSox2) can binds when co-expressed with Sox2, Klf4. While neither of them can generate iPSCs and their early binding profile differs from Oct4, they found that initial cellular responses, i.e. MET, down-regulation of somatic gene expression occurred in a similar way to when Oct4, Sox2, Klf4 were overexpressed (Claim 1). They also observed that Oct4defSox2 fails to open pluripotency enhancers, and authors concluded that Oct4 has limited ability to open closed chromatin loci without help of Sox2 (Claim 2). On the other hand, +Oct6 reprogramming opens pluripotency enhancers in a similar way to +Oct4 reprogramming, even though Oct6 does not bind those loci, and it does not result in pluripotency gene up-regulation (Claim 3). Nevertheless, Oct4defSox2 could maintain ESC self-renewal in the absence of wild-type Oct4, highlighting difference between pluripotency induction and maintenance (Claim 4). Oct4 bound loci, but not Oct6 bound loci, in MEFs can act as enhancers in ESCs (Claim 5).

I found this manuscript was interesting, and some claims were more convincing than the others.

Claim 1 – In a sense, it is obvious that the initial gene expression change that can be observed in a bulk population is not sufficient to generate iPSCs, since only since only ~1% of MEFs can become iPSCs. However, it is nice to see +Oct6 or +Oct4defSox2 reprogramming also shows similar MET gene, somatic TF expression changes. In the case of MET and early pluripotency genes, it seems true until day3 (Fig 1E). In the case of somatic TFs, it seems true until day 5 (Fig S1C). In my opinion, these data (as well as Fig S1B heatmap) should be highlighted more in the main figure and clarify how many days the ‘initial’ means. In addition, the authors should include data how many iPSCs colonies (not OG2 Oct4-GFP reporter+ colonies, but colonies with more reliable markers) are generated from how many MEFs in this reprogramming system. Fig 1C showed OG2 Oct4-GFP reporter+ colonies, but Fig S1B tells us that day8 is still far from ESCs (I guess the RNA-seq contains OG2 Oct4-GFP reporter-cells, so it is not very clear how close to ESCs the day 8 Oct4-GFP reporter+ cells are).

Claim 2 - In the abstract, the authors say ‘The early binding profile of Oct4defSox2 resembles Oct4’. Seeing the Venn diagram, Fig S3C. Oct4 binding sites are 11,224, Oct4defSox2 binding sites are 43,654, of which only 6,600 (<60% of Oct4 targets) overlap on day1. They don’t look very similar to me. The claim that Oct4defSox2 cannot open at CO sites (particularly with SoxOct motif) is based on the CO sites within Oct4 bound sites. Chromatin opening ability of Oct4defSox would need to be addressed by looking at sites bound by Oct4defSox. It would be ideal to analyse the Oct4 specific peaks (11,224 - 6,600 = 4,624), the common 6,600 peaks, Oct4defSox2 specific peaks (43,654 - 6,600 = 37,054) separately. In addition, I would like to know how many of them are co-occupied by Sox2, rather than classifying the peaks by the presence or absence of SoxOct motifs. The authors have the Sox2 ChIP-seq data. Then, how many of them are the closed in MEF, open in ESCs (in the published data), and of which how many of them become open in their day 1 data. This would clarify if Oct4, Oct4defSox2, Oct6 can open the target loci with or without Sox2.

Claim 3 - It is interesting, but I wonder why. If those loci are bound by Oct4defSox2, does it mean Oct4defSox2 inhibits opening? If Oct4defSox2 does not bind there, why they do not open in +Oct4defSox2 reprogramming? Is it possible that Sox2 is responsible for the opening those loci? Are Sox2 binding sites in +Oct4 similar to those in +Oct6 reprogramming, but different in +Oct4defSox2 reprogramming?

Claim 4 – The fact that Oct4defSox2 can replace Oct4 in ESCs is convincing, but the authors should show how many colonises Oct4defSox2 can generate compared to Oct4. Is the rescue frequency is the same as Oct4? If appearance of the rescued ESC colonies is very rare event, it is not very surprising that it does not work at all in iPSC generation.

Claim 5 – It is potentially very interesting, but the data, method have not been described sufficiently. What % of Oct4 bound regions are cloned into the STARR-seq vector? How many Oct4 bound regions showed the enhancer activity in ESCs? Plotting read counts from the library and the

reporter (GFP?) +ve cells would be informative. Are the regions which has the enhancer activity not Oct4 targets in ESCs?

More specific points

Page 4, 'We found that both Oct6-SK and Oct4defSox2 SK reprogramming cocktails can correctly progress through the early stages of reprogramming' It is not well understood what is 'correct' progress. I assume the authors mean MET, loss of somatic TF expression. However, Cdh1 expression is lower in Oct6-SK reprogramming even on day 3, in Oct4defSox2 SK reprogramming on day 5 than OSK reprogramming. Flow cytometry analysis of Cdh1 would clarify how MET kinetics, efficiency is similar/dissimilar in those conditions.

Page 6 'Late binding events were significantly associated with elevated gene expression at day 8 compared to MEF (Figures 2J, 2K) – till end of the paragraph' Of course, you can find genes which fits with the binding, too (Fig 2K). But are they majority? I do not think violin plot is suitable to see the correlation between binding and expression over the time course. Perhaps the authors can do K-means clustering then show what % of genes which belong to the each group show similar gene expression pattern to the binding patterns (i.e. 010 -> transient up).

Page 8 'The majority of sites bound by Oct4 at day 1 were inaccessible in MEF (Figure 4A).' It is not clear what the authors consider as 'inaccessible'. A cut off line would be helpful.

Page 8 'These loci show an increased accessibility upon Oct4 binding at day 1 indicating that they undergo a closed-to-open transition. These sites are also accessible in Oct6-SK conditions but accessibility is muted in Oct4 defSox2-SK conditions at day 1.' I can somewhat see what the authors claim, but some quantification and statistics (what % became more open compare to MEFs?) would make it more convincing.

Page 8 'At sites bound by Oct4 at day 5, ChIP-seq signals for Oct6 and Oct4defSox2 are very weak. Nevertheless, chromatin accessibility in the Oct6 condition is comparable to Oct4 but weaker in the Oct4defSox2 condition (Figure 4A, lower panel).' Is it because of Sox2 present/absent at those loci?

Page 9 'The finding that Oct6-SK potentially opens pluripotency enhancers establishes that these chromatin state stages by themselves are necessary but insufficient for successful reprogramming.' As far as I see Fig 4F's 3 pluripotency loci, opening by Oct6 is much more inefficient compared to Oct4, which correlates well with gene expression (Fig 4G). It is quite different from average of all O4 ChIP-seq peaks overlapped with CO (Fig 4E), which showed even more open by Oct6 than Oct4. A Fig like Fig 4E, but only with pluripotency gene loci would clarify if the authors' claim is true. In addition, even with Oct4, >90% of cells do not become iPSC. So, it is not very surprising that open chromatin at early stages does not correlate with iPSC formation.

Minor points

Page 3, Introduction – 'Replacing Sox2 with other Sox genes such as Sox1, Sox3, Sox4, Sox7 or Sox17 obliterates the activity of reprogramming cocktails' It is not true. Sox1 could replace Sox2 (ref 21).

Page 4, 'We found that both Oct6-SK and Oct4defSox2 SK reprogramming cocktails can correctly progress through the early stages of reprogramming' It is not well understood what is 'correct' progress. What authors can say is MET, loss of somatic TF expression

Figure 2K - I see Nodal has binding at a more proximal site on d1. Is Nodal also in 100 group? I assume the numbers at the right top corners are scale of Y axis. Trp63, Zeb1 should have scale 40, 49, respectively.

Having read counts graphs (like Fig 4E) on the top (or bottom) of all heatmaps (like 4D) would be helpful.

Reviewers' comments:

Reviewer #1 (Remarks to the Author):

While Oct4 induces pluripotency in somatic cells in combination with other reprogramming factors, highly paralogous factors such as Oct6 cannot induce pluripotency. In this manuscript, the authors tried to address why Oct4 has this unique ability by monitoring dynamic changes in occupancy landscape and accessibility patterns upon induction of Oct4, Oct6, and an Oct4 mutant that is defective in DNA-dependent dimerization with Sox2 during somatic cell reprogramming. Considering extensive previous studies and literature in the field, the message seems incremental in contributing to our understanding of the reprogramming process. In addition, the manuscript is very hard to read and it is not easy to follow the main claims of the manuscript due to the unclear manner in which they visualize their data analysis.

Major points:

1. The first claim of the manuscript, 'Oct6 and Oct4 mutant fail to conclude reprogramming', has been previously reported by the authors, at least partially.

>> We agree that we have previously reported both Oct6 and Oct4^{defSox2} cannot reprogram MEFs using 4 factor cocktails in Serum/LIF conditions. Yet, genome-wide analysis (RNA-seq, ChIP-seq and ATAC-seq data) to elucidate the reason for their failure has not been performed before. Previously, whilst we knew they cannot complete reprogramming, as measured by the expression of the Oct4-GFP reporter, we did not know the mechanism behind this failure. The Oct4-GFP reporter is late in the reprogramming process and a lot of cellular changes happen before its activation. Our study reveals the mechanism that at early stages of reprogramming transcriptional responses and chromatin opening do not require Oct4 while Oct4 is crucial for activation of pluripotency network at later stage. We have now revised the introduction to better explain the rationale for our study and concisely summarize the key findings that go beyond work previously published by ourselves and others in highlights, graphical summary and discussion. To avoid confusion we now re-phrase the section heading to "**Oct4 is dispensable for initiating somatic cell reprogramming**".

2. It is unclear why Oct6 was used in this manuscript. Why not use Oct1, Oct2, or Oct7?

>> We thank reviewer for this comment and subjected the introduction to a major revision to clarify our choice. Briefly, Oct6 is the most interesting POU family factor to compare with Oct4 at the molecular level to study the basis for the unique roles of Oct4. We have previously elucidated the crystal structure of Oct6 bound to *MORE* DNA and, in collaboration with co-author Hans Schöler, the crystal structure of Oct4 on *PORE* DNA^{1,2}. This sparked our interest in the molecular analysis. Functionally, Oct6 has previously been dissected using a pluripotency maintenance assay with detailed analysis of critical amino acids and domain swaps³. In addition to the structural analysis, we have performed quantitative biochemical assays and iPSCs induction assays and could convert

Oct6 into an iPSCs inducer⁴. Further, members of the POU III class have been reported to be unable to act as 'pioneer factors' to open chromatin during neural reprogramming⁵. As we wanted to study factors that are highly homologous yet have contrasting activities to direct cell fate conversion, we feel Oct6 is the obvious choice for our experiments.

3. Page 4: 'Under these conditions about 10% of the plated MEF give rise to iPSC colonies within 7 days in a highly synchronous manner'. However, in Figure 2, the genes belonging to the classes 101 and 001 show strikingly different expression patterns (day 8) compared to those of ES cells. Does it mean that the reprogramming system used in this study has some issues? Please explain.

>> We agree that the reprogramming system should be described more clearly and now performed Nanog immunostaining for iPSCs generated with the iCD1 medium at day 8. We also describe in the results section that iPSCs obtained using the iCD1 system can give rise to chimeric mice and germline transmission as early as day 7 without further culturing (with 33% success rate as recently published in *Molecular Cell* ⁶). We furthermore re-processed published expression data for ESCs and passaged iPSCs generated in chemically defined conditions as well as cultured in serum conditions from three laboratories which show that our day 8 cells correlate very well with such datasets (new Figure S1D).

We thank the reviewer for pointing out that our choice of genes and the previously used ESCs references was not ideal. We had previously used ESCs data cultured in serum conditions from Bing Ren's lab⁷ but in the current version of manuscript we replaced them with the iPSCs and ESCs data⁸ that were obtained from MEFs of OG2 mice and cultured under chemically defined conditions. We updated the expression plots in Figure 2K by adding iPSCs generated from OG2-MEFs. Aoah is transiently upregulated during reprogramming and becomes re-repressed during the maturation of iPSCs (Reviewer Figure 1A). We also replaced Nodal with Zyg11a to avoid confusion. Please note the reviewer figure below showing that Nodal expression fluctuates across published ESCs and iPSCs datasets (Reviewer Figure 1B).

Reviewer figure 1: Expression profiles of *AoaH* and *Nodal* genes. (A) Expression of *AoaH* and *Nodal* in somatic cell reprogramming in serum/LIF conditions obtained from the publically available RNA-seq datasets⁹. (B) Expression of *AoaH* and *Nodal* in top 50 cells types or tissues out of total 272 samples are shown as bar plots¹⁰.

4. It is unclear how significant the motif analysis shown in Figure 2 is. For example, ~45-55% of peaks in classes 100 and 010 are not associated with any motifs. What is the meaning of this?

>> We wish to politely point out that it is commonly observed that a subset of binding sites lacks detectable motif matches given a certain threshold used by the motif discovery algorithms. Indeed, it is a common and unexplained observation that many ChIP-seq peaks do not contain the motif for the appropriate transcription factor. The values we report here are consistent with what has been seen in studies by Jose Polo, Jacob Hanna, Kathrin Plath and Shaorong Gao (Gao- Fig 2G, Hanna Fig. 2E, early binding trajectory in Plath 2D, Polo Fig3B¹¹⁻¹⁴). For example Knaupp et al reported that 50% of their Oct4 ChIP-seq peaks during reprogramming contained matches to any of the POU motif (Fig. 3B,¹³).

Trajectories 100 and 010 are transiently bound locations and the fraction of sites that lack matches to motifs is lower than, for example, constitutively bound sites (111). In the absence of cognate

binding sites, binding in aggregate is short lived and could capture part of a TFs target search and/or weak binding. We amended the results section to highlight this point.

5. Page 7 bottom: ‘The different binding profiles of Oct6 and Oct4defSox2 suggest that their deficiency to reprogram has different underlying causes.’ Again, Oct6 is not a reprogramming factor and not highly expressed in ES/iPS cells. Therefore, the different binding patterns of Oct6 vs. Oct4 are not surprising at all. While Oct4 mutant (Oct4defSox2) data is meaningful, I don’t see how the Oct6-related data significantly contributes to their claim or presents anything novel.

>> We politely disagree. We have previously solved the crystal structure of Oct6 and performed detailed biochemical and structural comparison of Oct4 and Oct6. Both proteins have very similar DNA binding domains. Yet, it is subtle features (perhaps just a few amino acids) within these DNA binding domains that set them molecularly apart. We believe that understanding the sequence-structure-function relationship of TFs is critical to understand the basis of gene regulation and reprogramming and for the re-design of reprogramming factors. Further, although Oct6 is expressed at low levels in ESCs/iPSCs, but Oct6 and Sox2 are coexpressed as early as E6.5 through E7.5 suggesting they may be key factors for driving lineage divergence¹⁵. Oct6 might not be a pluripotency inducing factor but it has been shown that Oct6 overexpression in ESCs is both necessary and sufficient to generate neural progenitor cells¹⁶. Lastly, the fellow POUIII factor Brn2, was reported to be a non-pioneering TF unable to bind closed chromatin⁵ whereas Oct4 is regarded as a pioneer factor^{17,18}. Similarly, Ascl1⁵ and Myc^{17,18} are both bHLH (basic helix loop helix) proteins which were classified to be pioneer and non-pioneer factors respectively. Thus, there may be properties that allow the grouping of closely related TFs into pioneer vs non-pioneer and to study this, we believe, is a very interesting biological question. It is thought that the potency to bring about chromatin accessibility changes is a key property of reprogramming TFs. We revised the introduction to clarify our reasoning and perform additional experiments to test how Oct6 affects the mesenchymal-to-epithelial transition (MET) that are presented in the revised manuscript.

We summarize novel insights obtained using the Oct6 data:

1. Oct6 containing 3F cocktails are not compromised in chromatin opening and showed similar chromatin opening patterns at otherwise Oct4-bound sites at the early stage of reprogramming reminiscent to what is observed for Oct4-SK.
2. Oct6-SK increases additional specific accessibility changes at sites marked by *MORE* elements
3. Oct6 facilitates the downregulation of somatic genes at early reprogramming stages
4. Oct6-SK counteracts epithelialization illustrated by the mitigated activation of E-cadherin compared to Oct4-SK or just SK cocktails
5. Oct6 binds fundamentally different sites than Oct4 in contrast to near identical binding patterns of other functionally distinct paralogous factors such as MyoD and Ascl1 during neural reprogramming (Marius Wernig, unpublished data presented at the Guangzhou Stem Cell conference 2018)

6. The heatmap data shown in Figures 2, 3, and 4 are very confusing. Especially to perform a combined analysis of TF occupancy data and ATAC-seq data, the authors should re-design their analysis for enhanced reader clarity. For example, there must be a panel showing corresponding ATAC-seq data for the Oct4 occupancy heatmap in Figure 2A. A similar analysis for Oct6 and Oct4 mutant should be performed for unbiased comparisons between factor occupancy and chromatin accessibility. In addition, ATAC-seq data should be obtained from the cells that the authors used for the current study.

1. We would like to apologise for not making this clear in the previous manuscript, but we obtained our own ATAC-seq data for the three factors (Oct6-SK, Oct4-SK and Oct4^{defSox2}-SK) at days 1 and 5 using the chemically defined iCD1 reprogramming system. We added previously published 1 factor and 2 factor ATAC-seq data that were also generated using the same iCD1 system as our study (taken from a study led by Andrew Hutchins and Jiekai Chen; co-authors of the present study⁸). Therefore, the analysis was performed using cells subject to identical reprogramming conditions.
2. We re-designed parts of the analysis and have now performed chromVAR for an unbiased analysis of the ATAC-seq data and associated cis-regulatory elements. The analysis showed a clear distinction in motif sequences present in the sites that do not show an increase in accessibility in Oct4^{defSox2} -SK (sites marked by SoxOct motifs) or that distinctly show accessibility increase in Oct6-SK conditions. In Oct4^{defSox2} -SK many sites marked by single Sox motifs show a specific accessibility increase whilst in Oct6-SK many sites marked by single POU and MORE elements show opening. This analysis and other changes are presented in the revised Figure 4 following the combined suggestion of all reviewers
3. We wish to apologize that we are somewhat uncertain as to how to perform the additional analysis that reviewer suggests to integrate ChIP-seq and ATAC-seq data. We have prepared the below Reviewer figure 2, where we have added ATAC-seq data to the heatmap shown in Figure 2A and we have prepared analogous figures for Oct6 and Oct4^{defSox2}. We however feel that the data are not very informative if represented in this way and suggest to integrate ATAC-seq and ChIP-seq data as done in the revised Figure 4 (i.e. by defining sites that are closed or open in MEFs and sites that undergo closed-to-open transitions and monitor binding and accessibility changes across the analysed conditions).

Reviewer figure 2. Heatmap of ChIP-seq and corresponding ATAC-seq data for Oct4 (green) bound binary occupancy trajectories defined in manuscript figure 2A. Similar analysis is also shown for Oct6 (pink) and Oct4^{defSox2} (orange) conditions at days 1 and 5 of reprogramming.

7. Page 9: ‘We found that sites bound by Oct4 at day 1 also exhibit an increased accessibility at day 1 in Oct6-SK conditions (Figure 4A). We reasoned that two possible scenarios could explain this finding. First, Oct6 could substitute for Oct4 and fulfil its pioneering role. Second, POU TFs lack pioneering function and the opening is achieved indirectly by a mechanism not requiring the presence of POU TFs at these opening sites.’ The first scenario doesn’t make sense at all. There is no binding of Oct6 but the chromatin is open. How can Oct6 be a pioneer factor in this case? In addition, what is the authors’ conclusion about this issue? There is no clear answer in the manuscript.

>> We agree with the reviewer and modify our claims in the text. The revised analysis presented in Figure 4, which now suggests that Sox2 has the main role in chromatin opening and the role of POU factor is largely accessory in the opening of sites that get opened in Oct4-SK conditions (many of which contain composite *SoxOct* elements). To further address the question whether Oct6 can bring about chromatin opening factor, we have performed chromVAR analysis using the union of all ATAC-seq peaks and found that Oct6-SK opens a lot of ectopic sites not opened in the Oct4-SK (and Oct4^{defSox2}-SK) conditions. Often these ectopic sites are marked by *MORE* elements or non-composite POU motifs. Given the strong preference of Oct6 binding to *MORE* motifs, this is very likely that opening of these sites could be facilitated by Oct6. This observation is potentially interesting as it is in contrast to what has been reported for Brn2 during neural reprogramming which is unable to target closed chromatin. We describe these data in the revised manuscript and note our conclusion in the discussion.

Minor point: There are multiple typos in the text.

>> We carefully revised the manuscript to eliminate typos.

Reviewer #2 (Remarks to the Author):

Building on the known requirement for Oct-Sox DNA co-occupancy in reprogramming, Malik and colleagues aimed to reveal the Oct4-specific reprogramming capacity by comparing Oct4, Oct6 and an Oct4 mutant incapable of dimerizing with Sox2 (Oct4defSox2). Their results suggest that early patterns of chromatin opening are not predictive of reprogramming, while Sox2 and Oct4 co-binding is absolutely required for later pluripotency acquisition.

This work aims to extend concepts from their previously published research which showed that Sox17 can be converted into a reprogramming factor through mutations enabling Oct4-Sox17 heterodimerization (Jauch et al., Stem Cells 2011), and thematically from another study which characterized Oct6 preferential binding to MORE (Oct/Oct) motifs (Jerabek et al., EMBO 2017). This second study already showed that blocking Sox/Oct binding prevented reprogramming, and that mutations in Oct6 could convert it to a reprogramming factor by redistributing binding preference from MORE to Sox/Oct motifs. The impact of the current study could be significantly improved by distinguishing itself from these published works.

>> We thank the reviewer for this suggestion. It is correct that our two previous publications inspired the present study. Yet, the previous studies did not contain any genome-wide analyses which we are providing here (ChIP-seq, ATAC-seq, RNA-seq). We thoroughly revised the introduction to better distinguish the current study from the previous ones.

Major points

1. The functional differences between “palindromic MORE (more palindromic oct factor recognition element, ATGCATATGCAT-like) element” and “OctSox” need to be described more clearly in the introduction or Figure 1B where they are first presented, rather than waiting until Figure 2D. It is already known that MORE is related to somatic function while OctSox is related to pluripotency. This information makes it easy to interpret the following results. The preferential motif for Oct4 and Oct6 is also important, and needs to be described more clearly.

>> We thank the reviewer for pointing it out. We now introduce the various POU binding elements in the introduction section in depth and cite studies linking the *MORE* motif to somatic functions.

2. The current manuscript uses Oct6 and Oct4defSox2 to try to make a point that Oct4 occupancy at Sox/Oct motifs is key for reprogramming, although is unclear if Oct6 and Oct4defSox2 fail for similar reasons. Rather than focus on reprogramming failure, it would be more appropriate to focus on reprogramming success. In their previously published work the author’s Oct6 mutant, Oct6(151S), possesses a higher propensity to bind the Sox/Oct motif than MORE, resulting in high reprogramming capacity. In order to understand that profound observation, it is quite important for the authors to verify the Oct6(151S) binding

profile with ChIP and compare it to the current Oct4 or native Oct6 data. This will help clarify the importance of interaction with Sox2 and potentially identify target motif/genes which are critical in determining the acquisition of reprogramming.

>> We performed ChIP-seq for Oct6-M151S as well as Oct4-S151M at day 1 of reprogramming. As expected from previously published EMSAs, Oct6-M151S loses binding to *MORE* sites whilst Oct4-S151M gains preferences for *MORE* sites in a chromatin context. These results are included in the revised manuscript in supplementary Figure S2C-F. We wish to politely point out that Oct5M151S alone is not sufficient to convert Oct6 into pluripotency reprogramming factors⁴. Rather, additional modifications to the helix 1 of the POU₅ domain and further modifications to the linker are beneficial in order to enable the re-engineered Oct6 to promote pluripotency reprogramming in 4-Factors serum/LIF conditions. Still, the efficiency of the re-engineered Oct6 lags behind wild-type Oct4⁴. This in contrast to the re-engineered Sox17 (Sox17EK) that substantially outperforms wild-type Sox2¹⁹. We now clarify this in the revised introduction. We like to point out that we could also re-engineer the fellow Brn4 with an analogous set of mutations, which converts it into an iPSCs inducer under 4F serum conditions. However, in the chemically defined 3F conditions used here the Oct6 mutant cannot replace Oct4 and was thus not included in the study. We agree that it will be of great interest to study the Oct6 and Brn4 mutants using 4F serum conditions in a future study.

3. In Figure 1D and 1E, GFP-SK can induce the early phase of reprogramming based on epithelial gene activation, whereas mutant Oct proteins are unable to activate these epithelial genes. In fact, Oct6 appears to interfere with epithelialization. Is this related to the upregulation of neuronal and circulatory genes? Why does Oct4^{defSox2} interfere? The authors should discuss this.

>> We thank the reviewer for the critical comment. In response to this and a comment from reviewer 3, we performed FACS for Cdh1 (encoding E-cadherin) at days 3, 5 and 8 of reprogramming in various cocktails (SK, GFP-SK, O4-SK, O6-SK and O4^{defSox2}-SK). Indeed, as the reviewer astutely predicted, Oct6 is unable to activate epithelialization program, as the fraction of cells expressing E-cadherin was lower in Oct6-SK than in GFP-SK and SK conditions. Additionally we did see enrichment of somatic system development gene-ontology annotations like “neuronal and circulatory system” for differentially expressed genes for Oct6-SK cells. These results suggest that the upregulation of neuronal genes is indeed ultimately interfering with activation of the MET. We thank the reviewer for their insightful comment which has allowed us to clarify the role of Oct6 in epithelialisation. We discuss these data in detail in the results, discussion and revised Figures (1G, S2B-C).

Both wild-type Oct4 and Oct4^{defSox2} containing cocktails appear to support activation of E-cadherin from the beginning of reprogramming as the levels of E-cadherin in these two conditions are nearly 1.5-2 fold higher than in the conditions lacking them. However, E-cadherin levels start to drop in Oct4^{defSox2} condition by day 8 which could be correlated with the degrading Oct4^{defSox2} proteins from day 5 onwards of reprogramming as evident from western blots (revised manuscript, Figure S4B).

We also expanded our discussion as to the upregulation of somatic genes by Oct6 and the interference of Oct4^{defSox2} with the chromatin opening by Sox2 in the revised Figure 4.

4. Oct4 heterodimerization with Sox2 seems dispensable for maintenance of pluripotency. Oct6 is not capable of inducing pluripotency, but is it sufficient for maintenance?

>> We are grateful for these suggestion and would like to refer to two studies, including one from our group in collaboration with co-author Hans Schöler's group, that had previously shown that Oct6 is not capable of maintaining ESCs^{3,4}. We now clearly state these published findings in the revised introduction and results section of the revised manuscript. We have also added a whole well image for the Oct4^{defSox2} rescue experiment and a control where no POU factor was transduced (Figure S7D).

Minor points

5. Please describe the relevance of Brn2 in Figure 3A as there seems to be no description in the main text.

>> We thank reviewer for bringing this up. We have described our reasoning regarding relevance of Brn2 and other POU III family factors in the introduction section as well as in main text for Figure 3A of the revised manuscript. We wanted to explore whether Oct6 also binds sites bound by Brn2 in neural lineages. Yet, under iPSCs conditions Oct6 does not globally resemble the binding profile of Brn2 in neural lineages.

6. In the discussion, "Further, the possibility of replacement of Oct4 by an alternative POU (here Oct6) to induce chromatin opening suggests that chromatin status is a poor indicator of successful reprogramming as Oct6-SK cocktails do not generate iPSC (Figure 6D)." This statement (conclusion) may be interpreted as being too dismissive, as we may have simply not identified the key (indicating) sites of open chromatin.

>> We agree and revised the discussion section.

7. Abstract: the phrase "highly homologous paralogous factor" should be simplified to "similar" or "conserved".

>> We have corrected these.

Reviewer #3 (Remarks to the Author):

In this manuscript, Malik, et al, analysed where Oct6, an Oct4 mutant defective in the DNA dependent dimerization with Sox2 (Oct4defSox2) can binds when co-expressed with Sox2, Klf4. While neither of them can generate iPSCs and their early binding profile differs from Oct4, they found that initial cellular responses, i.e. MET, down-regulation of somatic gene expression occurred in a similar way to when Oct4, Sox2, Klf4 were overexpressed (Claim 1). They also observed that Oct4defSox2 fails to open pluripotency enhancers, and authors concluded that Oct4 has limited ability to open closed chromatin loci without help of Sox2 (Claim 2). On the other hand, +Oct6 reprogramming opens pluripotency enhancers in a similar way to +Oct4 reprogramming, even though Oct6 does not bind those loci, and it does not result in pluripotency gene up-regulation (Claim 3). Nevertheless, Oct4defSox2 could maintain ESC self-renewal in the absence of wild-type Oct4, highlighting difference between pluripotency induction and maintenance (Claim 4). Oct4 bound loci, but not Oct6 bound loci, in MEFs can act as enhancers in ESCs (Claim 5).

I found this manuscript was interesting, and some claims were more convincing than the others.

Claim 1 – In a sense, it is obvious that the initial gene expression change that can be observed in a bulk population is not sufficient to generate iPSCs, since only since only ~1% of MEFs can become iPSCs. However, it is nice to see +Oct6 or +Oct4defSox2 reprogramming also shows similar MET gene, somatic TF expression changes. In the case of MET and early pluripotency genes, it seems true until day3 (Fig 1E). In the case of somatic TFs, it seems true until day 5 (Fig S1C). In my opinion, these data (as well as Fig S1B heatmap) should be highlighted more in the main figure and clarify how many days the ‘initial’ means. In addition, the authors should include data how many iPSCs colonies (not OG2 Oct4-GFP reporter+ colonies, but colonies with more reliable markers) are generated from how many MEFs in this reprogramming system. Fig 1C showed OG2 Oct4-GFP reporter+ colonies, but Fig S1B tells us that day8 is still far from ESCs (I guess the RNA-seq contains OG2 Oct4-GFP reporter- cells, so it is not very clear how close to ESCs the day 8 Oct4-GFP reporter+ cells are).

>> We thank the reviewer for these suggestions and we have substantially revised Figure 1 and its associated two supplementary figures.

- 1) We highlighted the averaged expression for selected marker genes for MET, somatic and pluripotency as line plots in Figure 1F and moved detailed heatmaps to the supplementary figure S1D.
- 2) We have moved earlier Figure S1B heatmap to the main figure 1E now and also more clearly define the different stages of reprogramming in the iCD1 system. We have also replaced the ESCs data⁷ that were used in earlier version of figure with iPSCs generated with iCD1 medium conditions and ESCs data that were obtained from OG2 mice⁸.

- 3) We now show colony count data in the supplement (Figure S3H) and have performed an immunostaining with a Nanog antibody to show that *Oct4*-GFP positive cells are also Nanog positive at day 8. The whole well scan is now added to main Figure 1B and enlarged images of representative colonies are in supplementary Figure S1B. We agree that our day 8 cells are a mixed population of both successfully reprogrammed and failed cells. These cells need to be picked and passaged a few times to achieve a level that matches iPSCs and ESCs. However we would like to highlight that even the day 7 *Oct4*-GFP positive cells produced using the iCD1 system can generate chimeric mice with a success rate of 33% (6 out of 18 live chimeric pups)⁶. We now mention this in the main text.
- 4) We present additional publicly available data for ESCs and iPSCs cultured in chemically defined medium or serum/LIF (detailed culture conditions are in supplementary Table S13). These expression data correlate very well with our day 8 *Oct4*-SK cells grown in iCD1 medium (Figure S1E).

Claim 2 - In the abstract, the authors say ‘The early binding profile of Oct4defSox2 resembles Oct4’. Seeing the Venn diagram, Fig S3C. Oct4 binding sites are 11,224, Oct4defSox2 binding sites are 43,654, of which only 6,600 (<60% of Oct4 targets) overlap on day1. They don’t look very similar to me. The claim that Oct4defSox2 cannot open at CO sites (particularly with SoxOct motif) is based on the CO sites within Oct4 bound sites. Chromatin opening ability of Oct4defSox would need to be addressed by looking at sites bound by Oct4defSox. It would be ideal to analyse the Oct4 specific peaks (11,224 - 6,600 = 4,624), the common 6,600 peaks, Oct4defSox2 specific peaks (43,654 - 6,600 = 37,054) separately. In addition, I would like to know how many of them are co-occupied by Sox2, rather than classifying the peaks by the presence or absence of SoxOct motifs. The authors have the Sox2 ChIP-seq data. Then, how many of them are the closed in MEF, open in ESCs (in the published data), and of which how many of them become open in their day 1 data. This would clarify if Oct4, Oct4defSox2, Oct6 can open the target loci with or without Sox2.

>> We thank the reviewer for suggesting this analysis in a clear and constructive manner. We followed the guidelines of the reviewer and defined Oct4-specific, common Oct4/ Oct4^{defSox2} and Oct4^{defSox2} specific sites and asked how opening proceeds in the absence/presence of Sox2. As suggested by the reviewer we performed this analysis for sites that are ‘closed in MEFs, open in ESCs’. We have completely revised Figure 4 and moved our *SoxOct* motif centered analysis to the Figure S6A. We show Venn diagrams listing the number of binding sites in each category (Figure S5E). Overall, based on this new analysis, we conclude:

- 1) Oct4 has a limited capacity to increase chromatin accessibility by itself
- 2) Sox2 appears to be a better pioneer factor than Oct4
- 3) Oct4 augments the chromatin opening capacity of Sox2,
- 4) Oct4^{defSox2} interferes with chromatin opening.

Claim 3 - It is interesting, but I wonder why. If those loci are bound by Oct4defSox2, does it mean Oct4defSox2 inhibits opening? If Oct4defSox2 does not bind there, why they do not open in +Oct4defSox2 reprogramming? Is it possible that Sox2 is responsible for the opening those loci? Are Sox2 binding sites in +Oct4 similar to those in +Oct6 reprogramming, but different in +Oct4defSox2 reprogramming?

>> In response to this insightful comment we reworked our analysis to include Sox2. As the reviewer correctly guessed, we find that Sox2 is indeed responsible for the majority of opening events and we have performed additional analyses to clarify this point (a new revised Figure 4). We also find evidence that Oct4^{defSox2} in many instances inhibits Sox binding and chromatin opening (Figure 4B, and 'Sox2 bound Oct4/Oct4^{defSox2} common cluster' in 4C). This is also revealed by our new chromVAR analysis, which shows that many ectopic peaks that open only in the Oct4^{defSox2} condition are marked by single (i.e. non-composite *SoxOct*) *Sox* motifs. This finding is consistent with a model that presence of Oct4^{defSox2} leads to a re-distribution of Sox factors which (i.e. from sites with composite *SoxOct* element to sites with single *Sox* elements), in turn, opens sites not opened in Oct4-SK and Oct6-SK conditions. These findings are shown and discussed in the revised manuscript in Figure 4A. We do not observe a higher similarity in Sox2 binding in the presence of Oct4 versus Oct6 and Oct4 versus Oct4^{defSox2} conditions (Figure S4D). It is possible that Sox2 facilitates the chromatin changes prior to day 1 or endogenous (Sox) factors contribute to the process. To further study the role of Sox2 in chromatin opening right from the addition of reprogramming factors to MEFs (i.e. day 0). Further, the pioneer concept may need more detailed scrutiny. It suggests that there is a direct causal relationship between TF binding and chromatin opening. Our study suggests that Sox2 is a key facilitator of chromatin opening and this activity is compromised in the presence of Oct4^{defSox2}. Yet, whether this is brought about by a direct dominant negative effect (i.e. Oct4^{defSox2} binds at otherwise opening sites and prevents access by Sox2) is less clear. We now discuss this aspect in more detail in the discussion.

Claim 4 - The fact that Oct4defSox2 can replace Oct4 in ESCs is convincing, but the authors should show how many colonises Oct4defSox2 can generate compared to Oct4. Is the rescue frequency is the same as Oct4? If appearance of the rescued ESC colonies is very rare event, it is not very surprising that it does not work at all in iPSC generation.

>> We now show a whole plate image of rescued colonies with AP staining at passage 1 which shows that the ZHBTc4 ESCs rescue by Oct4^{defSox2} is comparable to Oct4^{WT} and is not a rare event. We now also more clearly refer to previous studies showing that Oct6 is unable to rescue pluripotency in an assay that involves the replacement of Oct4 with alternative TFs.

Claim 5 - It is potentially very interesting, but the data, method have not been described sufficiently. What % of Oct4 bound regions are cloned into the STARR-seq vector? How many Oct4 bound regions showed the enhancer activity in ESCs? Plotting read counts from the library and the reporter (GFP?) +ve cells would be informative. Are the regions which has the enhancer activity not Oct4 targets in ESCs?

>> We have now added heatmaps to Figure S8D that show the reads from STARR-seq library corresponding the Oct4 and Oct6 ChIP peaks. We now show that the coverage for Oct4 bound regions in the STARR-seq library was higher at reprogramming days 1 and 5 (18 and 30%) than for Oct6 bound regions (9 and 14%; Figure S8D). We now indicate these fractions of overlap of POU binding with the STARR-seq library in the Figure S8D as well in main text. We have also added more details to the method section.

For the last question, we used Oct4 ChIP-seq peaks from Plath lab (GSE90895;¹²) in ESCs and intersected them with STARR-seq library. We found that around 50% of Oct4 bound regions in ESCs have an enhancer activity (Reviewer Figure 3 below).

Reviewer figure 3: Heatmaps showing STARR-seq and input library signal on Oct4 bound sites in ESCs downloaded from GSE90895¹². STARR-seq tracks are normalized to input library signal and peaks are ranked by STARR-seq signal.

More specific points

Page 4, 'We found that both Oct6-SK and Oct4defSox2 SK reprogramming cocktails can correctly progress through the early stages of reprogramming' It is not well understood what is 'correct' progress. I assume the authors mean MET, loss of somatic TF expression. However, Cdh1 expression is lower in Oct6-SK reprogramming even on day 3, in Oct4defSox2 SK reprogramming on day 5 than OSK reprogramming. Flow cytometry analysis of Cdh1 would clarify how MET kinetics, efficiency is similar/dissimilar in those conditions.

>> We are really grateful for the reviewer's suggestion for these insightful comments. In response to both this comment and one from reviewer 2; we performed FACS analysis with antibodies for

Cdh1 expression during reprogramming in different cocktails. Indeed Oct6-SK cocktail amongst all other cocktails was the least efficient to activate expression of the Cdh1 protein. We added the new Figures-1G and S2B-C and modified the text in section titled “**Oct4 is dispensable for initiating somatic cell reprogramming**” as well as in discussion.

Page 6 ‘Late binding events were significantly associated with elevated gene expression at day 8 compared to MEF (Figures 2J, 2K) – till end of the paragraph’ Of course, you can find genes which fits with the binding, too (Fig 2K). But are they majority? I do not think violin plot is suitable to see the correlation between binding and expression over the time course. Perhaps the authors can do K-means clustering then show what % of genes which belong to the each group show similar gene expression pattern to the binding patterns (i.e. 010 -> transient up).

We classified gene expression to be unchanged, transiently upregulated, immediately upregulated and late upregulated using differentially expressed genes at days 1 and 8 compared to MEFs and assigned them to the Oct4 binding categories (Reviewer Figure 4). We find that in many cases fractions of expression pattern broadly resemble binding patterns. For example, the late bound category (‘001’) is associated with the highest fraction of genes that are upregulated late. 011, 101 and 111 categories are associated with the highest fraction of constitutively upregulated genes. We however feel that the violin plots are more informative for this figure and like to suggest to keep them in the main text. We are now improving the violin plots by correcting p-values using multiple testing correcting (Holm method) to have a more stringent assessment of the significance level.

Reviewer Figure 4: Bar plots represent the fraction of genes associated with the Oct4 binding trajectories as defined in Figure 2A. Expression categories represent: constitutive up: Expression of genes was up at both day 1 and day 8; unchanged: absolute of $\log_2(\text{FC}) < 0.5$ and transiently upregulated at day 1 or at late stage day 8. Downregulated genes compared to MEFs were discarded.

Page 8 'The majority of sites bound by Oct4 at day 1 were inaccessible in MEF (Figure 4A).' It is not clear what the authors consider as 'inaccessible'. A cut off line would be helpful. **Page 8** 'These loci show an increased accessibility upon Oct4 binding at day 1 indicating that they undergo a closed-to-open transition. These sites are also accessible in Oct6-SK conditions but accessibility is muted in Oct4 defSox2-SK conditions at day 1.' I can somewhat see what the authors claim, but some quantification and statistics (what % became more open compare to MEFs?) would make it more convincing.

>> We updated this figure and separate Oct4 ChIP-seq peaks into sites with high, medium and low accessibility in MEFs based on ATAC-seq read coverage in MEFs at Oct4 ChIP-peaks (Figure 4B, S5C). We have also indicated the total numbers of sites for the three classes of MEFs accessibility in the revised Figures 4B and S5D; and calculated p-values corrected for multiple testing using the pairwise Wilcoxon test.

Page 8 ‘At sites bound by Oct4 at day 5, ChIP-seq signals for Oct6 and Oct4defSox2 are very weak. Nevertheless, chromatin accessibility in the Oct6 condition is comparable to Oct4 but weaker in the Oct4defSox2 condition (Figure 4A, lower panel).’ Is it because of Sox2 present/absent at those loci?

>> Yes, we think Sox2 (and possibly endogenous Sox factors that collaborate with exogenously provided POU factors) have the main role in chromatin opening. We have updated this figure following the request to separate binding into low, medium and high-accessibility in MEFs. We now also plotted Sox2 ChIP-seq signals next to these loci (Figure S5D). At the sites with low accessibility in MEFs, ATAC-seq signals are significantly low for Oct4^{defSox2}-SK condition than the Oct4-SK and Oct6-SK conditions. However, the Sox2 ChIP signals are not significantly different in the three POU conditions. Considering the repulsive behavior of Oct4^{defSox2} towards Sox2, we cannot deny a possibility that these sites got opened at an earlier time point than day 5 in Oct4-SK and Oct6-SK conditions. This phenomenon could be partially explained by our another analysis using chromVAR analysis that showed that in the presence of Oct4^{defSox2} may open sites marked by single Sox motifs right from day 1 that would not open in Oct4 and Oct6 conditions (Figure 4A). This suggests that Oct4^{defSox2} profoundly perturbs Sox-factor mediated chromatin opening. The mutations in Oct4^{defSox2} are expected to interfere with Sox2 binding on the canonical *SoxOct* element but, for example, not on *SoxOct* element with 3-base pair spacers such as on the *Fgf4* element²⁰. We elaborate on these findings in the revised manuscript.

Page 9 ‘The finding that Oct6-SK potently opens pluripotency enhancers establishes that these chromatin state stages by themselves are necessary but insufficient for successful reprogramming.’ As far as I see Fig 4F’s 3 pluripotency loci, opening by Oct6 is much more inefficient compared to Oct4, which correlates well with gene expression (Fig 4G). It is quite different from average of all O4 ChIP-seq peaks overlapped with CO (Fig 4E), which showed even more open by Oct6 than Oct4. A Fig like Fig 4E, but only with pluripotency gene loci would clarify if the authors’ claim is true. In addition, even with Oct4, >90% of cells do not become iPSC. So, it is not very surprising that open chromatin at early stages does not correlate with iPSC formation.

>> We thank the reviewer for this insightful suggestion. We followed the suggestion and intersected the Oct4 bound loci containing *SoxOct* motifs that undergo open-to-close transition with mouse ESCs enhancer sets defined by Bing Ren’s group⁷. From this analysis it is indeed clear now that at day 5 the chromatin accessibility of pluripotency enhancer loci containing *SoxOct* in Oct6-SK condition is lower than Oct4-SK condition. We have now added these observations to figure S6E and updated figure S6F and discuss that there is an important subset of sites that don’t open well in the Oct6-SK condition at day 5.

Minor points

Page 3, Introduction – ‘Replacing Sox2 with other Sox genes such as Sox1, Sox3, Sox4, Sox7 or Sox17 obliterates the activity of reprogramming cocktails’ It is not true. Sox1 could replace Sox2 (ref 21).

>> Yes that was indeed a mistake. It’s corrected now.

Page 4, ‘We found that both Oct6-SK and Oct4defSox2 SK reprogramming cocktails can correctly progress through the early stages of reprogramming’ It is not well understood what is ‘correct’ progress. What authors can say is MET, loss of somatic TF expression

>> In light of results from new FACS analysis we re-wrote results section describing Figure 1G and in the discussion.

Figure 2K - I see Nodal has binding at a more proximal site on d1. Is Nodal also in 100 group? I assume the numbers at the right top corners are scale of Y axis. Trp63, Zeb1 should have scale 40, 49, respectively.

>> We have replaced the three examples with better representative tracks now.

Having read counts graphs (like Fig 4E) on the top (or bottom) of all heatmaps (like 4D) would be helpful.

>> We feel that boxplots better illustrate quantitative effects as line plots are distorted by outliers. We now add boxplots for ChIP-seq and ATAC-seq signals to most of our heatmaps.

References

- 1 Esch, D. *et al.* A unique Oct4 interface is crucial for reprogramming to pluripotency. *Nature cell biology* **15**, 295-301, doi:10.1038/ncb2680 (2013).
- 2 Jauch, R., Choo, S. H., Ng, C. K. & Kolatkar, P. R. Crystal structure of the dimeric Oct6 (POU3f1) POU domain bound to palindromic MORE DNA. *Proteins* **79**, 674-677, doi:10.1002/prot.22916 (2011).
- 3 Nishimoto, M. *et al.* Oct-3/4 maintains the proliferative embryonic stem cell state via specific binding to a variant octamer sequence in the regulatory region of the UTF1 locus. *Molecular and cellular biology* **25**, 5084-5094, doi:10.1128/MCB.25.12.5084-5094.2005 (2005).
- 4 Jerabek, S. *et al.* Changing POU dimerization preferences converts Oct6 into a pluripotency inducer. *EMBO reports* **18**, 319-333, doi:10.15252/embr.201642958 (2017).
- 5 Wapinski, O. L. *et al.* Hierarchical mechanisms for direct reprogramming of fibroblasts to neurons. *Cell* **155**, 621-635, doi:10.1016/j.cell.2013.09.028 (2013).
- 6 Guo, L. *et al.* Resolving Cell Fate Decisions during Somatic Cell Reprogramming by Single-Cell RNA-Seq. *Molecular cell* **73**, 815-829 e817, doi:10.1016/j.molcel.2019.01.042 (2019).
- 7 Shen, Y. *et al.* A map of the cis-regulatory sequences in the mouse genome. *Nature* **488**, 116-120, doi:10.1038/nature11243 (2012).
- 8 Li, D. *et al.* Chromatin Accessibility Dynamics during iPSC Reprogramming. *Cell stem cell* **21**, 819-833 e816, doi:10.1016/j.stem.2017.10.012 (2017).
- 9 Fang, H. T. *et al.* Global H3.3 dynamic deposition defines its bimodal role in cell fate transition. *Nature communications* **9**, 1537, doi:10.1038/s41467-018-03904-7 (2018).
- 10 Hutchins, A. P. *et al.* Models of global gene expression define major domains of cell type and tissue identity. *Nucleic acids research* **45**, 2354-2367, doi:10.1093/nar/gkx054 (2017).
- 11 Chen, J. *et al.* Hierarchical Oct4 Binding in Concert with Primed Epigenetic Rearrangements during Somatic Cell Reprogramming. *Cell reports* **14**, 1540-1554, doi:10.1016/j.celrep.2016.01.013 (2016).
- 12 Chronis, C. *et al.* Cooperative Binding of Transcription Factors Orchestrates Reprogramming. *Cell* **168**, 442-459 e420, doi:10.1016/j.cell.2016.12.016 (2017).
- 13 Knaupp, A. S. *et al.* Transient and Permanent Reconfiguration of Chromatin and Transcription Factor Occupancy Drive Reprogramming. *Cell stem cell* **21**, 834-845 e836, doi:10.1016/j.stem.2017.11.007 (2017).
- 14 Zviran, A. *et al.* Deterministic Somatic Cell Reprogramming Involves Continuous Transcriptional Changes Governed by Myc and Epigenetic-Driven Modules. *Cell stem cell* **24**, 328-341 e329, doi:10.1016/j.stem.2018.11.014 (2019).
- 15 Cui, G. *et al.* Mouse gastrulation: Attributes of transcription factor regulatory network for epiblast patterning. *Development, growth & differentiation* **60**, 463-472, doi:10.1111/dgd.12568 (2018).
- 16 Zhu, Q. *et al.* The transcription factor Pou3f1 promotes neural fate commitment via activation of neural lineage genes and inhibition of external signaling pathways. *eLife* **3**, doi:10.7554/eLife.02224 (2014).
- 17 Soufi, A., Donahue, G. & Zaret, K. S. Facilitators and impediments of the pluripotency reprogramming factors' initial engagement with the genome. *Cell* **151**, 994-1004, doi:10.1016/j.cell.2012.09.045 (2012).
- 18 Soufi, A. *et al.* Pioneer transcription factors target partial DNA motifs on nucleosomes to initiate reprogramming. *Cell* **161**, 555-568, doi:10.1016/j.cell.2015.03.017 (2015).
- 19 Jauch, R. *et al.* Conversion of Sox17 into a pluripotency reprogramming factor by reengineering its association with Oct4 on DNA. *Stem cells* **29**, 940-951, doi:10.1002/stem.639 (2011).

- 20 Remenyi, A. *et al.* Crystal structure of a POU/HMG/DNA ternary complex suggests differential assembly of Oct4 and Sox2 on two enhancers. *Genes & development* **17**, 2048-2059, doi:10.1101/gad.269303 (2003).

Reviewers' comments:

Reviewer #1 (Remarks to the Author):

The authors have addressed almost all of my previous concerns and the manuscript seems appropriate for publication in Nature Communications.

Reviewer #2 (Remarks to the Author):

The authors have made a commendable effort to address the questions of the reviewers, in particular clarifying the context of their study's contribution with regards to previous research and including additional data. These changes significantly improve readability of the manuscript. I only have a few comments outlined below.

What additional information is provided by 3C in addition to 3B?

Sup2 C clearly shows the phenotypic differences in MET and pluripotency induction. Still, I am not 100% convinced by the author's explanation of how Oct6 inhibits MET. If Oct6 binds MORE through homodimerization then why does it impede MET induction, which can be induced by SK only? Does Oct6 interact with Klf4? Although this might be not relevant to their conclusions, the point remains unclear.

The authors claim that Oct6 suppress the somatic program similarly to Oct4. If so, then Oct6 also drives early steps towards reprogramming. However, suppression of the original cell program is not surprising and not necessarily related to iPSC reprogramming since Oct6 also induces cell fate conversion into the neuronal lineage. In accordance, the section title (pg 5) "Oct4 is dispensable for initiating somatic cell reprogramming" should be changed to "Oct4 is dispensable for early silencing of the somatic gene network" as suggested in the "Highlights".

The section title (pg 13) "Sox2/Oct4 heterodimerization is dispensable to target pluripotency genes" is confusing given their results, and suggest it be changed to "Sox2/Oct4 heterodimerization is necessary for persistent binding of pluripotency genes"

With these minor changes, I can recommend the manuscript to be accepted for publication in Nature Communications.

Reviewer #3 (Remarks to the Author):

Cdh1 FACS made it clear that MET happens efficiently in O4DefS2SK reprogramming, but not in O6SK reprogramming. Including RNA-seq data of 'iPSCs generated with iCD1medium conditions and ESCs data that were obtained from OG2 mice (from Ref 8) (Fig S1D)' did not support the claim that the day 8 cells are similar to iPSCs. Fig S1C clearly shows d8 cells are quite different from ESCs. Referencing 'iPSC lines derived with this system show a high rate of contribution to chimeric mice and germline transmission at day 7 even without additional culturing (Ref 44, 46)' also does not guarantee that all the cells in this paper have the same potential, unless it has been confirmed by themselves. Fig S2F (Cdh1, Oct4-GFP FACS) clearly showed that only 50% of day 8 cells are Oct4-GFP+, which is an early marker of pluripotency in reprogramming. Anyway, this time point is not very relevant to this work, and it is not surprising that the RNA-seq data of day 8 is far from ESCs/iPSCs if they did not sort the iPSC-like colonies. They should just describe that the cells and the RNA-seq samples are

heterogeneous honestly and objectively, rather than claiming something which is not true. Nevertheless, authors made the description of the reprogramming system clearer.

I do not think new Fig 4BCDEF are particularly good. Fig 4B shows that ATAC-seq signal from day 1 O4defS2SK reprogramming is lower, but it is based on the Oct4 bound loci in d1 OKS reprogramming (11,224 loci), isn't it? Authors should look at all O4defS2 bound loci. It would be better to analyse the Oct4 specific peaks ($11,224 - 6,600 = 4,624$), the common 6,600 peaks, O4defS2 specific peaks ($43,654 - 6,600 = 37,054$) separately, as I previously recommended. (I am not sure why the authors removed the Venn diagram, which made very clear how similar Oct4, O4defS2 and Oct6 bindings are.)

New Fig 4C is too complicated. I am not sure why the authors are looking at only the ATAC closed open loci between MEFs and ESCs in this figure. Why do they not look at all Oct4, Sox2 bound loci? What I can see in this figure is that even in not Sox2 bound-Oct4 bound sites (top panels in Fig 4C - which should not be influenced by exogenous Sox2 directly), the common Oct4 bound sites are more open in O4SK than O4defSoxSK, i.e. independent from Sox2, Oct4defSox2 has less chromatin opening activity than Oct4. However, it would need to be confirmed in all non-Sox2 bound, Oct4 and O4defS2 common binding sites. In Sox2 bound-Oct4 bound sites (bottom panels in Fig 4C), many of the common bound sites in O4defS2SK does not have Sox2 binding, thus the boxplot is comparing Oct4+Sox2 bound loci vs O4defS2 bound loci, which meaning I do not know. If you look at only sites which have Sox2 co-binding, the difference in the ATAC signals in O4SK and O4defS2SK is small.

A better way to ask if Oct4defSox2 interiors chromatin opening by Sox2 could be;

Using MEF ATAC-seq and O4SK d1 ChIP-seq, ATAC-seq data

1. Select loci which became open on d1 (C->O)
2. Group them into Sox2 bound or non-bound sites
3. Sub group based on Oct4 bound or not.
4. Do the same analysis with MEF ATAC-seq and O4defSoxSK d1 ChIP-seq, ATAC-seq data

If the number of Sox2 bound, O4defSox bound sites is proportionally much smaller than that of Sox2 bound, Oct4 bound sites, I would agree that O4defSox interferes chromatin opening at the Sox2 binding sites. (d1 ATAC-seq signals in the 4 groups with heatmap, box plot could also be used to check if any of the groups have stronger signals, although all loci are supposed to be 'open'.)

While I did not think looking at only Oct4 bound loci in O4defS2SK reprogramming is fair, if I look at Fig 4B and S5C, I agree that these loci are less open in O4DefS2SK reprogramming and have less Sox2 binding. This needs to be confirmed in all O4defS2 bound sites, in comparison to Oct4 bound loci in OSK reprogramming. If it is confirmed, it would be fair to say that O4defS2 has less chromatin opening activity compared to Oct4 (independent from Sox2) and less prone to co-bind with Sox2, which has strong chromatin opening activity, rather than saying O4defS2 interferes with chromatin opening.

Fig 4D. I do not think looking at the Oct4/O4defS2 binding loci from this work in the 1/2 factor reprogramming data (GSE93029. Li, et al., Fig 4E, S6CD) is fair to make the conclusion 'Sox2 appears to be a better pioneer factor than Oct4', either. In 1/2 factor expressed MEFs, opened chromatin loci might be different. Isn't it fairer to compare the numbers of open chromatin loci, signal intensity across the whole genome in the published O, S, OS ATAC-Seq data? Although the conclusion might be the same, the authors need to make sure the sampled loci represent the global data. Anyway, I'm not surprised that Oct4+Sox2 can open more than Oct4 alone or Sox2 alone. If Oct4 alone has limited chromatin opening activity compared to Sox2 alone, it could also be confirmed with GSE90895 (from the Plath lab)?

Similarly, I do not think Fig S6AB is a good analysis, since O4 ChIP-seq peaks does not overlap well with Oct6 or O4defS2.

Fig S5E let me realize that Sox2 binding sites are much fewer in O4defS2SK reprogramming (5,760), compared to OSK reprogramming (15,195), with common binding sites 2,106. It is quite opposite of Oct4/O4defS2 binding (O4defS2SK = 37,054, Oct4 = 4,626, including common 6,600). It is interesting that loci with Sox motifs are more open in O4defS2SK reprogramming (Fig 4A). Is it possible that O4defS2 open different loci from Oct4 in collaboration with other endogenously expressed Sox protein?

In order to claim 'we find that Sox2 is indeed responsible for the majority of opening events', the authors need to analyse if how many of Oct4 unique bound loci, Sox2 bound loci, Oct4/Sox2 co-bound loci became open from MEF to d1 of OSK reprogramming. If there is such an analysis, I apologize, but I do not think I can find it. In addition, I wonder the conclusion 'binding of Sox2 appears to be a better determinant of chromatin accessibility changes than POU factor binding' is really true. ATAC-seq data seem to be similar in OSK, O4defS2SK, O6SK reprogramming day 1 (Fig S5A), but POU and Sox2 CHIP-seq data doesn't seem to be similar (Fig S4CD).

'Oct4defSox2 could maintain ESC self-renewal in the absence of wild-type Oct4, highlighting difference between pluripotency induction and maintenance' is supported better with the new data.

I still could not understand the STARR-seq data well. DNA fragments from E14 FAIRE was cloned into the reporter vector. What % of enhancers (or ESC ATAC-seq peaks) were covered by this library (before expressed in ESCs)? How much of them overlapped with Plath lab's Oct4 CHIP-seq data? And 50% of the library had enhancer (i.e. reporter) activity? Fig S8D shows overlap with CHIP-seq data and STARR-seq library (before the reporter assay)? The signals are read counts from the reporter assay. How many loci in each category of Fig S8D are considered to be positive in the STARR-seq? Conversely, what % of functional enhancers detected in ESC STARR-seq could be bound by Oct4 (O4defS2, Oct4), Sox2 in OSK, O4defS2SK, O6SK reprogramming day 1? It is potentially very interesting data, but it still need more explanation and data analyses.

The Reviewer Figure 4 showed that only <30% of 0-0-1 genes are Late up. It means 70% of Oct4 binding does not correlate with expression pattern. Similarly, only ~25% of 1-0-0 genes has transient up (d1), i.e. 75% of genes does not show the matched expression change. Namely, majority of gene expression pattern does not fit with Oct4 binding. Fig 2J does not show that Oct4 binding and expression correlate, Fig 2K is not representing the majority of the gene expression changes, i.e. a dishonest figure.

Fig S4C should be in the main figure instead of Fig 3B, which is very difficult to understand the message.

Reviewer #1 (Remarks to the Author):

The authors have addressed almost all of my previous concerns and the manuscript seems appropriate for publication in Nature Communications.

>> We thank the reviewer for the efforts and suggestions to improve the manuscript.

Reviewer #2 (Remarks to the Author):

The authors have made a commendable effort to address the questions of the reviewers, in particular clarifying the context of their study's contribution with regards to previous research and including additional data. These changes significantly improve readability of the manuscript. I only have a few comments outlined below.

What additional information is provided by 3C in addition to 3B?

>> We have added this figure in response to a request by reviewer 3 in the 1st round of revision:

"Having read counts graphs (like Fig 4E) on the top (or bottom) of all heatmaps (like 4D) would be helpful."

We find boxplots more realistic representations than read count graphs as the latter could be skewed by some outliers with disproportionately high signals. We have adjusted Figure 3C so it aligns with the heatmaps in Figure 3B to clarify that it is a quantitative representation of the data in the heatmap..

Sup2 C clearly shows the phenotypic differences in MET and pluripotency induction. Still, I am not 100% convinced by the author's explanation of how Oct6 inhibits MET. If Oct6 binds MORE through homodimerization then why does it impede MET induction, which can be induced by SK only? Does Oct6 interact with Klf4? Although this might be not relevant to their conclusions, the point remains unclear.

>> The observation that Cdh1 is suppressed by the Oct6-SK cocktail but not by the two-factor SK cocktail is indeed an interesting finding. We agree with the reviewer that the present study does not provide a conclusive answer as to the mechanism how Oct6 impedes the MET. To explore the possibility of a direct inhibition of Klf4 by Oct6 raised by the reviewer, we intersected our day 1 Oct4 and Oct6 day 1 ChIP-seq with Klf4 ChIP-seq in pMX-OSK system used by the Plath lab¹. The fraction of peaks overlapping with Klf4 peaks is higher for Oct4 than for Oct6 (28% versus 13%; reviewer Figure 1). Whilst this is neither direct evidence for nor against a physical interaction between Klf4 and Oct6; it implies that it is unlikely for Oct6 to physically bind to and compete with Klf4.

On a related note: A recently published study by Hans Schöler's lab (co-author in our study) showed that a chimeric Brn4-Klf4 fusion protein could bind to known pluripotency enhancers and support pluripotency induction but separated Brn4 and Klf4 fail to do so². A shift from MORE motifs (Brn4 alone) to SoxOct motifs (Brn4-Klf4 fusion protein) was suggested to be responsible for this effect. This argues against a general interference of POUIII factors with Klf4 function in pluripotency induction. We rather think that Oct6 impedes the MET indirectly rather than by directly antagonising Klf4. As the reviewer correctly pointed out these interesting aspects are not relevant for the conclusion of the present manuscript and should be addressed in future studies. We amend the discussion on pages 15/16.

Reviewer Figure 1: Intersection of ChIP-seq peaks of Oct4 (left Venn) and Oct6 (Right Venn) with Klf4 ChIP-seq at 48 hours of reprogramming. Klf4 ChIP-seq (from pMX-OSK) peaks were downloaded from¹.

The authors claim that Oct6 suppress the somatic program similarly to Oct4. If so, then Oct6 also drives early steps towards reprogramming. However, suppression of the original cell program is not surprising and not necessarily related to iPSC reprogramming since Oct6 also induces cell fate conversion into the neuronal lineage. In accordance, the section title (pg 5) "Oct4 is dispensable for initiating somatic cell reprogramming" should be changed to "Oct4 is dispensable for early silencing of the somatic gene network" as suggested in the "Highlights".

>> We thank the reviewer and changed the section title.

The section title (pg 13) "Sox2/Oct4 heterodimerization is dispensable to target pluripotency genes" is confusing given their results, and suggest it be changed to "Sox2/Oct4 heterodimerization is necessary for persistent binding of pluripotency genes"

>> We followed the suggestion and modified the section title.

With these minor changes, I can recommend the manuscript to be accepted for publication in Nature Communications.

>> We thank the reviewer for the many constructive comments that helped us to improve the manuscript.

Reviewer #3 (Remarks to the Author):

Cdh1 FACS made it clear that MET happens efficiently in O4DefS2SK reprogramming, but not in O6SK reprogramming. Including RNA-seq data of 'iPSCs generated with iCD1medium conditions and ESCs data that were obtained from OG2 mice (from Ref 8) (Fig S1D)' did not support the claim that the day 8 cells are similar to iPSCs. Fig S1C clearly shows d8 cells are quite different from ESCs. Referencing 'iPSC lines derived with this system show a high rate of contribution to chimeric mice and germline transmission at day 7 even without additional culturing (Ref 44, 46)' also does not guarantee that all the cells in this paper have the same potential, unless it has been confirmed by themselves. Fig S2F (Cdh1, Oct4-GFP FACS) clearly showed that only 50% of day 8 cells are Oct4-GFP+, which is an early marker of pluripotency in reprogramming. Anyway, this

time point is not very relevant to this work, and it is not surprising that the RNA-seq data of day 8 is far from ESCs/iPSCs if they did not sort the iPSC-like colonies. They should just describe that the cells and the RNA-seq samples are heterogeneous honestly and objectively, rather than claiming something which is not true. Nevertheless, authors made the description of the reprogramming system clearer.

>> We agree with the reviewer and revised the text to now explain on page 5 (results section) that day 8 Oct4-SK cells are heterogeneous unsorted bulk cell populations as we did not sort for Oct4-GFP positive and negative cells and point out that our day 8 do not fully resemble mature and passaged ESCs/iPSCs.

I do not think new Fig 4BCDEF are particularly good. Fig 4B shows that ATAC-seq signal from day 1 Oct4-defSox2 reprogramming is lower, but it is based on the Oct4 bound loci in d1 OKS reprogramming (11,224 loci), isn't it?

>> We had prepared Figure 4B (previously Figure 4A) in response to an astute suggestion by reviewer 3 in the previous round of revision:

"Page 8 'The majority of sites bound by Oct4 at day 1 were inaccessible in MEF (Figure 4A).' It is not clear what the authors consider as 'inaccessible'. A cut off line would be helpful. Page 8 'These loci show an increased accessibility upon Oct4 binding at day 1 indicating that they undergo a closed-to-open transition. These sites are also accessible in Oct6-SK conditions but accessibility is muted in Oct4-defSox2-SK conditions at day 1.' I can somewhat see what the authors claim, but some quantification and statistics (what % became more open compare to MEFs?) would make it more convincing."

We feel this figure is very helpful to understand the close-to-open dynamics at Oct4 bound sites compared to the starting MEFs and like to retain this figure. The separate analysis compare Oct4 versus Oct4^{defSox2} sites follows the revised supplementary Figures S6A-E.

Authors should look at all Oct4-defSox2 bound loci. It would be better to analyse the Oct4 specific peaks (11,224 - 6,600 = 4,624), the common 6,600 peaks, Oct4-defSox2 specific peaks (43,654 - 6,600 = 37,054) separately, as I previously recommended. (I am not sure why the authors removed the Venn diagram, which made very clear how similar Oct4, Oct4-defSox2 and Oct6 bindings are.)

>> We had updated the Venn diagram replacing the Oct6 ChIP-seq peaks with Sox2 in Oct4 and Sox2 in Oct4^{defSox2} ChIP-seq peaks at day 1 during the revision based on the suggestion made in the previous round of the revision by reviewer 3 (figure S6A). In addition we now also include the previous Venn diagram in a revised Figure S4E. We regret the confusion but we had in fact analysed Oct4 specific, Oct4^{defSox2} specific and common peaks in Figure 4C of the previous version. We now elaborate on and refine this analysis using the detailed guidance given by the reviewer below.

New Fig 4C is too complicated. I am not sure why the authors are looking at only the ATAC closed open loci between MEFs and ESCs in this figure. Why do they not look at all Oct4, Sox2 bound loci?

>> We agree that the analysis to separately analyse loci common and unique for Oct4/Oct4^{defSox2} is useful and informative. We furthermore agree with the reviewer that a filter that considers only loci closed in MEFs and open in ESCs is not required and it is preferable to look at all Sox2 and Oct4 bound loci. We now prepared a revised figure S6A-D replacing figure 4C that

eliminates the close-to-open peaks filter (i.e. we consider all sites rather than just the sites closed in MEFs and open in ESCs). With this larger set of peaks we can corroborate the two key conclusions that (i) the presence a Sox2 more strongly correlates with opening than the presence of Oct4 and (ii) that Oct4 opens chromatin more effectively than Oct4^{defSox2} at co-bound Oct4/Sox2 sites. We moved this analysis to the supplement as we now present the analyses inspired by the additional constructive suggestions following below that leads to a revised and in our opinion much improved main text Figure 4C-I.

What I can see in this figure is that even in not Sox2 bound-Oct4 bound sites (top panels in Fig 4C - which should not be influenced by exogenous Sox2 directly), the common Oct4 bound sites are more open in O4SK than O4defSoxSK, i.e. independent from Sox2, Oct4^{defSox2} has less chromatin opening activity than Oct4. However, it would need to be confirmed in all non-Sox2 bound, Oct4 and O4defS2 common binding sites. In Sox2 bound-Oct4 bound sites (bottom panels in Fig 4C), many of the common bound sites in O4defS2SK does not have Sox2 binding, thus the boxplot is comparing Oct4+Sox2 bound loci vs O4defS2 bound loci, which meaning I do not know.

>> We agree that additional analysis as to whether Oct4^{defSox2} brings about accessibility changes at sites not bound by Oct4 in the absence/presence of Sox2 is beneficial. We thus performed an analysis step-by-step as the reviewer suggest below and included the panels in an updated figures 4C-E and S5D-F.

If you look at only sites which have Sox2 co-binding, the difference in the ATAC signals in O4SK and O4defS2SK is small.

A better way to ask if Oct4^{defSox2} interiors chromatin opening by Sox2 could be; Using MEF ATAC-seq and O4SK d1 ChIP-seq, ATAC-seq data

1. Select loci which became open on d1 (C->O)

>> Done. We used our ATAC-seq data and defined MEF to d1 closed-to-open (CO) category (updated Figure 4C-D)

2. Group them into Sox2 bound or non-bound sites

>> Done. We classified these sites by the presence/absence of Sox2 (Figure 4E)

3. Sub group based on Oct4 bound or not.

>> Done. We further classified these sites by the absence/presence of Oct4 (Figure 4E)

4. Do the same analysis with MEF ATAC-seq and O4defSoxSK d1 ChIP-seq, ATAC-seq data

>> Done. We performed the same analysis for Oct4^{defSox2} (Figure S5 D-F)

>> We are grateful for these in depth suggestions which leads to clearer panels (Figure 4C-E) indicating that Sox2 facilitates chromatin opening more potently than Oct4 and Oct4 augments opening when it co-binds with Sox2.

If the number of Sox2 bound, O4defSox bound sites is proportionally much smaller than that of Sox2 bound, Oct4 bound sites, I would agree that O4defSox interferes chromatin opening at the Sox2 binding sites. (d1 ATAC-seq signals in the 4 groups with heatmap, box plot could also be used to check if any of the groups have stronger signals, although

all loci are supposed to be 'open'.)

>> We wish to politely point that we find that Oct4^{defSox2} interferes with Sox2 driven chromatin opening in the context of the canonical *SoxOct* motif (consensus sequence CATTGTCATGCAAAT). It does not necessarily interfere with the opening activity at other (possibly ectopic) sites that lack this motif. Yet, sites with the canonical *SoxOct* are critical for pluripotency reprogramming. We now clarify this point in updated Figures 4F-I that show that there is co-binding for Oct4/Sox2 as well as for Oct4^{defSox2}/Sox2. However, the majority of this co-binding occurs at dissimilar genomic locations (Figure 4F-G). Whilst ~40% of co-bound Oct4/Sox2 sites have matches for canonical *SoxOct* motifs, only ~10% of Oct4^{defSox2}/Sox2 co-bound sites show matches to this motif (Figure 4H). Oct4^{defSox2} can target many of the Oct4/Sox2 co-bound sites. However, in the presence of Oct4^{defSox2}, Sox2 ChIP-seq signals are strongly reduced indicating that in the context of the *SoxOct* element Sox2 binding is blocked (Figure 4F-I).

While I did not think looking at only Oct4 bound loci in O4defS2SK reprogramming is fair, if I look at Fig 4B and S5C, I agree that these loci are less open in O4DefS2SK reprogramming and have less Sox2 binding. This needs to be confirmed in all O4defS2 bound sites, in comparison to Oct4 bound loci in OSK reprogramming. If it is confirmed, it would be fair to say that O4defS2 has less chromatin opening activity compared to Oct4 (independent from Sox2) and less prone to co-bind with Sox2, which has strong chromatin opening activity, rather than saying O4defS2 interferes with chromatin opening.

>> We revised Figures 4C-I; supplementary figure S5D-F; and S6 following the 4 steps outline by the reviewer above to accommodate this suggestion.

The reason for looking at Oct4 bound loci in Oct4^{defSox2} was motivated by the goal to understand chromatin dynamics at loci that are critical for successful reprogramming. We had therefore focused on Oct4 bound loci as reference.

Fig 4D. I do not think looking at the Oct4/O4defS2 binding loci from this work in the 1/2 factor reprogramming data (GSE93029. Li, et al., Fig 4E, S6CD) is fair to make the conclusion 'Sox2 appears to be a better pioneer factor than Oct4', either. In 1/2 factor expressed MEFs, opened chromatin loci might be different. Isn't it fairer to compare the numbers of open chromatin loci, signal intensity across the whole genome in the published O, S, OS ATAC-Seq data?

>> We wish to politely point out that such an analysis has already been performed by Li et al. *Cell Stem Cell* in their Figure 4A, B and C³. So we feel replicating their analysis is not necessary in the present manuscript. However, Li et al did not include ChIP-seq data. Using our ChIP-seq data as reference, we can now demonstrate that in single and two-factor conditions (S, O, OS) closed-to-open transitions are substantially stronger at sites bound by Sox2 (Figure 4E).

Although the conclusion might be the same, the authors need to make sure the sampled loci represent the global data. Anyway, I'm not surprised that Oct4+Sox2 can open more than Oct4 alone or Sox2 alone. If Oct4 alone has limited chromatin opening activity compared to Sox2 alone, it could also be confirmed with GSE90895 (from the Plath lab)?

>> We thank the reviewer for suggesting to include this important dataset in our analysis pipeline. We now added O and S data from the Plath lab¹ in a revised Figure 4E and supplementary figure S5F. Data are consistent with O, S and OS data from Li et al³ and corroborate our conclusion that Sox2 is more potent than Oct4 in bringing about chromatin opening.

Similarly, I do not think Fig S6AB is a good analysis, since O4 ChIP-seq peaks does not overlap well with Oct6 or O4defS2.

>> We feel that our manuscript benefits from an analysis for sites filtered for the presence of the canonical *SoxOct* motif. This complements the analysis shown in Figures 4F-I. The Oct4^{defSox2} was designed to disrupt Sox2/Oct4 heterodimers on the canonical *SoxOct* DNA element but not on other DNA elements⁴. Figure S6AB (S7AB in the current version) highlights that the effect of interfering with opening is more pronounced on sites containing *SoxOct* motif versus sites without this DNA motif. We include additional Figures 4F, 4H showing that opening sites bound by Oct4^{defSox2} and Sox2 have a smaller fraction of sites with *SoxOct* than Oct4/Sox2 shared sites. The global chromVAR analysis further supports the conclusion that opening by Oct4^{defSox2} is compromised at sites with *SoxOct* elements but proceeds unabated at, for example, sites with single Sox elements (Figure 4A). This Figure is complementary to other analyses that we carried out in response to the reviewer.

Fig S5E let me realize that Sox2 binding sites are much fewer in O4defS2SK reprogramming (5,760), compared to OSK reprogramming (15,195), with common binding sites 2,106. It is quite opposite of Oct4/O4defS2 binding (O4defS2SK = 37,054, Oct4 = 4,626, including common 6,600). It is interesting that loci with Sox motifs are more open in O4defS2SK reprogramming (Fig 4A). Is it possible that O4defS2 open different loci from Oct4 in collaboration with other endogenously expressed Sox protein?

>> Yes it is possible – we mention the possibility of a role of endogenous Sox factors in the discussion. We believe the opening of ectopic sites marked by single Sox motifs in the Oct4^{defSox2} condition is due to a re-distribution of exogenous Sox2 and possibly also endogenously expressed Sox factors. We discuss this possibility in the manuscript. Our new analysis in figure 4F, G, H shows that the Oct4^{defSox2}/Sox2 co-binding happens at the sites that are mostly different from Oct4/Sox2 co-bound sites and lack *SoxOct* motifs.

In order to claim ‘we find that Sox2 is indeed responsible for the majority of opening events’, the authors need to analyse if how many of Oct4 unique bound loci, Sox2 bound loci, Oct4/Sox2 co-bound loci became open from MEF to d1 of OSK reprogramming. If there is such an analysis, I apologize, but I do not think I can find it. In addition, I wonder the conclusion ‘binding of Sox2 appears to be a better determinant of chromatin accessibility changes than POU factor binding’ is really true. ATAC-seq data seem to be similar in OSK, O4defS2SK, O6SK reprogramming day 1 (Fig S5A), but POU and Sox2 ChIP-seq data doesn’t seem to be similar (Fig S4CD).

>> We re-phrase this statement to “Sox2 appears to be a better facilitator of chromatin accessibility changes than POU factors”. We performed the analysis suggested by the reviewer and found that in our dataset accessibility increase is stronger in singly bound Sox2 or sites co-bound by Oct4/Sox2 than for singly bound Oct4 sites (Figure 4E). In addition, this finding is supported by the accessibility patterns observed by the Pei and Plath labs at these location for single factor (O, S) and double factor (OS) experiments.

‘Oct4defSox2 could maintain ESC self-renewal in the absence of wild-type Oct4, highlighting difference between pluripotency induction and maintenance’ is supported better with the new data.

>> We thank the reviewer for appreciating our revision.

I still could not understand the STARR-seq data well. DNA fragments from E14 FAIRE was cloned into the reporter vector. What % of enhancers (or ESC ATAC-seq peaks) were covered by this library (before expressed in ESCs)?

>> In the revised manuscript, we have expanded the materials and methods section for a more comprehensive description of the STARR-seq experiments. In addition, we have revised the last paragraph of the results describing the STARR-seq analysis with the aim to make it easier to understand. Our library covers 59% of DNase-I peaks in E14 cells (ENCODE, GSM1014154^s sFig9 E) and also genome-wide read distribution of our library correlates well with the DNase-seq data as can be seen on sFig9 D.

How much of them overlapped with Plath lab's Oct4 ChIP-seq data? And 50% of the library had enhancer (i.e. reporter) activity?

>> To better illustrate the overlap between STARR-seq signal and regions occupied by Oct4 in ESC, we added sFig9F. The analysis shows that 18,066 (49.5%) of all 36,477 Oct4 binding sites are significantly enriched in the input library and that 1,940 (11%) of these 18,066 Oct4 sites showed an overlap with called active STARR-seq enhancers. Notably, our criteria to call active enhancers peaks is quite stringent and likely underestimates the overlap between Oct4 bound regions and STARR-seq peaks. Accordingly, when we plot STARR-seq reads for all Oct4 occupied regions, we find a clear signal for most occupied regions in both the input library and for the experiments where we assayed STARR-seq activity (heatmaps sFig 9F). Regarding the % of the library with enhancer activity: Our library contained 352,062 regions of which 6,022 were active (2%). For Oct4 bound regions we called 1,940 active enhancers out of 18,066 regions covered in the library (11%) indicating that Oct4 bound regions are disproportionately likely to be active enhancers in our ESCs.

Fig S8D shows overlap with ChIP-seq data and STARR-seq library (before the reporter assay)? The signals are read counts from the reporter assay.

>> We agree with the reviewer, that this figure needs further explanation. To assess Oct4 or Oct6 bound regions for their STARR-seq activity at day 1 and day 5 of reprogramming; we first filtered for sites, which are represented in the STARR-seq library (sFig 9G). Next, these overlapping regions were grouped by enriched sequence motifs and for each group we plotted the input-normalized STARR-seq signal to get quantitative information regarding enhancer activity for each group (sFig 9H). A more detailed description of how these plots were generated has been added to the materials and methods section.

How many loci in each category of Fig S8D are considered to be positive in the STARR-seq?

>> To address this request, we have added a table (sFig 9G), which shows the percentage of overlap for Oct4 or Oct6 bound regions at day 1 and day 5 of reprogramming with STARR input library as well as STARR-seq enhancers.

Conversely, what % of functional enhancers detected in ESC STARR-seq could be bound by Oct4 (O4defS2, Oct4), Sox2 in OSK, O4defS2SK, O6SK reprogramming day 1? It is potentially very interesting data, but it still need more explanation and data analyses.

>> We agree with the reviewer that the STARR-seq data likely contains a wealth of additional information and have plans to write a manuscript with a more thorough analysis in the future.

For the current manuscript, we chose to limit our analysis to the comparison of Oct4 versus Oct6 occupied regions which is in part motivated by the fact that the manuscript in its current form already contains an abundant amount of information.

The Reviewer Figure 4 showed that only <30% of 0-0-1 genes are Late up. It means 70% of Oct4 binding does not correlate with expression pattern. Similarly, only ~25% of 1-0-0 genes has transient up (d1), i.e. 75% of genes does not show the matched expression change. Namely, majority of gene expression pattern does not fit with Oct4 binding. Fig 2J does not show that Oct4 binding and expression correlate, Fig 2K is not representing the majority of the gene expression changes, i.e. a dishonest figure.

>> Figures 2J and 2K served illustrative purposes and are not required to support the key findings of this study. As the reviewer feels they are not helpful and may distract the reader we decided to remove them from the revised version of the manuscript.

Fig S4C should be in the main figure instead of Fig 3B, which is very difficult to understand the message.

>> The aim of this figure is to show that Oct4^{defSox2} is able to target many Oct4 locations at day 1 whilst Oct6 is binding different sites at both time points. We would like to request to maintain the same order of figures as we feel that the Fig S4C are partially redundant with figure 6A where we compare Oct4 and Oct6 differentially bound sites. In addition we have added revised Venn diagrams for S4C-D in S4E-F. To further improve clarity of Figure 3B we now align it with the boxplot in Figure 3C.

References

- 1 Chronis, C. *et al.* Cooperative Binding of Transcription Factors Orchestrates Reprogramming. *Cell* 168, 442-459 e420, doi:10.1016/j.cell.2016.12.016 (2017).
- 2 Velychko, S. *et al.* Fusion of Reprogramming Factors Alters the Trajectory of Somatic Lineage Conversion. *Cell reports* 27, 30-39 e34, doi:10.1016/j.celrep.2019.03.023 (2019).
- 3 Li, D. *et al.* Chromatin Accessibility Dynamics during iPSC Reprogramming. *Cell stem cell* 21, 819-833 e816, doi:10.1016/j.stem.2017.10.012 (2017).
- 4 Remenyi, A. *et al.* Crystal structure of a POU/HMG/DNA ternary complex suggests differential assembly of Oct4 and Sox2 on two enhancers. *Genes & development* 17, 2048-2059, doi:10.1101/gad.269303 (2003).
- 5 Yue, F. *et al.* A comparative encyclopedia of DNA elements in the mouse genome. *Nature* 515, 355-364, doi:10.1038/nature13992 (2014).

REVIEWERS' COMMENTS:

Reviewer #3 (Remarks to the Author):

The authors have addressed most of my concerns, and I think the manuscript became better. The below is my comment and the authors can take or ignore.

I still have questions about the STARR-seq data. e.g. Is 'The STARR-input library showed a good correlation with DNase-seq (DNase I hypersensitive sites sequencing) performed in E14 ESCs with 59% overlapping peaks (Supplementary Figure 9D-E).' true? Only 15% of the STARR -input library overlaps with the DNase-seq data. Is the quality of STARR-seq data good? Only 6,022 active enhancers from 352,062 regions have identified (1.7%), in contrast human ESC STARR-seq data 32,353 active enhancers/361,737 regions in the library (8.9%) (Barakat, 2018). Although this might be acceptable, considering even CHIP-seq data of Oct4 in ESCs from different labs do not overlap well.

The authors described 'Oct6 bound regions covered by the STARR-seq library are virtually devoid of enhancer activity, indicating that most of the Oct6 binding occurs at regions that do not function as enhancers in ESCs.' I have calculated the numbers of active enhancers in ESC that are occupied by Oct4, Oct6 during reprogramming (see the attached table – I hope it is correct interpretation of the description.). It is true that 99.6% of Oct6 bound sites in day 1 of reprogramming are not ESC active enhancer, but 94.7% of Oct4 bound sites in ESCs are not active enhancers, either, i.e. most of Oct4 bound sites in ESCs do not function as enhancers in ESCs. The data is more informative if the authors use the numbers of ESC active enhancers bound by Oct4/Oct6 (yellow in the attached table) and % of them (green in the attached table). It becomes clear that in ESCs, 32% of active enhancers are bound by Oct4 (a reasonable number). In reprogramming, Oct4 can binds only 2.2% (at day1), 3.2% (at day5) of ESC active enhancers. If you use Oct6 it becomes even lower, 0.8% at day1, 0.3% at day5. This reflects more why efficiency of successful reprogramming is so low. The conclusion is the same, Oct6 is worse than Oct4, but these numbers seem to be more meaningful to understand reprogramming mechanisms.

Finally, using Venn diagrams which reflect size of numbers is easier for the readers to interpret the data. It is something these authors can consider.

	ES (GSE90895)		Oct4 reprogramming		
		% of all active enhancers in ESC	Day1	% of all active enhancers in ESC	Day5
Oct4/Oct6 bound	36,477		8,159		4,842
Overlap with STARR input library	18,066		1,428		1,428
No. of ESC active enhancers bound by Oct4/Oct6 (of 6,022)	1,940	32.2	130	2.2	194
% of Oct4/Oct6 bound sites that are active enhancer in ESC	5.3		1.6		4.0

	Oct6 reprogramming			
% of all active enhancers in ESC	Day1	% of all active enhancers in ESC	Day5	% of all active enhancers in ESC
	12,390		3,201	
	1,090		445	
3.2	50	0.8	20	0.3

	0.4		0.6	
--	-----	--	-----	--

REVIEWERS' COMMENTS:

Reviewer #3 (Remarks to the Author):

The authors have addressed most of my concerns, and I think the manuscript became better.

>> We thank the reviewer for their insightful comments and time to help us improve the manuscript. We really appreciate the detailed input to improve the manuscript.

The below is my comment and the authors can take or ignore.

I still have questions about the STARR-seq data. e.g. Is 'The STARR-input library showed a good correlation with DNase-seq (DNase I hypersensitive sites sequencing) performed in E14 ESCs with 59% overlapping peaks (Supplementary Figure 9D-E).' true? Only 15% of the STARR -input library overlaps with the DNase-seq data. Is the quality of STARR-seq data good?

>> We appreciate the reviewers request for a proper quality check of the used FAIRE-STARR-seq input library. To this end, we performed a genome-wide read coverage correlation analysis (Supplementary Fig. 9D) comparing our input library to the DNase-seq coverage from ENCODE. We chose this approach since it shows overlap of open regions in a quantitative way in contrast to a simple intersection analysis using called peaks which is heavily influenced by the cut-offs chosen to call significant peaks (Supplementary Fig. 9E). The correlation analysis reveals that the strongly enriched regions are clearly shared between the two datasets, while the low to medium enriched regions show a shift towards a higher enrichment for our FAIRE-STARR-seq input library. Notably, these two datasets represent two different biochemical assays, addressing the same question, and were performed and analyzed in two different laboratories. Hence, some discrepancy in the number and positions of called peaks is to be expected. Taken together, we are confident that our FAIRE-STARR-seq input library comprehensively reflects accessible regions within the genome of mouse ESCs.

Only 6,022 active enhancers from 352,062 regions have identified (1.7%), in contrast human ESC STARR-seq data 32,353 active enhancers/361,737 regions in the library (8.9%) (Barakat, 2018). Although this might be acceptable, considering even ChIP-seq data of Oct4 in ESCs from different labs do not overlap well.

>> We agree that these diverging absolute numbers of active enhancers between similar experiments might be confusing. However, as the reviewer mentioned different analyses pipelines and applied thresholds make it difficult to compare absolute numbers.

The authors described 'Oct6 bound regions covered by the STARR-seq library are virtually devoid of enhancer activity, indicating that most of the Oct6 binding occurs at regions that do not function as enhancers in ESCs.' I have calculated the numbers of active enhancers in ESC that are occupied by Oct4, Oct6 during reprogramming (see the attached table - I hope it is correct interpretation of the description.). It is true that 99.6% of Oct6 bound sites in day 1 of

reprogramming are not ESC active enhancer, but 94.7% of Oct4 bound sites in ESCs are not active enhancers, either, i.e. most of Oct4 bound sites in ESCs do not function as enhancers in ESCs.

>> 95.4% of all Oct6 binding sites at day 1 during reprogramming are not active enhancers (51/1,096 Oct6 sites in the library are active). This number is 90.9% for Oct4 binding sites covered by the library. However, as Supplementary Fig. 9F shows, the majority of Oct4 binding sites in mESCs are actually covered in the input library and show STARR-seq activity. Thus, the low percentage can be explained by the stringent criteria we chose to call active regions and likely underestimate the actual percentage of Oct4-bound regions with enhancer activity.

The data is more informative if the authors use the numbers of ESC active enhancers bound by Oct4/Oct6 (yellow in the attached table) and % of them (green in the attached table). It becomes clear that in ESCs, 32% of active enhancers are bound by Oct4 (a reasonable number). In reprogramming, Oct4 can binds only 2.2% (at day1), 3.2% (at day5) of ESC active enhancers. If you use Oct6 it becomes even lower, 0.8% at day1, 0.3% at day5. This reflects more why efficiency of successful reprogramming is so low. The conclusion is the same, Oct6 is worse than Oct4, but these numbers seem to be more meaningful to understand reprogramming mechanisms.

Finally, using Venn diagrams which reflect size of numbers is easier for the readers to interpret the data. It is something these authors can consider.

>> We agree that this is a good mechanistic insight into lower reprogramming efficiencies by TF mediated cell fate conversions. But we think that Venn diagrams might be misleading since they are heavily dependent on the cutoff criteria and don't show quantitative information in contrast to heatmaps and correlation plots. We have added a statement to the discussion,

"Oct4 binds to only a small fraction of active ESC enhancers at early reprogramming stages. This indicates that Oct4 does not engage pluripotency genes on-target and could explain why iPSC generation is rather slow and inefficient."